# Understanding the Gain from Data Filtering in Multimodal Contrastive Learning

**Divyansh Pareek**    **Sewoong Oh**    **Simon S. Du**

*Paul G. Allen School of Computer Science and Engineering*

*University of Washington, Seattle, WA*

{dpareek,sewoong,ssdu}@cs.washington.edu

## Abstract

The success of modern multimodal representation learning relies on internet-scale datasets. Due to the low quality of a large fraction of raw web data, data curation has become a critical step in the training pipeline. Filtering using a trained model (i.e., teacher-based filtering) has emerged as a successful solution, leveraging a pre-trained model to compute quality scores. To explain the empirical success of teacher-based filtering, we characterize the performance of filtered contrastive learning under the standard bimodal data generation model. Denoting $\eta \in (0, 1]$ as the fraction of data with correctly matched modalities among $n$ paired samples, we utilize a linear contrastive learning setup to show a provable benefit of data filtering: $(i)$ the error without filtering is upper and lower bounded by $1/\eta\sqrt{n}$, and $(ii)$ the error with teacher-based filtering is upper bounded by $1/\sqrt{\eta n}$ in the large $\eta$ regime, and by $1/\sqrt{n}$ in the small $\eta$ regime.

## 1 Introduction

The seminal work of Radford et al. [29] introduced CLIP, a large-scale multimodal training paradigm that leverages contrastive learning on image and language modalities. This marked a significant advancement in general purpose representation learning that enabled unprecedented zero-shot downstream performance. A crucial factor in the success of CLIP and other vision-language models (VLMs) was the shift towards training on massive datasets [39], often comprising billions of image-text pairs scraped from the internet (e.g., LAION-5B [30] and DataComp-1B [11]). The sheer *quantity* of data unlocks the capability to learn robust representations [9]. However, due to the inherently noisy nature of web data, this introduces significant challenges regarding the *quality*, resulting in the need for data curation. Smaller but higher quality subsets of the data have been observed to result in better models than larger but noisier datasets [39, 25, 11]. Gadre et al. [11, Figure 2] observe that training on only a selected 30% of the dataset results in a better performing model than training on the full corpus. To handle such a significant fraction of low-quality data, data curation has become a critical step in modern internet-scale pretraining pipeline of foundation models [1].

For vision-language datasets, a number of methods have been introduced for data filtering [10, 35, 18, 8, 32, 22]. Among these, *teacher-based filtering*, where a pre-trained model is used to score samples and retain high-quality ones, has emerged as a particularly effective strategy [10, 35]. This approach marks a progression from earlier efforts which relied on heuristic-based filtering (e.g., the WIT400M dataset used in CLIP [29]). Subsequent and ongoing curation efforts have increasingly leveraged strong existing models, like CLIP itself, to refine datasets further [30, 11].

In the theory community, the success of CLIP models has been attributed to two factors: the choice of using a contrastive loss and the use of multimodal datasets. A series of modeling and analyses followed to explain the benefits from these two factors under various scenarios [24, 14, 33, 27, 17, 13, 6, 7].

39th Conference on Neural Information Processing Systems (NeurIPS 2025).

However, despite the empirical successes of data filtering in the CLIP training pipeline, a theoretical understanding of this phenomenon has been lacking. Our goal is to provide a deeper understanding of the benefits of using teacher-based data filtering in the CLIP training pipeline, i.e., multimodal representation learning with a contrastive loss. In particular, we aim to understand the benefits of data filtering against the baseline of contrastive learning without filtering, by focusing on one key parameter of interest: the *fraction* of high-quality data present. Using $\eta \in (0, 1]$ to denote the fraction of high-quality data pairs within the dataset, for both the filtering and no-filtering approaches, we ask the question: How does the quality of the learned representation behave as a function of $\eta$?

The choice of the data corruption model is crucial. In the related field of robust statistics, similar questions have been studied under adversarial corruptions. However, for large multimodal datasets, we posit that a *stochastic* corruption model is more relevant in capturing the nature of real data. For instance, in vision-language data, a significant portion of the misalignment arises randomly: images paired with irrelevant or tangentially related captions due to the processes of automated web scraping and the uncontrolled nature of internet data (see, e.g., [26, Figure 1] for examples). We adopt such a model (detailed in Section 3.1), where a fraction $\eta$ of pairs are correctly aligned, while the remaining $1 - \eta$ fraction has mismatched modalities. Under the stochastic corruption model of Section 3.1 and the contrastive learning setup of Section 3.2, we analyze the performance of teacher-based filtering (Figure 1c) and compare against the baseline of no filtering (Figure 1a).

**Contributions.** We demonstrate a provable benefit of data filtering. The error of the unfiltered contrastive learning with $n$ samples and $\eta$ clean fraction depends as $1/\eta\sqrt{n}$, as shown by an upper bound in Corollary 1 (result from Nakada et al. [24, Theorem 3.1]) and a lower bound in Proposition 1. On the other hand, for teacher-based filtering (Theorem 1, main result), the dependency on $\eta$ is improved to $1/\sqrt{\eta n}$ when $\eta$ is large, and to $1/\sqrt{n}$ when $\eta$ is small. Note that our result includes the training of the teacher model on the given dataset, i.e., we do not assume the existence of any strong pre-trained model. In Section 7, we empirically demonstrate the benefit of teacher-based data filtering in a synthetic experimental setting. Figure 3a verifies the $1/\eta$ dependence of the unfiltered contrastive learning, and the improved dependence achieved by the teacher-based filtering in two regimes, namely $1/\sqrt{\eta}$ for large $\eta$ and independent of $\eta$ for small $\eta$. Figure 3b restates the finding of Fang et al. [10, Figure 4] to show that the qualitative observation of improved $\eta$ dependence via filtering holds true even with real data.

## 2 Related work

Our theoretical investigation of data filtering builds upon existing analyses of multimodal contrastive learning [24, 14, 33]. In particular, Nakada et al. [24, Theorem 3.1] gives the rate for the unfiltered contrastive learning, and we study the rate with data filtering. The theory of contrastive learning (CL) has been studied in many other contexts [13, 6, 7, 17, 27]. Chen et al. [6] build a theoretical understanding for zero-shot transfer in CLIP-style models. Huang et al. [13] theoretically compare unimodal and multimodal CL, and Daunhawer et al. [7] study identifiability of the latent factors with the CL objective. We remark that the assumptions on the data generative model across these works are related but sometimes subtly different.

The practical need for data curation arises from the inherent noise in web-scale datasets used for training vision-language models [39, 25, 11] and increasingly, large language models [1, 20, 38, 34, 37]. In the multimodal context, numerous empirical techniques have been developed [10, 35, 18, 22, 8, 32], with community benchmarks like DataComp [11] facilitating systematic evaluation. Teacher-based filtering, the focus of our work, is a widely adopted and effective empirical strategy [11, 35, 39], but we note that other approaches have also been explored, in particular, editing bad data [26] (with some theoretical explanations [28, 41]). However, theoretical studies of data filtering are limited. Some works include the study of data selection under weak supervision in general statistical models [19], and selecting data during training [31].

## 3 Setup

Section 3.1 describes our model for multimodal data and the assumptions on the related parameters. Section 3.2 formulates the contrastive learning objective on data pairs from the model.

## 3.1 Bimodal data model

Building on recent theoretical work in multimodal contrastive learning [36, 14, 24], we assume the signal has a low-rank structure, while the noise is unstructured and dense. Adopting a *linear* generative model, the paired bimodal data, $x \in \mathbb{R}^d$ and $\widetilde{x} \in \mathbb{R}^{\widetilde{d}}$, is expressed as:

$$x = \mathbf{U}\, z + \xi \,, \quad \widetilde{x} = \widetilde{\mathbf{U}}\, \widetilde{z} + \widetilde{\xi} \,, \tag{1}$$

representing, for example, image and text in the case of vision-language data. Here $z, \widetilde{z} \in \mathbb{R}^r$ denote the latent variables lying in a shared $r$-dimensional space that captures the common underlying concept. The first terms $\mathbf{U}z$ and $\widetilde{\mathbf{U}}\widetilde{z}$ represent the signals of interest, residing in $r$-dimensional subspaces spanned by the columns of $\mathbf{U}$ and $\widetilde{\mathbf{U}}$, and the terms $\xi$ and $\widetilde{\xi}$ represent the dense noise. For simplicity, we assume that the maps $\mathbf{U} \in \mathbb{R}^{d \times r}$ and $\widetilde{\mathbf{U}} \in \mathbb{R}^{\widetilde{d} \times r}$ are composed of unit-norm orthogonal columns, fixing the scale of this problem.

We say a bimodal paired example is *corrupted* if the individual modalities do not correspond to the same latent concept. This models how a large fraction of image-text pairs found on the internet are corrupted by arbitrary captions that are unrelated to the content of the image. We formalize this in Assumption 1, with $\eta$ denoting the clean fraction. Figure 4 in Appendix A provides an illustration. For the noise, we assume a Gaussian distribution with a diagonal covariance (Assumption 2).

**Assumption 1** (Corruption model). *Let $z_1, z_2 \sim \mathcal{N}(0, \mathbf{I}_r)$ be two independent draws from the $r$-dimensional standard Gaussian. For an $\eta \in (0, 1]$, the joint distribution on $(z, \widetilde{z})$ is induced by*

$$w.p. \quad \eta \quad , \quad z = z_1 = \widetilde{z} \,, \ and \qquad \text{(Clean case)}$$
$$w.p. \ 1 - \eta \,, \quad z = z_1, \ \widetilde{z} = z_2 \,. \qquad \text{(Corrupted case)}$$

**Assumption 2** (Noise model). *The noise $\{\xi, \widetilde{\xi}\}$ are mutually independent and independent of $\{z, \widetilde{z}\}$, and are zero-mean Gaussian variables given by $\xi \sim \mathcal{N}\left(0, \gamma^{-1}\mathbf{I}_d\right)$ and $\widetilde{\xi} \sim \mathcal{N}\left(0, \widetilde{\gamma}^{-1}\mathbf{I}_{\widetilde{d}}\right)$.*

The signal is unit-scale in $r$-dimensions since $\|\mathbf{U}\| = 1$ and $\mathrm{Cov}(z) = \mathbf{I}_r$, hence the signal-to-noise ratios (SNRs) for the two modalities are $\gamma\,(r/d)$ and $\widetilde{\gamma}\,(r/\widetilde{d})$ respectively. This model is parametrized by $(\eta, \mathbf{U}, \widetilde{\mathbf{U}}, \gamma, \widetilde{\gamma}, r, d, \widetilde{d})$, and the aim is to recover $\mathbf{U}$ and $\widetilde{\mathbf{U}}$, given paired samples. This is a standard model in bimodal contrastive learning [36, 14, 24] and is inspired by the spiked covariance model [3, 40]. Consider an extreme case where the images are matched to randomly shuffled captions. This corresponds to $\eta = 0$, and recovering the subspaces $\mathbf{U}$ and $\widetilde{\mathbf{U}}$ becomes akin to two separate unimodal estimation problems, whose optimal (up to constants) error rate is known with tight upper and lower bounds [5, Eq. (9)]:

$$\mathbb{E}\left[\mathrm{ERR}(\hat{\mathbf{U}}, \hat{\widetilde{\mathbf{U}}})\right] \asymp \sqrt{\frac{r \max\left\{d\,\gamma^{-1}(1 + \gamma^{-1}), \ \widetilde{d}\,\widetilde{\gamma}^{-1}(1 + \widetilde{\gamma}^{-1})\right\}}{n}} \,, \tag{2}$$

where ERR is defined via the chordal distance between two subspaces in Eq. (4). This follows from the fact that $\mathbb{E}[xx^\top] = \mathbf{U}\mathbf{U}^\top + \gamma^{-1}\,\mathbf{I}_d$. The $\sqrt{d/n}$ dependence is expected from the concentration, the $\sqrt{r}$ dependence comes from the error metric being chordal (frobenius norm) as opposed to projection (spectral norm), and $\gamma^{-1/2}$ dependence captures how the error vanishes with high SNR. Refer to Appendix B.2 for a description of how to arrive at Eq. (2) using the result from Cai et al. [5, Eq. (9)]. When $\eta > 0$ fraction of data is correctly matched, our goal is to characterize the error rate achieved by the contrastive learning on the paired data and show that data filtering can improve the error rate compared to the baseline of no filtering.

**Notation.** For a matrix $Q = USV^\top$ and an integer $a$, let $\mathtt{SVD}_a(Q) = U_a S_a V_a^\top$ denote the projection of $Q$ onto its top-$a$ components. Let $\mathtt{lsv}(Q)$ denote the left singular vectors of $Q$, and $\mathtt{lsv}_a(Q)$ denote its top $a$ left singular vectors. Similarly, let $\mathtt{rsv}(Q)$ and $\mathtt{rsv}_a(Q)$ be defined for the right singular vectors. We use $O(.)$ to denote asymptotic upper bounds, and $\widetilde{O}(.)$ to denote upper bounds with only $\eta, n$ factors (omitting the dimension and SNR parameters). Similarly, we use the standard notation $\Omega(.), \omega(.)$ to denote asymptotic lower bounds. The notation $\gtrsim, \lesssim$ hides absolute constants, and we write $a \asymp b$ when $a \lesssim b$ and $a \gtrsim b$ holds simultaneously. Additionally, we will sometimes use the random variable $c \sim \mathrm{Ber}(\eta) \in \{0, 1\}$ to denote the (hidden) 'coin toss' in accordance with Assumption 1, with $c = 1$ denoting to the clean case.

## 3.2 Contrastive learning formulation

We utilize a linear contrastive learning framework from [24, 14]. By linear we mean $(i)$ the encoders that map the data, $x$ and $\widetilde{x}$, to the shared embedding space are linear, and $(ii)$ the contrastive loss computed on the embeddings is linear. This setting corresponds to the choice of $\epsilon = 0$, $\psi = \phi = \mathrm{Id}$ maps in Nakada et al. [24, eq 2.1], an equation that captures a more general contrastive loss framework. We refer the reader to Tian [33, Figure 1] for different contrastive learning setups achieved by different choices of $\psi$ and $\phi$.

Let $\mathbf{G} \in \mathbb{R}^{r \times d}$ and $\widetilde{\mathbf{G}} \in \mathbb{R}^{r \times \widetilde{d}}$ denote the learnable encoders for the input, $x$ and $\widetilde{x}$ respectively. Figure 5 in Appendix A provides a helpful visualization. The similarity score of a pair $(x, \widetilde{x})$ is computed as the inner product $\langle \mathbf{G}x, \widetilde{\mathbf{G}}\widetilde{x} \rangle$, which is widely used theoretically [24, 14, 15, 33] and empirically [29, 12]. The multimodal contrastive loss maximizes the similarity of observed pairs, while minimizing the similarity of 'generated' pairs. Given $n$ paired samples $\{(x_i, \widetilde{x}_i)\}_{i=1}^n$, the parameters $\mathbf{G}, \widetilde{\mathbf{G}}$ are learned by minimizing the $\rho$-regularized objective given by:

$$\mathcal{L}_\rho(\mathbf{G}, \widetilde{\mathbf{G}}) := \frac{1}{2n(n-1)} \Big( \sum_{i=1}^n \Big( \sum_{\substack{j=1 \\ j \neq i}}^n (s_{ij} - s_{ii}) + \sum_{\substack{j=1 \\ j \neq i}}^n (s_{ji} - s_{ii}) \Big) \Big) \; + \; R_\rho(\mathbf{G}, \widetilde{\mathbf{G}}) \,, \quad (3)$$

where $s_{ij} := \langle \mathbf{G}x_i, \widetilde{\mathbf{G}}\widetilde{x}_j \rangle$ for $i, j \in [n]$ is the similarity score, and $R_\rho(\mathbf{G}, \widetilde{\mathbf{G}}) := (\rho/2)\|\mathbf{G}^\top \widetilde{\mathbf{G}}\|_F^2$ is the regularizer with strength $\rho > 0$. The regularizer ensures that the learned parameters have finite norms. Indeed, Eq. (6) shows that this objective has a closed-form solution with a $1/\rho$ multiplier, which becomes infinite if $\rho = 0$. Note that CLIP [29] does not need a regularizer since the inner product is taken with *normalized* vectors (i.e. $(1/\|\mathbf{G}x\|)\mathbf{G}x$ instead of $\mathbf{G}x$). The parameters $\mathbf{G}$ and $\widetilde{\mathbf{G}}$ *assume* the knowledge of the latent dimension $r$ (since they are of sizes $r \times d$ and $r \times \widetilde{d}$). In practice, the latent dimension is typically a design choice and is therefore known at training time. Theoretically, assuming the latent dimension is known allows us to isolate the effects of data filtering from the separate, well-studied problem of subspace rank estimation (for e.g., in Cai et al. [5]).

Also note that this objective is in a *full-batch* setting, i.e. the entire $n \times n$ grid of similarities is computed to maximize the diagonals and minimize the off-diagonals. This does not cause computational issues since the objective has a closed-form solution, given by Eq. (6).

To measure the quality of a solution, we use the chordal distance between two subspaces in Definition 1. This is a standard measure of how well $\mathbf{G}, \widetilde{\mathbf{G}}$ recover $\mathbf{U}, \widetilde{\mathbf{U}}$ respectively [24, 14].

**Definition 1.** *The error metric for a learned embedding $\mathbf{G}, \widetilde{\mathbf{G}}$ is defined as*

$$\mathrm{ERR}(\mathbf{G}, \widetilde{\mathbf{G}}) := \max \left\{ \left\| \sin \Theta \left( \mathtt{rsv}\left(\mathbf{G}\right), \mathbf{U} \right) \right\|_F, \left\| \sin \Theta \left( \mathtt{rsv}\left(\widetilde{\mathbf{G}}\right), \widetilde{\mathbf{U}} \right) \right\|_F \right\} . \quad (4)$$

We note two points. First, the metric only considers the *right singular vectors*. This is because the essential information in $\mathbf{G}, \widetilde{\mathbf{G}}$ is contained in the right subspaces. Indeed, the loss in Eq. (3) is only affected by $\mathbf{G}^\top \widetilde{\mathbf{G}}$, which is preserved under the transformation $\mathbf{G} \leftarrow A\mathbf{G}$, $\widetilde{\mathbf{G}} \leftarrow A\widetilde{\mathbf{G}}$ for any orthonormal matrix $A$. Second, the metric uses the $\sin \Theta$ distance, which is a geometrically intuitive way to measure closeness between two subspaces (refer to Appendix B.1 for a background).

## 4 Baseline: unfiltered contrastive learning

We study the error rate of the unfiltered contrastive learning (Figure 1a). We show that the error is upper and lower bounded by $\widetilde{O}\left(1/\eta\sqrt{n}\right)$. The upper bound is given in Corollary 1, which is a result from Nakada et al. [24]. We show a matching lower bound in Proposition 1.

**Corollary 1** (Corollary of [24, Theorem 3.1])**.** *Given a dataset of pairs $\{(x_i, \widetilde{x}_i)\}_{i=1}^n$ generated i.i.d. according to the bimodal data model in Eq.* (1) *satisfying Assumptions 1 and 2, the solution of minimizing the contrastive loss in Eq.* (3) *satisfies with probability $1 - \exp(-\Omega(\max\{d, \widetilde{d}\}))$:*

$$\mathrm{ERR}(\mathbf{G}, \widetilde{\mathbf{G}}) \; \lesssim \; \frac{1}{\eta} \sqrt{\frac{r \max\{d, \widetilde{d}\}\left(1 + \gamma^{-1}\right)\left(1 + \widetilde{\gamma}^{-1}\right)}{n}} + \widetilde{O}\left(\frac{1}{n}\right) ,$$

*provided the number of samples $n \gtrsim (1/\eta^2) \max\{d, \widetilde{d}\}\left(1 + \gamma^{-1}\right)\left(1 + \widetilde{\gamma}^{-1}\right)$.*

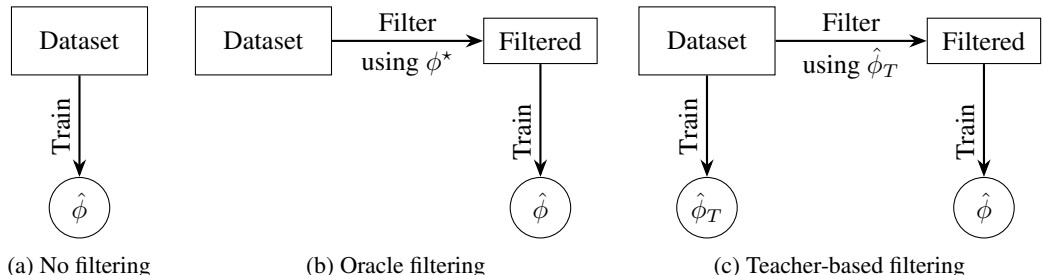

(a) No filtering   (b) Oracle filtering   (c) Teacher-based filtering

Figure 1: Our goal is to analyze the *Train-Filter-Train* approach illustrated in (c) and show that it improves upon the no filtering approach of (a). Here $\phi^\star$ denotes the ground-truth parameters and $\hat{\phi}$ denotes the learned version. In our setting, $\phi^* \equiv \{\mathbf{U}, \widetilde{\mathbf{U}}\}$ and $\hat{\phi} \equiv \{\mathbf{G}, \widetilde{\mathbf{G}}\}$.

**Remark 4.1** (Looseness in SNR parameters compared to Eq. (2)). *The dependence on SNR parameters $(\gamma, \widetilde{\gamma})$ in Corollary 1 is looser than the unimodal estimation counterpart in Eq. (2). As stated, the error upper bound in Corollary 1 does not become zero when $\gamma \to \infty$. We remark that this is an artifact of the analysis. Indeed a tighter analysis is possible that recovers a $\sqrt{\gamma^{-1}\widetilde{\gamma}^{-1}}$ term also in the upper bound, for instance, using the ideas in Cai et al. [5, Section 7], in particular [5, Eq. (39)].*

A complete proof of Corollary 1 is presented in Appendix D, which is largely a reconstruction from Nakada et al. [24] with some minor corrections. The analysis has three parts. First, the unregularized term of the contrastive loss in Eq. (3) simplifies to $\mathcal{L}_0(\mathbf{G}, \widetilde{\mathbf{G}}) = -\mathrm{Tr}\left(\mathbf{G}\mathbf{S}_n\widetilde{\mathbf{G}}^\top\right)$, where $\mathbf{S}_n \in \mathbb{R}^{d\times\widetilde{d}}$ denotes the cross-covariance matrix of the data, defined as

$$\mathbf{S}_n := \frac{1}{n-1}\sum_{i\in[n]}(x_i - \overline{x})\left(\widetilde{x}_i - \overline{\widetilde{x}}\right)^\top \approx \frac{1}{n}\sum_{i\in[n]}x_i\widetilde{x}_i^\top . \tag{5}$$

Second, the regularized contrastive loss, albeit nonconvex, admits a closed-form solution as the SVD of $\mathbf{S}_n$, given in Eq. (6). Due to this, we can directly analyze the solution without the need for optimization analysis.

$$\arg\min_{\mathbf{G},\widetilde{\mathbf{G}}} \mathcal{L}_\rho\left(\mathbf{G}, \widetilde{\mathbf{G}}\right) = \left\{\left(\mathbf{G}, \widetilde{\mathbf{G}}\right) \ \middle| \ \mathbf{G}^\top\widetilde{\mathbf{G}} = \frac{1}{\rho}\,\mathrm{SVD}_r\left(\mathbf{S}_n\right)\right\} . \tag{6}$$

The third key piece is concentration of $\mathbf{S}_n$. We show finite sample concentration of $\mathbf{S}_n$ in operator norm, namely *w.h.p.* $\left\|\mathbf{S}_n - \mathbf{S}\right\| \lesssim {}^{1}\!/\!{\rho\sqrt{n}}$, for the limiting quantity $\mathbf{S} = (\eta/\rho)\,\mathbf{U}\widetilde{\mathbf{U}}^\top$. Using a Davis-Kahan like result, we can translate the operator norm concentration to a distance between the angles of subspaces, for both left and right singular vectors, yielding *w.h.p.* $\mathrm{ERR}(\mathbf{G}, \widetilde{\mathbf{G}}) \lesssim {}^{1}\!/\!{\eta\sqrt{n}}$. Note the dependence on the regularization strength $\rho$ vanishes (as long as $\rho > 0$) due to its appearance in both the numerator (via op-norm concentration) and denominator (since the singular values of $\mathbf{S}$ scale as $\eta/\rho$). This sketch describes the ${}^{1}\!/\!{\eta}$ dependence of the unfiltered contrastive learning. In Proposition 1, we show that this dependence is tight. We present a proof of Proposition 1 in Appendix E by constructing a hard problem instance (parameterized by $\eta$).

**Proposition 1.** *Under the setting of Corollary 1, there is a class of problem instances with latent dimension $r = 1$ such that the error achieved by the minimizer of Eq. (3) is lower bounded (up to absolute constants) with probability $1 - \exp(-\Omega(\max\{d, \widetilde{d}\}))$ as:*

$$\mathrm{ERR}(\mathbf{G}, \widetilde{\mathbf{G}}) \ \gtrsim \ \frac{1}{\eta}\sqrt{\frac{\max\left\{d\,\gamma^{-1}, \widetilde{d}\,\widetilde{\gamma}^{-1}\right\}}{n}} .$$

## 5 Our approach: teacher-based filtering

In the previous section, we concluded that the unfiltered contrastive learning achieves a tight error dependence of ${}^{1}\!/\!{\eta}$. In this section, we ask: can filtering algorithms improve upon the $\eta$ dependency? Intuitively, we expect the answer to be yes, since filtering can identify corrupted samples and remove

them (increasing the clean fraction $\eta$). Indeed, if the filter could perfectly identify all clean samples, it would achieve a dependence of $1/\sqrt{\eta}$ (since this would be akin to the unfiltered contrastive learning with $\eta \leftarrow 1$ and $n \leftarrow \eta n$). We will now study the $\eta$ dependence of teacher-based filtering.

Teacher-based filtering, which follows a *Train-Filter-Train* approach, has proven to be a successful method in practice [10, 35]. In the first training step, a teacher model is trained on (potentially a part of) the dataset. In the filter step, (the remaining part of) the dataset is filtered by using the teacher to compute a similarity score to evaluate the quality of each sample. The filtering usually happens by selecting samples with score above a certain *threshold* $\theta \in \mathbb{R}$. In the second training step, a student model is trained on the filtered dataset. Refer to Figure 1c for an illustration. The student can be initialized at the teacher's solution, or even at a fresh random initialization. The intuition is that the teacher can extract useful signal from the dataset despite the presence of corrupted samples, which can help in identifying and discarding corrupted samples. Algorithm 1 describes this process in the setup of Section 3. The split of the dataset into two halves is for the convenience of analysis, by ensuring the filtering rule (which depends on the first half of samples and $\theta$) is independent of the samples being filtered (the second $n/2$ samples). We now state our main result.

---

**Algorithm 1** Teacher-based filtering in the setup of Section 3.

   **Input:** Dataset $D = \{(x_i, \widetilde{x}_i)\}_{i=1}^n$, Threshold $\theta \in \mathbb{R}$.
   **Step 1 (Train):** Obtain $\mathbf{G}_T, \widetilde{\mathbf{G}}_T$ by minimizing Eq. (3) on the first $n/2$ samples $\{(x_i, \widetilde{x}_i)\}_{i \leq n/2}$.
   **Step 2 (Filter):** Create $D_{\text{filt}}(\theta)$ from $\{(x_i, \widetilde{x}_i)\}_{i > n/2}$ by retaining sample $i$ iff $\langle \mathbf{G}_T x_i, \widetilde{\mathbf{G}}_T \widetilde{x}_i \rangle > \theta$.
   **Step 3 (Train):** Output $\mathbf{G}(\theta), \widetilde{\mathbf{G}}(\theta)$ by minimizing Eq. (3) on $D_{\text{filt}}(\theta)$.

---

**Theorem 1.** *Under the model in Eq.* (1) *satisfying Assumptions 1 and 2 with $r \geq 2$, there exists a threshold $\theta^* \in \mathbb{R}$ such that, given a dataset of pairs $\{(x_i, \widetilde{x}_i)\}_{i=1}^n$ generated i.i.d. according to the model, the output of Algorithm 1 satisfies with probability $1 - \exp(-\Omega(\max\{d, \widetilde{d}\}))$:*

$$\mathrm{ERR}\left(\mathbf{G}(\theta^*), \widetilde{\mathbf{G}}(\theta^*)\right) \;\lesssim\; \min\{T_{0.5}, T_0\} \;,$$

*provided $n \gtrsim (1/\eta^2) \max\{d, \widetilde{d}\} \left(1 + \gamma^{-1}\right) \left(1 + \widetilde{\gamma}^{-1}\right)$. Here $T_{0.5}, T_0$ are defined as*

$$T_{0.5} = \sqrt{\frac{r \, \max\{d, \widetilde{d}\} \, \mathrm{poly}\left(\gamma^{-1}, \widetilde{\gamma}^{-1}\right)}{\eta n}} + \widetilde{O}\left(\frac{1}{n}\right) \;,$$

$$T_0 = \sqrt{\frac{r^3 \, \max\{d, \widetilde{d}\} \, \mathrm{poly}\left(\gamma^{-1}, \widetilde{\gamma}^{-1}\right)}{n}} + \widetilde{O}\left(\frac{1}{n}\right) \;.$$

We provide a full proof in Appendix G, and discuss the sketch in Section 6. Certain observations are in order. First, we see two regimes of behavior. The error behaves as $1/\sqrt{\eta}$ for large values of $\eta$, and becomes independent of $\eta$ for small values of $\eta$ (note that $\eta$ still needs to large enough to satisfy the requirement of $n \gtrsim 1/\eta^2$ for theorem to be valid). Both these regimes exhibit a better dependence on $\eta$ than the unfiltered contrastive learning's rate of $1/\eta$. From the expressions, we note that the switch between the regimes happens at $\eta = 1/r^2$ (up to constants). Second, this result is stated for the optimal filtering threshold $\theta^*$. The optimal choice of this hyperparameter depends on the problem quantities, particularly $n$ and $\eta$. Understanding this dependence is an interesting direction of research, but outside the scope of the current work. Our analysis considers two fixed choices of $\theta$ that recover each of the regimes. We also present a small experiment on varying the filtering threshold $\theta$ in the vicinity of $\theta^*$ in Appendix H. Third, we remark that it remains an interesting research question to study whether an improved dependence on $\eta$ (at least something better than $1/\eta$) can be achieved with a single training loop on the data (as the teacher-based filtering is a two-step training process).

It is perhaps surprising that the error can become independent of the clean fraction $\eta$, which is better than the oracle rate of $1/\sqrt{\eta}$. This counter intuitive benefit stems from the use of the inner product to compute similarities (Section 3.2) on the corruption model given by Assumption 1. Owing to this, the distribution of the similarity scores before filtering follows a very typical structure, explained in Figure 2. Filtering can retain samples from the right tail of the noisy score distribution $\mathcal{D}_0$, and these samples provide useful signal to recover the ground-truth $\mathbf{U}$ parameter. Finally, we remark on the assumptions needed for this result. Assumption 2 makes this setting somewhat special, since Nakada

et al. [24] allow for a general covariance $\Sigma_\xi, \Sigma_{\widetilde{\xi}}$ (with bounded norms) on the noise. Handling a more general noise covariance is trivial for unfiltered contrastive learning, but significantly more challenging in the case of filtering. We argue that Assumption 2 preserves the essential characteristics of the problem though, while simplifying the analysis of filtering. In the following section, we discuss the proof ideas in more detail.

# 6    Analysis of the filtering algorithm

In this section, we describe the main ideas behind the proof of Theorem 1. In Section 6.1, we study the distribution of the scalar score used for filtering samples. In Section 6.2, we use the score characterization to understand filtering by thresholding on the scores.

## 6.1    The score used for filtering

For a sample $(x, \widetilde{x})$, let $S(x, \widetilde{x}; \mathbf{A})$ for a matrix $\mathbf{A} \in \mathbb{R}^{d \times \widetilde{d}}$ denote the score of the sample, defined in Eq. (7). This scalar score is meant to capture the quality of the sample $(x, \widetilde{x})$. Treating $(x, \widetilde{x})$ as a random i.i.d. sample from the model in Section 3.1, we characterize the distribution of the score. Note that the teacher-based filtering is simply using $\mathbf{A} := \mathbf{G}_\mathrm{T}^\top \widetilde{\mathbf{G}}_\mathrm{T}$ to score the data (the subscript is used to denote the teacher's parameters). To understand teacher-based filtering, an intermediate step will be to understand filtering using an 'oracle' which has access to the ground-truth problem parameters (refer to Figure 1b). The oracle scores data using $\mathbf{A} := \mathbf{U}\widetilde{\mathbf{U}}^\top$, given in Eq. (8). Since $\mathbf{G}_\mathrm{T}^\top \widetilde{\mathbf{G}}_\mathrm{T} \to (\eta/\rho)\,\mathbf{U}\widetilde{\mathbf{U}}^\top$ as the number of samples $n \to \infty$, we expect the teacher filtering to resemble the oracle filtering in the large $n$ regime. The positive scaling factor of $\eta/\rho$ does not affect threshold-based filtering, as the ordering of samples remains unchanged.

$$S(x, \widetilde{x}; \mathbf{A}) := x^\top \mathbf{A} \widetilde{x} \,, \tag{7}$$

$$S(x, \widetilde{x}; \mathbf{U}\widetilde{\mathbf{U}}^\top) = (\mathbf{U}\,z + \xi)^\top \mathbf{U}\widetilde{\mathbf{U}}^\top \left(\widetilde{\mathbf{U}}\,\widetilde{z} + \widetilde{\xi}\right) = z^\top \widetilde{z} + \underbrace{z^\top \widetilde{\mathbf{U}}^\top \widetilde{\xi} + \xi^\top \mathbf{U}\,\widetilde{z} + \xi^\top \mathbf{U}\widetilde{\mathbf{U}}^\top \widetilde{\xi}}_{\text{zero-mean terms involving } (\xi, \widetilde{\xi})} \,. \tag{8}$$

**Remark 6.1** (Two versions of oracle). *There are two possibilities for an 'oracle' in this setup. The first kind has access to the ground-truth problem parameters, which is what we study. The second kind has access to the clean/corrupted status of each sample. The second kind can trivially achieve an error dependence of $1/\sqrt{\eta n}$ by choosing to only use the clean samples.*

Recalling Assumption 1, since $\widetilde{z} = z$ for clean samples, the score in Eq. (8) is defined through the independent randomness in $z, \xi, \widetilde{\xi}$. For corrupted samples, it is defined via the independent randomness in all $z, \widetilde{z}, \xi, \widetilde{\xi}$. We characterize the distribution in both cases, detailed in Appendix F. The main observations are illustrated in Figure 2a. $\mathcal{D}_0$ denotes the distribution of the score in the corrupted case, with mean $\mu(\mathcal{D}_0) = 0$ (since $z, \widetilde{z}$ are independent), and variance $\sigma_0^2 = r\left(1 + \gamma^{-1}\right)\left(1 + \widetilde{\gamma}^{-1}\right)$. Similarly, $\mathcal{D}_1$ denotes the distribution in the clean case, with mean $\mu(\mathcal{D}_1) = r$ (since $z = \widetilde{z}$ leading to a squared term), and variance $\sigma_1^2 = r + r\left(1 + \gamma^{-1}\right)\left(1 + \widetilde{\gamma}^{-1}\right)$. Note that $\sigma_0^2 \leq \sigma_1^2 \leq 2\sigma_0^2$.

Since clean and corrupted data are mixed with $\eta, 1 - \eta$ proportions, the score of a generic sample from the population is given by the mixture distribution $\mathcal{D} := \eta\mathcal{D}_1 + (1 - \eta)\mathcal{D}_0$. Figure 2 provides an illustration of the score distribution $\mathcal{D}$. Due to i.i.d. data, the oracle filtering algorithm's scores are $n$ i.i.d. draws from $\mathcal{D}$. The filtering threshold $\theta$ can be picked in various ways, leading to various algorithms for filtering. The threshold $\theta \to -\infty$ corresponds to no filtering.

**Remark 6.2.** *Since $\sigma_0 \leq \sigma_1 = \sqrt{2r\left(1 + \gamma^{-1}\right)\left(1 + \widetilde{\gamma}^{-1}\right)}$, the condition $\gamma, \widetilde{\gamma} = \omega(\sqrt{r})$ ensures that $r/\sigma_1 = \omega(1)$, leading to a separation between the modes of $\mathcal{D}_0$ and $\mathcal{D}_1$. In this case, the clean and corrupted data become well-separated via the oracle score $S(x, \widetilde{x}; \mathbf{U}\widetilde{\mathbf{U}}^\top)$.*

## 6.2    Analysis of thresholding on the score distribution

In this section, we discuss an analysis for the oracle filtering algorithm (Figure 1b), which captures the main conceptual ideas of data filtering in the setup of Section 3. The proof for the teacher-based filtering (Theorem 1) is given in Appendix G, which uses the ideas from this section, along with the

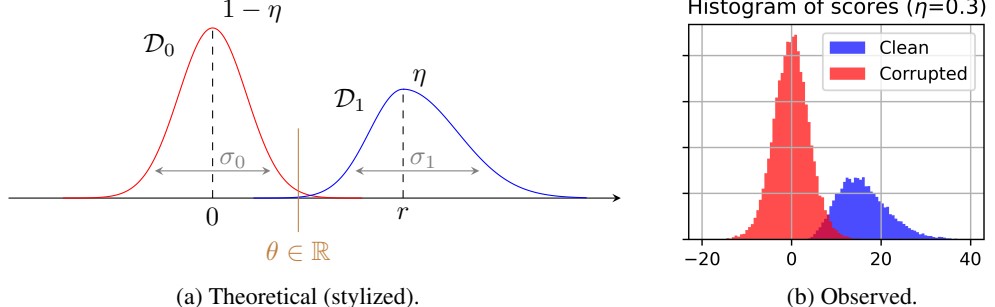

| (a) Theoretical (stylized). | (b) Observed. |

Figure 2: Distribution of the oracle score $S(x, \widetilde{x}; \mathbf{U}\widetilde{\mathbf{U}}^\top)$ is given by the mixture of $\mathcal{D}_0, \mathcal{D}_1$ with weights $(1 - \eta), \eta$ respectively. Here $\sigma_0^2, \sigma_1^2$ depend on parameters $r, \gamma, \widetilde{\gamma}$. The threshold $\theta \in \mathbb{R}$ is used to filter the datapoints (score $> \theta$ are retained, others are discarded). Subfigure (b) shows the observed histogram in a synthetic setting for $n = 50000$ samples with $r = 16, \gamma = \widetilde{\gamma} = 10^4$.

operator norm concentration in Corollary 1 to bound the deviation caused by the difference between the teacher scores and the oracle scores. Given a dataset $\{(x_i, \widetilde{x}_i)\}_{i=1}^n$, let $n_{\text{sel}}(\theta)$ denote the number of samples retained after oracle filtering, and let $I_{\text{sel}}(\theta) \subseteq [n]$ denote the indices of the samples selected, defined by the condition $i \in I_{\text{sel}}(\theta) \iff S(x_i, \widetilde{x}_i; \mathbf{U}\widetilde{\mathbf{U}}^\top) > \theta$. Analogous to Eq. (5), we define $\mathbf{S}_n(\theta)$ to be the empirical cross-covariance of the filtered data, given by Eq. (9).

$$\mathbf{S}_n(\theta) := \frac{1}{n_{\text{sel}}(\theta) - 1} \sum_{i \in I_{\text{sel}}(\theta)} (x_i - \overline{x}) \left(\widetilde{x}_i - \overline{\widetilde{x}}\right)^\top , \qquad \overline{\mathbf{S}}_n(\theta) := \frac{1}{nP(\theta)} \sum_{i \in I_{\text{sel}}(\theta)} x_i \widetilde{x}_i^\top , \quad (9)$$

$$\mathbf{S}(\theta) := \mathbb{E}\left[\overline{\mathbf{S}}_n(\theta)\right] = \mathbb{E}[\, x\widetilde{x}^\top \mid S(x, \widetilde{x}; \mathbf{U}\widetilde{\mathbf{U}}^\top) > \theta \,] . \qquad (10)$$

Observe that similar to Eq. (6), the closed-form solution of the optimization holds even on the filtered dataset. The step that changes is the concentration, namely, the characterization of how $\mathbf{S}_n(\theta)$ concentrates as $n$ increases, according to the distributions of the involved random quantities. In the following, we argue that $\mathbf{S}_n(\theta)$ concentrates to $\mathbf{S}(\theta)$, given by Eq. (10), and characterize the behavior of $\mathbf{S}(\theta)$ to recover a guarantee akin to Theorem 1.

**Notation**. We set up some useful notation on the score distributions $\mathcal{D}_0, \mathcal{D}_1$ from Figure 2a. For any $a \in \mathbb{R}$, let $P_0(a) = \mathbb{P}_{Z \sim \mathcal{D}_0}(Z > a)$ and $P_1(a) = \mathbb{P}_{Z \sim \mathcal{D}_1}(Z > a)$ denote the probabilities of the upper tails of the corrupted and clean parts respectively, and let $P(a) = \mathbb{P}_{Z \sim \mathcal{D}}(Z > a) = \eta P_1(a) + (1 - \eta)P_0(a)$ denote the probability of selection from the mixture distribution. Similarly for expectations, define $E_0(a) := \mathbb{E}_{Z \sim \mathcal{D}_0}[Z|Z > a], E_1(a) := \mathbb{E}_{Z \sim \mathcal{D}_1}[Z|Z > a]$.

**Concentration of $\mathbf{S}_n(\theta)$ to $\mathbf{S}(\theta)$.** We claim that $\mathbf{S}_n(\theta) \approx \overline{\mathbf{S}}_n(\theta)$ by using two approximations. First, the un-centered version in $\overline{\mathbf{S}}_n(\theta)$ approximates the centered version in $\mathbf{S}_n(\theta)$. Second, although $n_{\text{sel}}(\theta)$ is a random quantity, it concentrates around $nP(\theta)$. We formally bound the error due to both these approximations in the full proof. Since the filtering threshold $\theta$ is chosen independent of the samples being filtered, the selected samples satisfy the *i.i.d property under the conditional law* of the score being above $\theta$. This allows us to show that the approximate version, $\overline{\mathbf{S}}_n(\theta)$, concentrates around its expectation, $\mathbf{S}(\theta)$, by bounding the spectral norm of the difference via a Matrix-Bernstein type inequality. Overall, we get

$$\textit{w.p. } 1 - \exp(-\Omega(\max\{d, \widetilde{d}\})) , \quad \|\mathbf{S}_n(\theta) - \mathbf{S}(\theta)\| \lesssim \sqrt{\frac{\max\{d, \widetilde{d}\}}{nP(\theta)}} . \qquad (11)$$

**Analysis of $\mathbf{S}(\theta)$ and $P(\theta)$.** Simplifying $\mathbf{S}(\theta)$ reveals that it is simply a scaled version of $\mathbf{U}\widetilde{\mathbf{U}}^\top$, with the scaling coefficient depending on $\theta$ described by the conditional expectations $E_0(\theta)$ and $E_1(\theta)$. Concretely, $\mathbf{S}(\theta) = \frac{1}{r} (\eta E_1(\theta) + (1 - \eta) E_0(\theta)) \mathbf{U}\widetilde{\mathbf{U}}^\top$. Owing to this, the application of a Davis-Kahan result on Eq. (11) will dictate the guarantee of recovering $\mathbf{U}, \widetilde{\mathbf{U}}$ for the filtering algorithm. The error behaves as:

$$\text{ERR} \propto \frac{r}{(\eta E_1(\theta) + (1 - \eta) E_0(\theta))} \frac{1}{\sqrt{\eta P_1(\theta) + (1 - \eta) P_0(\theta)}} \frac{1}{\sqrt{n}} .$$

The behavior of the functions $E_0(\theta), E_1(\theta)$ and $P_0(\theta), P_1(\theta)$ precisely quantify this rate. As a sanity check, setting $\theta = -\infty$ recovers the $1/\eta$ behavior of the unfiltered contrastive learning, as $E_1(-\infty) = r, E_0(-\infty) = 0$ and $P_1(-\infty) = 1, P_0(-\infty) = 1$. Since $E_0(\theta), E_1(\theta)$ are increasing functions in $\theta$, whereas $P_0(\theta), P_1(\theta)$ are decreasing, we observe a tradeoff. A larger threshold $\theta$ results in larger conditional expectations $E_0(\theta), E_1(\theta)$, but smaller probabilities of selection $P_0(\theta), P_1(\theta)$. In the Appendix, we formally characterize this behavior, involving calculations on the conditional expectations and probabilities of the Gaussian distribution. Here, we discuss the two choices of $\theta$ that recover the two regimes of the filtering behavior. The threshold $\theta = 0$ results in $E_1(0) \geq r$, $E_0(0) \geq 2/\pi$ and $P_1(0) \geq 0.5$, $P_0(0) = 0.5$, recovering the independent of $\eta$ regime. And the threshold $\theta = r/2$ results in $E_1(r/2) \geq r$, $E_0(r/2) \geq r/2$ (using a trivial lower bound for the conditional expectation), and $P_1(r/2) \geq 0.5$ (but $P_0(r/2)$ is small), recovering the $1/\sqrt{\eta}$ regime. The optimal $\theta^*$ will achieve a rate better than the above two special points, hence the upper bound on the error is given by the $\min$ of these two regimes, recovering the upper bound in Theorem 1.

## 7 Experiments

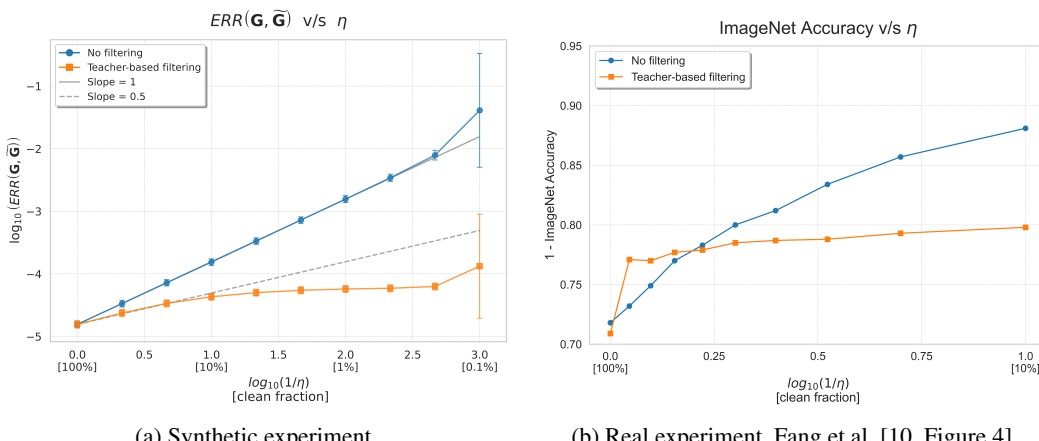

(a) Synthetic experiment.    (b) Real experiment, Fang et al. [10, Figure 4].

Figure 3: **(a).** Observed dependence of $\mathrm{ERR}(\mathbf{G}, \widetilde{\mathbf{G}})$ on $\eta$ for a synthetic experiment. The error of the unfiltered contrastive learning follows a $1/\eta$ dependence, but deviates for small $\eta$ since the requirement of $n \gtrsim 1/\eta^2$ in Corollary 1 gets violated. The error of the filtering algorithm follows a $1/\sqrt{\eta}$ dependence in the large $\eta$ (or small $1/\eta$) regime, and an independent of $\eta$ dependence in the small $\eta$ regime. Going beyond to even smaller $\eta$ causes deviations since Theorem 1 also requires $n \gtrsim 1/\eta^2$. The teacher-based filtering is with the threshold $\theta = 0$. **(b).** A similar trend on real data observed by Fang et al. [10]. The y-axis shows $1 - \mathrm{Accuracy}$, which is different than the error metric in (a). However, we note that the qualitative trend of the orange line having a smaller slope than the blue line still holds. Numbers from Fang et al. [10] are reproduced with permission.

In this section, we validate our theoretical results with a synthetic setup. With parameters $d = 10, \widetilde{d} = 8, r = 4$, and SNR $\gamma = \widetilde{\gamma} = 10^4$, and with randomly generated $\mathbf{U}, \widetilde{\mathbf{U}}$, we generate $n = 10M$ samples according to the model in Section 3.1, and vary the clean fraction $\eta$. We experiment over 10 values of $\eta$ geometrically decreasing from 1 to $10^{-3}$. This experiment was run on a cluster of 50 CPUs with 500G memory, and required less than 10 minutes. Figure 3a shows the result and discusses the observations, which validate Corollary 1 and Theorem 1. To extend these observations to real settings, the main limitations are posed by the modeling assumptions in Section 3. Despite the limitations, Figure 3b shows evidence that the qualitative conclusions drawn from the theory hold with real image-text data too. Concretely, it shows that the downstream model performance on reducing the clean fraction $\eta$ degrades more steeply without data filtering.

## 8 Conclusion and Broader Impacts

This paper presents a theoretical investigation into teacher-based data filtering for multimodal contrastive learning with stochastically corrupted data. We rigorously establish its benefit, demonstrating

that filtering improves the error dependence on the clean data fraction, $\eta$, from $1/\eta$ (no filtering) to $1/\sqrt{\eta}$ in the large $\eta$ regime, and perhaps surprisingly, to independent of $\eta$ in the small $\eta$ regime. The latter finding suggests that teacher-based filtering can be particularly beneficial when data quality is low, achieving performance independent of the initial clean fraction. Our results provide a formal basis for the empirical success of teacher-based data filtering. The main limitations are posed by the assumption of linearity in Section 3, and the model of stochastic corruptions in Assumption 1. Future work could explore the optimal selection of filtering thresholds and investigate whether similar gains can be achieved with one-step filtering algorithms.

Our contributions are largely on the theoretical understanding of data filtering, and its potential benefits. At a high-level, effective data filtering can reduce the compute cost needed to train models, which has positive potential impacts through more judicious use of energy resources. On the other hand, data filtering can exacerbate the biases present in a dataset by selecting certain subpopulations more than the others. If this goes unchecked, it has potential negative impacts to society.

## Acknowledgements

SSD acknowledges the support of NSF DMS 2134106, NSF IIS 2143493, the Sloan Fellowship, and the AI2050 program at Schmidt Sciences. SO acknowledges the support of NSF grants no. 2112471, 2229876, and 2505865.

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

## A Additional Illustrations

In this section, we provide some useful illustrations. Figure 4 illustrates the corruption model described in Assumption 1. Figure 5 illustrates the linear maps $\mathbf{G}, \widetilde{\mathbf{G}}$ used to generate the embeddings from observed data (according to the model in Fig 4). Figure 6 accompanies Remark A.1.

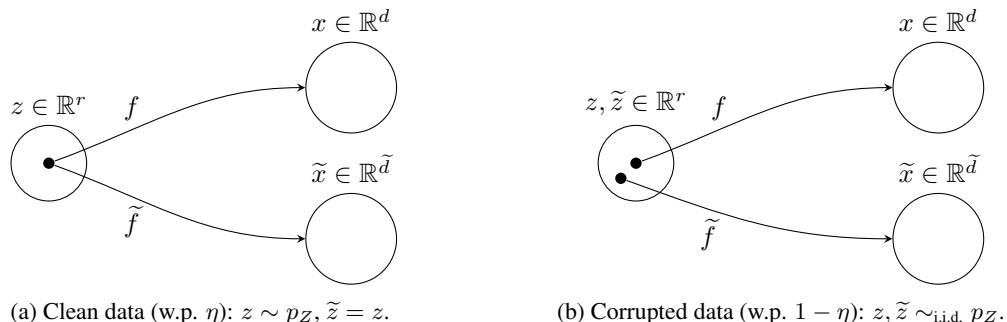

(a) Clean data (w.p. $\eta$): $z \sim p_Z, \widetilde{z} = z$.  (b) Corrupted data (w.p. $1 - \eta$): $z, \widetilde{z} \sim_{\text{i.i.d.}} p_Z$.

Figure 4: Model for stochastic corruptions. In this work, the forward maps $f, \widetilde{f}$ are linear (refer to Eq. (1)) and the latent distributions are Gaussians.

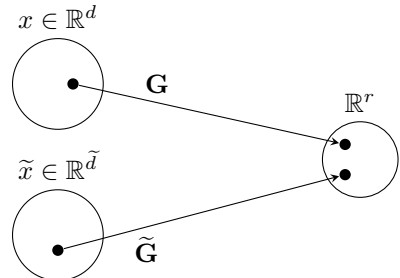

Figure 5: On seeing multimodal data $(x, \widetilde{x})$, linear maps $\mathbf{G}, \widetilde{\mathbf{G}}$ (learnable parameters) create the embeddings that lie in $\mathbb{R}^r$ (the knowledge of $r$, the true latent dimension, is assumed). The similarity is measured with the inner product $\langle \mathbf{G}x, \widetilde{\mathbf{G}}\widetilde{x} \rangle$.

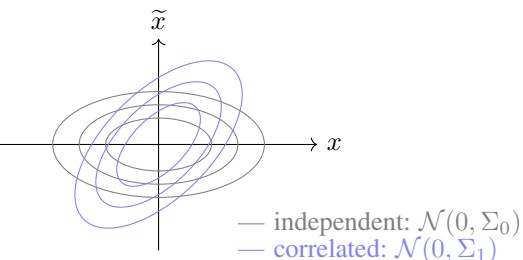

— independent: $\mathcal{N}(0, \Sigma_0)$
— correlated: $\mathcal{N}(0, \Sigma_1)$

Figure 6: Illustration of the joint distribution of $(x, \widetilde{x})$. The overall distribution is a mixture of two zero mean Gaussians: the independent case (w.p. $1 - \eta$) and the correlated case (w.p. $\eta$).

**Remark A.1.** *The distribution of $(x, \widetilde{x}) \in \mathbb{R}^{d+\widetilde{d}}$ from Section 3.1 is a mixture of two zero-mean Gaussians. With weight $\eta$, the covariance matrix is $\Sigma_1$ (for $c = 1$, i.e. the clean case). With weight $1 - \eta$, the covariance is $\Sigma_0$ (for $c = 0$). Figure 6 provides an illustration.*

$$\Sigma_1 = \begin{bmatrix} \mathbf{U}\mathbf{U}^\top + \gamma^{-1}\mathbf{I}_d & \mathbf{U}\widetilde{\mathbf{U}}^\top \\ \widetilde{\mathbf{U}}\mathbf{U}^\top & \widetilde{\mathbf{U}}\widetilde{\mathbf{U}}^\top + \widetilde{\gamma}^{-1}\mathbf{I}_{\widetilde{d}} \end{bmatrix}, \; \Sigma_0 = \begin{bmatrix} \mathbf{U}\mathbf{U}^\top + \gamma^{-1}\mathbf{I}_d & \mathbf{0} \\ \mathbf{0} & \widetilde{\mathbf{U}}\widetilde{\mathbf{U}}^\top + \widetilde{\gamma}^{-1}\mathbf{I}_{\widetilde{d}} \end{bmatrix}.$$

# B Background

This section covers some useful background concepts.

## B.1 Measuring the distance between subspaces

The concept of principal angles provides a geometrically intuitive way to measure the closeness between two subspaces. Let $\mathcal{X}$ and $\mathcal{Y}$ be two $r$-dimensional subspaces within a larger Euclidean space $\mathbb{R}^d$. There exist $r$ principal angles $0 \leq \theta_1 \leq \theta_2 \leq \cdots \leq \theta_r \leq \pi/2$ that describe the relative orientation of these subspaces.

- $\theta_1$ represents the *smallest* possible angle between any two unit vectors $x \in \mathcal{X}$ and $y \in \mathcal{Y}$.
- Subsequent angles $\theta_k$ capture the minimum angles within directions orthogonal to those defining the previous angles $\theta_1, \ldots, \theta_{k-1}$.
- The cosines $\cos(\theta_i)$ measure the alignment (1 means aligned, 0 means orthogonal within that principal direction), while the sines $\sin(\theta_i)$ measure the separation or angle.

To aggregate this information into a single distance metric, we often use the frobenius norm of the sine of the principal angles, denoted $\| \sin \Theta(\mathcal{X}, \mathcal{Y}) \|_F$. It is defined as

$$\| \sin \Theta(\mathcal{X}, \mathcal{Y}) \|_F = \sqrt{\sum_{i=1}^{r} \sin^2(\theta_i)} \ .$$

This metric provides an overall measure of the difference between the subspaces. It's zero if and only if $\mathcal{X} = \mathcal{Y}$ (since all $\theta_i = 0$), and it increases as the subspaces diverge.

Computing this metric relies on matrix operations involving orthonormal bases for the subspaces. Let $\mathbf{X} \in \mathbb{R}^{d \times r}$ be a matrix whose columns form an orthonormal basis for $\mathcal{X}$ (so $\mathbf{X}^\top \mathbf{X} = \mathbf{I}_r$). Similarly, let $\mathbf{Y} \in \mathbb{R}^{d \times r}$ be a matrix with orthonormal columns forming a basis for $\mathcal{Y}$. The distance metric $\| \sin \Theta(\mathcal{X}, \mathcal{Y}) \|_F$ can be computed using $\mathbf{X}$ and $\mathbf{Y}$ via the following formula

$$\| \sin \Theta(\mathcal{X}, \mathcal{Y}) \|_F = \left\| \mathbf{X}_\perp^\top \mathbf{Y} \right\|_F \ .$$

Here, $\mathbf{X}_\perp$ is *any* $d \times (d-r)$ matrix such that its columns form an orthonormal basis for the orthogonal complement of $\mathcal{X}$, denoted $\mathcal{X}^\perp$. This means that the combined matrix $[\mathbf{X} \ \mathbf{X}_\perp]$ must be a $d \times d$ orthogonal matrix. Notationally, we often just write $\| \sin \Theta(\mathbf{X}, \mathbf{Y}) \|_F$ instead of using $\mathcal{X}, \mathcal{Y}$.

## B.2 Optimal unimodal estimation rates in the spiked covariance model

Eq. (1) uses the well-known spiked covariance model for each of the two modalities, originally introduced by Johnstone [16] and well-studied in the literature [3, 40, 5]. Cai et al. [5] establish optimal (minimax) estimation rates for the covariance matrix (i.e. $\mathbf{U}\mathbf{U}^\top + \gamma^{-1}\mathbf{I}_d$) and the principal subspace (i.e. $\mathbf{U}$) in a more general *sparse* spiked covariance model. In particular, [5, Eq. (7)] describes the minimax rate for covariance estimation, and [5, Eq. (9)] describes the minimax rate for subspace estimation. We use the latter result to get Eq. (2). Since the problem of subspace estimation is invariant to scaling, we instantiate [5, Eq. (9)] for the estimation of data with covariance $\gamma \mathbf{U}\mathbf{U}^\top + \mathbf{I}_d$ (since $\sigma = 1$ is assumed in [5, Eq. (1)] to fix the problem scaling). With this, the paramaters map as $\lambda = \gamma$, $p = d$ and $k = d$ (since our model is not sparse). This establishes a rate (up to constants) of $\sqrt{\frac{d\gamma^{-1}(1+\gamma^{-1})}{n}}$ for the estimation in 2-norm. An additional factor of $\sqrt{r}$ appears since we use the Frobenius-norm (i.e. the chordal distance in Definition 1), and Eq. (2) follows.

# C Lemmas

This section presents Lemmas used in the proofs. The first three Lemmas are standard results in the literature, and we include them without proof.

**Lemma 1** (Weyl's Inequality). *For matrices $A, B \in \mathbb{R}^{m \times n}$, let $p = \min(m,n)$ and let $\sigma_1(M) \geq \sigma_2(M) \geq \cdots \geq \sigma_p(M) \geq 0$ denote the singular values for $M \in \{A, B\}$. Then, for all $j = 1, \ldots, p$, it holds that*

$$|\sigma_j(A) - \sigma_j(B)| \leq \|A - B\|_2 \ .$$

**Lemma 2** (Wedin's Theorem). *Let $A, \hat{A} \in \mathbb{R}^{m \times n}$ be matrices of the same size. Let $r \leq \min(m, n)$ be the rank of both $A$, $\hat{A}$, and let the SVDs be $A = U\Sigma V^\top$ and $\hat{A} = \hat{U}\hat{\Sigma}\hat{V}^\top$. Let $\sigma_r(A) > 0$ denote the $r^{th}$ singular value of $A$, and assume $\sigma_r(A) > \|\hat{A} - A\|_2$. Then it holds that:*

$$\left\| \sin \Theta(\hat{U}, U) \right\|_F \leq \frac{\|\hat{A} - A\|_F}{\sigma_r(A) - \|\hat{A} - A\|_2} \ ,$$

$$\left\| \sin \Theta(\hat{V}, V) \right\|_F \leq \frac{\|\hat{A} - A\|_F}{\sigma_r(A) - \|\hat{A} - A\|_2} \ .$$

**Lemma 3** (Whittle's Inequality). *Let $X_1, X_2, \ldots$ be a sequence of independent random variables such that: $(i)$ $\mathbb{E}[X_k] = 0$ for all $k \geq 1$, and $(ii)$ the distribution of each $X_k$ is symmetric about zero (i.e., $X_k$ and $-X_k$ have the same distribution). Let $S_n = \sum_{k=1}^n X_k$ be the partial sum (with $S_0 = 0$). If $\phi : \mathbb{R} \to \mathbb{R}$ is a convex function such that $\phi(0) = 0$, then the sequence $\mathbb{E}[\phi(S_n)]$ is non-decreasing in $n$. That is, for all $n \geq 1$:*

$$\mathbb{E}[\phi(S_n)] \geq \mathbb{E}[\phi(S_{n-1})] \ .$$

**Lemma 4.** *Let $A, B \in \mathbb{R}^{m \times n}$ with $\mathrm{rank}(A) = r \geq 1$. If $\|A - B\|_2 < \sigma_r(A)$, then for every $t \in [0, 1]$, it holds that $\mathrm{rank}\big((1 - t)A + tB\big) \geq r$.*

*Proof.* Let $X_t = (1 - t)A + tB$. For any matrices $M, N$ and any $k$,

$$\sigma_k(M) \geq \sigma_k(N) - \|M - N\|_2,$$

which follows Lemma 1. Applying this with $M = X_t$, $N = A$, and $k = r$,

$$\sigma_r(X_t) \geq \sigma_r(A) - \|X_t - A\|_2 = \sigma_r(A) - t\|A - B\|_2 \geq \sigma_r(A) - \|A - B\|_2 > 0,$$

for all $t \in [0, 1]$ because $\|A - B\|_2 < \sigma_r(A)$. Hence $\sigma_r(X_t) > 0$, so $\mathrm{rank}(X_t) \geq r$. $\qquad\square$

**Lemma 5.** *Let $X$ be a random variable with a log-concave density, mean $\mu_X$, and variance $\sigma_X^2$. It holds that*

$$\mathbb{E}[X \mid X > \theta] \leq \theta + e\,\sigma_X \ , \quad \textit{for } \theta \geq \mu_X \ .$$

*Proof.* Let $m(x) = \mathbb{E}[X - x \mid X > x]$ be the mean residual life function. We want to bound $\mathbb{E}[X \mid X > \theta] = \theta + m(\theta)$ for $\theta \geq \mu_X$. Due to log-concavity of $X$, $m(x)$ is non-increasing (see, eg, Bagnoli and Bergstrom [2, Theorem 6]). Since $m(x)$ is non-increasing, $m(\theta) \leq m(\mu_X) = \mathbb{E}[X - \mu_X \mid X > \mu_X]$. We will now bound the conditional expectation for this case of $\theta = \mu_X$.

Let $Y = X - \mu_X$. Then $\mathbb{E}[Y] = 0$ and $\mathbb{V}(Y) = \sigma_X^2$. $m(\mu_X) = \mathbb{E}[Y \mid Y > 0] = \frac{\mathbb{E}[Y^+]}{\mathbb{P}(Y > 0)}$, where $Y^+ = \max(0, Y)$. We know $\mathbb{E}[Y^+] \leq \sqrt{\mathbb{E}[(Y^+)^2]} \leq \sqrt{\mathbb{E}[Y^2]} = \sigma_X$. As for the denominator, we know that for any random variable $X$ with a log-concave density and mean $\mu_X$, $\mathbb{P}(X \geq \mu_X) \geq 1/e$ (see, eg, Lovász and Vempala [21, Lemma 5.4]). Thus, $m(\mu_X) \leq \frac{\sigma_X}{1/e} = e\,\sigma_X$. $\qquad\square$

**Lemma 6.** *Let $x, y \in \mathbb{R}^d$ and $\widetilde{x}, \widetilde{y} \in \mathbb{R}^{\widetilde{d}}$ be random vectors. Assume that the pair $(x, \widetilde{x})$ is independent of the pair $(y, \widetilde{y})$. Let $\mathbf{A}$ be a fixed $d \times \widetilde{d}$ matrix and let $\theta \in \mathbb{R}$ be a scalar threshold. Define the events $C_x = \{x^\top \mathbf{A}\widetilde{x} > \theta\}$ and $C_y = \{y^\top \mathbf{A}\widetilde{y} > \theta\}$. Assume that these events have non-zero probability, i.e., $\mathbb{P}(C_x) > 0$ and $\mathbb{P}(C_y) > 0$. Then the conditional expectation of the outer product $x\widetilde{y}^\top$ given both events $C_x$ and $C_y$ factorizes as follows:*

$$\mathbb{E}\left[ x\widetilde{y}^\top \mid x^\top \mathbf{A}\widetilde{x} > \theta, \ y^\top \mathbf{A}\widetilde{y} > \theta \right] = \mathbb{E}\left[ x \mid x^\top \mathbf{A}\widetilde{x} > \theta \right] \cdot \left( \mathbb{E}\left[ \widetilde{y} \mid y^\top \mathbf{A}\widetilde{y} > \theta \right] \right)^\top .$$

*Proof.* The definition of conditional expectation given multiple events is conditioning on their intersection. Here $\mathbb{I}$ denotes the indicator function.

$$\mathbb{E}\left[ x\widetilde{y}^\top \mid C_x, C_y \right] = \mathbb{E}\left[ x\widetilde{y}^\top \mid C_x \cap C_y \right] = \frac{\mathbb{E}\left[ x\widetilde{y}^\top \mathbb{I}_{C_x \cap C_y} \right]}{\mathbb{P}(C_x \cap C_y)} \ .$$

The event $C_x$ is determined solely by the random variables $x$ and $\widetilde{x}$. The event $C_y$ is determined solely by the random variables $y$ and $\widetilde{y}$. By the initial assumption, the pair $(x, \widetilde{x})$ is independent of

the pair $(y, \widetilde{y})$. Therefore, the event $C_x$ is independent of the event $C_y$. This implies $\mathbb{P}(C_x \cap C_y) = \mathbb{P}(C_x) \mathbb{P}(C_y)$. Hence the denominator factorizes (and is non-zero since $\mathbb{P}(C_x) > 0$ and $\mathbb{P}(C_y) > 0$).

Now consider the numerator. Since $C_x$ and $C_y$ are independent, $\mathbb{I}_{C_x \cap C_y} = \mathbb{I}_{C_x} \mathbb{I}_{C_y}$, which implies

$$\mathbb{E}\left[x\widetilde{y}^\top \, \mathbb{I}_{C_x \cap C_y}\right] = \mathbb{E}\left[x\widetilde{y}^\top \, \mathbb{I}_{C_x} \mathbb{I}_{C_y}\right] = \mathbb{E}\left[x \, \mathbb{I}_{C_x}\right] \cdot \mathbb{E}\left[\widetilde{y} \, \mathbb{I}_{C_y}\right]^\top ,$$

again, due to independence of the pairs. Hence the numerator also factorizes. $\qquad\square$

**Lemma 7.** *Let $x \in \mathbb{R}^d$ and $\widetilde{x} \in \mathbb{R}^{\widetilde{d}}$ be random vectors such that their joint distribution is a multivariate normal distribution with zero mean. Let $\mathbf{A}$ be a fixed $d \times \widetilde{d}$ matrix, and consider the conditioning event $\mathcal{R} = \{(x, \widetilde{x}) \mid x^\top \mathbf{A} \widetilde{x} > \theta\}$ for some threshold $\theta \in \mathbb{R}$. Assume that the probability of this event is non-zero, i.e., $\mathbb{P}(\mathcal{R}) > 0$. Then*

$$\mathbb{E}\left[x \mid x^\top \mathbf{A}\widetilde{x} > \theta\right] = \mathbf{0}_d .$$

*Proof.* Let $Z = (x, \widetilde{x}) \in \mathbb{R}^{d+\widetilde{d}}$. The joint probability density function of $Z$, denoted by $p(Z)$, corresponds to the $\mathcal{N}(0, \Sigma_{joint})$ distribution for some covariance matrix $\Sigma_{joint}$. The conditional expectation is defined as:

$$\mathbb{E}\left[x \mid x^\top \mathbf{A} \widetilde{x} > \theta\right] = \mathbb{E}\left[x \mid Z \in \mathcal{R}\right] = \frac{\int_{\mathcal{R}} x\, p(Z)\, dZ}{\int_{\mathcal{R}} p(Z)\, dZ} = \frac{\int_{\mathcal{R}} x\, p(Z)\, dZ}{P(\mathcal{R})}$$

We focus on the numerator integral and show that it is zero owing to symmetry. First note that $p(Z)$ is symmetric around the origin. That is, $p(Z) = p(-Z)$ for all $Z \in \mathbb{R}^{d+\widetilde{d}}$. Second, observe that under the transformation $Z \mapsto -Z$, the condition becomes $(-u)^\top \mathbf{A}(-\widetilde{u}) > \theta$, which simplifies to $u^\top \mathbf{A}\widetilde{u} > \theta$. Thus, the region $\mathcal{R}$ is symmetric with respect to the origin: $Z \in \mathcal{R} \iff -Z \in \mathcal{R}$. $\quad\square$

**Lemma 8.** *Consider the random variable $z := uv$, where $u, v$ are jointly Gaussian as*

$$\begin{pmatrix} u \\ v \end{pmatrix} \sim \mathcal{N}\left(0, \begin{pmatrix} \sigma_u^2 & \gamma \\ \gamma & \sigma_v^2 \end{pmatrix}\right), \quad \text{with } 0 < \sigma_u^2, \sigma_v^2, \text{ and } 0 \leq \gamma < \sigma_u \sigma_v .$$

*Let $\{z_k\}_{k=1}^r$ be $r$ independent copies. The conditional expectation is upper and lower bounded as*

$$\mathbb{E}\left[z_i \,\Big|\, \sum_{k=1}^r z_k > \theta\right] \geq \max\left\{\gamma, \frac{\theta}{r}\right\} \text{ for all } \theta \in \mathbb{R} ,$$

$$\mathbb{E}\left[z_i \,\Big|\, \sum_{k=1}^r z_k > \theta\right] \leq \max\left\{\gamma, \frac{\theta}{r}\right\} + e\sqrt{\frac{\sigma_u^2 \sigma_v^2 + \gamma^2}{r}} \text{ for } \theta \geq 0 .$$

*For the specific case of $\gamma = 0$ (i.e. $u, v$ independent) and $\theta = 0$, a stronger lower bound is*

$$\mathbb{E}\left[z_i \,\Big|\, \sum_{k=1}^r z_k > 0\right] \geq \frac{2}{\pi r}\, \sigma_u\, \sigma_v .$$

*Proof.* **Simplify the expression.** Observe that $z_k$ are i.i.d. random variables. The expectation is $\mathbb{E}[z_k] = \mathbb{E}[uv] = \gamma$ (since $\mathbb{E}[u] = 0 = \mathbb{E}[v]$). Let $S = \sum_{k=1}^r z_k$, and let $p_S(.)$ denote the PDF of $S$. The expectation is $\mathbb{E}[S] = r\gamma$, and the variance is $\mathbb{V}[S] = r\left(\sigma_u^2 \sigma_v^2 + \gamma^2\right)$.

Due to the symmetry among the i.i.d. variables $z_k$, the conditional expectation $\mathbb{E}[z_i \mid S > \theta]$ is the same for all $i \in \{1, \ldots, r\}$. Let $Q(\theta) = \mathbb{E}[z_i \mid S > \theta]$. By linearity of expectation, we have

$$\mathbb{E}[S \mid S > \theta] = \mathbb{E}\left[\sum_{k=1}^r z_k \,\Big|\, S > \theta\right] = \sum_{k=1}^r \mathbb{E}[z_k \mid S > \theta] = r\, Q(\theta) .$$

$$\implies Q(\theta) = \frac{1}{r}\, \mathbb{E}[S \mid S > \theta] . \tag{12}$$

**Proof of lower bounds: general case lower bound $\frac{\theta}{r}$.** Observe that

$$\mathbb{E}[S \mid S > \theta] = \frac{\int_\theta^\infty s\, p_S(s) ds}{\int_\theta^\infty p_S(s) ds} \tag{13}$$

$$\geq \frac{\int_\theta^\infty \theta\, p_S(s) ds}{\int_\theta^\infty p_S(s) ds} = \theta \ .$$

Combining this with Eq. (12) shows the $\theta/r$ lower bound.

**Proof of lower bounds: general case lower bound $\gamma$.** For this, we show $\mathbb{E}[S \mid S > \theta]$ is non-decreasing in $\theta$. Let $h(\theta) = \mathbb{E}[S \mid S > \theta]$. Using Eq. (13), its derivative is given by

$$h'(\theta) = \frac{-\theta\, p_S(\theta) \int_\theta^\infty p_S(s) ds + p_S(\theta) \int_\theta^\infty s\, p_S(s) ds}{\mathbb{P}(S > \theta)^2}$$

$$= \frac{p_S(\theta)}{\mathbb{P}(S > \theta)^2} \int_\theta^\infty \underbrace{(s - \theta)}_{\geq 0}\, p_S(s) ds \geq 0 \ . \tag{14}$$

Thus $\mathbb{E}[S \mid S > \theta]$ is non-decreasing in $\theta$. In particular, $\mathbb{E}[S \mid S > \theta] \geq \mathbb{E}[S]$ (i.e. the unconditional limit in the limit $\theta \to -\infty$). Since $\mathbb{E}[S] = r\gamma$, using this in Eq. (12) shows the lower bound of $\gamma$.

**Proof of lower bounds: the specific case of $\gamma = 0$ and $\theta = 0$.** Since the distribution of $z_k$ is symmetric around zero, the distribution of $S = \sum_k z_k$ is also symmetric around zero. Therefore, $\mathbb{P}(S > 0) = 1/2$. Using this, we get

$$\mathbb{E}[S \mid S > 0] = \frac{\int_0^\infty s\, p_S(s) ds}{\mathbb{P}(S > 0)} = 2 \int_0^\infty s\, p_S(s) ds \ . \tag{15}$$

Also, the expectation of the absolute value is $\mathbb{E}[|S|] = \int_{-\infty}^\infty |s|\, p_S(s) ds$. Due to symmetry (i.e. $p_S(-s) = p_S(s)$), we get

$$\mathbb{E}[|S|] = \int_{-\infty}^0 (-s)\, p_S(s) ds + \int_0^\infty s\, p_S(s) ds = 2 \int_0^\infty s\, p_S(s) ds \ . \tag{16}$$

Using Eq. (15) and Eq. (16), we get

$$\mathbb{E}[S \mid S > 0] = \mathbb{E}[|S|] = \mathbb{E}\left[\left|\sum_{k=1}^r z_k\right|\right]$$

$$\geq^{(\dagger)} \mathbb{E}[|z_1|]$$

$$= \mathbb{E}[|u_1 v_1|] = \mathbb{E}[|u_1|\, |v_1|] = \mathbb{E}[|u_1|]\, \mathbb{E}[|v_1|] \qquad \text{(using independence)}$$

$$= \sigma_u \sigma_v\, \mathbb{E}[|a|]^2 = \frac{2}{\pi} \sigma_u \sigma_v \ . \qquad \text{(for } a \sim \mathcal{N}(0,1)\text{)}$$

Eq (†) holds intuitively. To formally show it, we invoke Lemma 3 (Whittle's inequality) on the convex function $\phi(x) = |x|$. Using this with Eq. (12) gives the desired result.

**Proof of the upper bound.** The probability density function of $z = uv$ is given by

$$f_z(x) = \frac{1}{\pi \sigma_u \sigma_v \sqrt{1 - \rho^2}} \exp\left(\frac{\rho x}{\sigma_u \sigma_v (1 - \rho^2)}\right) K_0\left(\frac{|x|}{\sigma_u \sigma_v (1 - \rho^2)}\right) \ ,$$

where $\rho = \gamma/(\sigma_u \sigma_v)$ denotes the correlation factor. Note that $|\rho| < 1$ is ensured via $\gamma < \sigma_u \sigma_v$ in the lemma statement. The function $K_0(a|x|)$ is log-concave for $a > 0$. The term $\exp(bx)$ is log-linear (hence log-concave). The product of log-concave functions is log-concave. Thus, $f_z(x)$ is log-concave. Since $S$ is a sum of $r$ i.i.d. random variables with log-concave densities, $S$ also has a log-concave density. We use Lemma 5 to get that $\mathbb{E}[S \mid S > \theta] \leq \theta + e\,\sqrt{r\,(\sigma_u^2 \sigma_v^2 + \gamma^2)}$ for $\theta \geq r\gamma$. For $\theta \in [0, r\gamma]$, we use the non-decreasing property of $\mathbb{E}[S \mid S > \theta]$ from Eq. (14). Plugging into Eq. (12) concludes the argument. □

**Lemma 9.** *Consider Gaussian random variables $x, y \in \mathbb{R}^r$, such that*

$$\begin{pmatrix} x \\ y \end{pmatrix} \sim \mathcal{N}\left(0, \begin{pmatrix} a_x \mathbf{I}_r & a_{xy}\,\mathbf{I}_r \\ a_{xy}\,\mathbf{I}_r & a_y \mathbf{I}_r \end{pmatrix}\right), \quad \text{with } a_x, a_y > 0,\ a_{xy} \geq 0\ .$$

*For $\theta \in \mathbb{R}$, define $\mathbf{A}(\theta) := \mathbb{E}\left[xy^\top \mid x^\top y > \theta\right]$. It holds that $\mathbf{A}(\theta)$ satisfies*

$$\mathbf{A}(\theta) = f(\theta)\,\mathbf{I}_r\ ,$$

*where $f(\theta)$ is a scalar function of $\theta \in \mathbb{R}$, such that*

$$\max\left\{a_{xy}, \frac{\theta}{r}\right\} + e\,\sqrt{\frac{a_x a_y + a_{xy}^2}{r}} \geq f(\theta) \geq \max\left\{a_{xy}, \frac{\theta}{r}\right\}\ .$$

*In the special case of $a_{xy} = 0$, it further holds that $f(0) \geq {}^{2\sqrt{a_x a_y}}/\pi r$.*

*Proof.* We first build an intuition for the quantity $\mathbf{A}(\theta) \in \mathbb{R}^{r \times r}$. For $\theta = -\infty$, $\mathbf{A}(\theta)$ becomes the unconditional expectation, which is $a_{xy}\,\mathbf{I}_r$ according to the given covariance structure. As $\theta$ increases in $\mathbb{R}$, we expect $\mathbf{A}(\theta)$ to increase.

$\mathbf{A}(\theta)$ **is diagonal.** We first show that $\mathbf{A}(\theta)$ is a diagonal matrix. The $(i,j)$-th entry is $\mathbf{A}(\theta)_{ij} = \mathbb{E}[x_i y_j \mid Z > \theta]$, where $Z = x^\top y = \sum_{l=1}^r x_l y_l$. Consider the transformation $T_i : \mathbb{R}^{2r} \to \mathbb{R}^{2r}$ that maps $(x, y)$ to $(x', y')$ where $x_l' = x_l$ for $l \neq i$, $x_i' = -x_i$, and $y_l' = y_l$ for $l \neq i$, $y_i' = -y_i$.

First, note that $Z' = \sum_{l \neq i} x_l y_l + (-x_i)(-y_i) = Z$. Hence the condition $Z > \theta$ is invariant under the transformation $T_i$. Second, due to independence and the block diagonal structure of the covariance, the overall joint density is a product of univariate Gaussians centered around zero. Due to the symmetry of a univariate Gaussian, the overall density is also invariant under $T_i$. Third, the entry $x_i y_j$ becomes $-x_i y_j$ under the transformation $T_i$. Due to this symmetry, we conclude that the off-diagonal entries are zero.

**All the diagonal entries of $\mathbf{A}(\theta)$ are equal by symmetry.** The diagonal entries are $\mathbf{A}(\theta)_{ii} = \mathbb{E}[x_i y_i \mid Z > \theta]$. Let $Z_i = x_i y_i$, meaning $Z = \sum_{l=1}^r Z_l$. Due to the block diagonal structure on $(x, y)$, each $Z_i$ is independent and identically distributed. Hence, $\mathbf{A}(\theta)_{ii} = \mathbf{A}(\theta)_{jj}$ for any $i, j \in [r]$.

**Properties of $f(\theta)$.** From the above two steps, we conclude that $\mathbf{A}(\theta) = f(\theta)\,\mathbf{I}_r$ for some scalar function $f : \mathbb{R} \to \mathbb{R}$. Using the trace trick, we see that

$$f(\theta) \cdot \mathrm{Tr}(\mathbf{I}_r) = \mathrm{Tr}\left(\mathbb{E}[xy^\top \mid x^\top y > \theta]\right)$$
$$\implies f(\theta) = \frac{1}{r}\,\mathbb{E}[x^\top y \mid x^\top y > \theta]\ .$$

Since the covariances of $x, y$ are scaled identity, each $x_i y_i, i \in [r]$ is identically distributed. This distribution is akin to $uv$ for $u \sim \mathcal{N}(0, a_x), v \sim \mathcal{N}(0, a_y)$ with $Cov(u, v) = a_{xy}$. Hence

$$f(\theta) = \mathbb{E}\left[u_1 v_1 \,\Bigg|\, \sum_{i=1}^r u_i v_i > \theta\right]\ ,$$

for $u_i, v_i$ i.i.d. according to the described distribution. Lemma 8 shows the required properties on this conditional expectation, showing the desired inequalities in the statement of this lemma. $\square$

**Lemma 10.** *Let $x \in \mathbb{R}^d$ and $\widetilde{x} \in \mathbb{R}^{\widetilde{d}}$ be jointly Gaussian vectors with mean zero and joint covariance matrix $\Sigma_{full}$ which is positive definite. Consider $\mathbf{M}_O, \mathbf{M}_T \in \mathbb{R}^{d \times \widetilde{d}}$ satisfying $\mathrm{rank}(\mathbf{M}_O) \geq 2$ and $\|\mathbf{M}_T - \mathbf{M}_O\| < \sigma_{\mathrm{rank}(\mathbf{M}_O)}(\mathbf{M}_O)$. For any $\mathbf{A} \in \mathbb{R}^{d \times \widetilde{d}}$, let $Y_{\mathbf{A}} := x^\top \mathbf{A}\,\widetilde{x}$. For a real $\theta \geq 0$, define:*

$$\Delta P(\theta) := \left|\mathbb{P}\{Y_{\mathbf{M}_T} > \theta\} - \mathbb{P}\{Y_{\mathbf{M}_o} > \theta\}\right|\ , \tag{17}$$
$$\Delta \mathbf{E}(\theta) := \left\|\mathbb{E}[x\widetilde{x}^\top \mathbb{I}(Y_{\mathbf{M}_T} > \theta)] - \mathbb{E}[x\widetilde{x}^\top \mathbb{I}(Y_{\mathbf{M}_o} > \theta)]\right\|_2\ , \tag{18}$$

*where the randomness is over the Gaussian $(x, \widetilde{x})$. Then, there exist constants $C_P(\theta, \Sigma_{full}, \mathbf{M}_O) > 0$ and $C_{\mathbf{E}}(\theta, \Sigma_{full}, \mathbf{M}_O) > 0$ that depend on $\theta$, the covariance $\Sigma_{full}$, and $\mathbf{M}_O$, such that:*

$$\Delta P(\theta) \leq C_P(\theta, \Sigma_{full}, \mathbf{M}_O)\,\|\mathbf{M}_T - \mathbf{M}_O\|_2\ , \tag{19}$$
$$\Delta \mathbf{E}(\theta) \leq C_{\mathbf{E}}(\theta, \Sigma_{full}, \mathbf{M}_O)\,\|\mathbf{M}_T - \mathbf{M}_O\|_2\ . \tag{20}$$

*Proof.* We prove the two bounds using differentiability arguments. Define $\Delta \mathbf{M} := \mathbf{M_T} - \mathbf{M_O}$, and define the scalar $Y_t := x^\top (\mathbf{M_O} + t\Delta\mathbf{M}) \, \widetilde{x}$. Note that we have overloaded notation by reusing $Y$; it shall be clear from the context that $Y_t$ for a scalar $t$ and $Y_\mathbf{A}$ for a matrix $\mathbf{A}$ mean different things.

Using Lemma 4 with the given condition on $\|\mathbf{M_T} - \mathbf{M_O}\|$, we conclude that $\text{rank}(\mathbf{M_O} + t\Delta\mathbf{M}) \geq \text{rank}(\mathbf{M_O}) \geq 2$ for all $t \in [0, 1]$. Since $(x, \widetilde{x})$ is jointly Gaussian, $\text{rank} \geq 2$ ensures that $Y_t$ for all $t \in [0, 1]$ have a smooth and bounded density everywhere. This is because the random variable $Y_\mathbf{A}$ is equivalent to the quadratic form on a Gaussian, $(1/2)z^\top \mathbf{H}\, z$ with

$$z := \begin{pmatrix} x \\ \widetilde{x} \end{pmatrix} \sim \mathcal{N}(0, \Sigma_{\text{full}}) \,, \quad \mathbf{H} = \begin{pmatrix} 0 & \mathbf{A} \\ \mathbf{A}^\top & 0 \end{pmatrix} \,.$$

This quadratic form has a known characteristic function as below (Mathai and Provost [23, Sec 3.2])

$$\phi(t) \propto \frac{1}{\sqrt{\det \left( I - 2it\,\Sigma_{full}\,\mathbf{H} \right)}} \,.$$

One can see that $\text{rank}(\mathbf{H}) = 2 \cdot \text{rank}(\mathbf{A})$ and $|\phi(t)|$ decays as $|t|^{-\text{rank}(\mathbf{H})/2}$ as $|t| \to \infty$. This shows that $\text{rank}(\mathbf{H}) \geq 4$ ensures at least a $|t|^{-2}$ decay, which ensures boundedness everywhere.

**(i) Probability Difference Bound (eq. (19)).** Define the path $h(t) := \mathbb{P}\{Y_t > \theta\}$ for $t \in [0, 1]$. Then by the Mean Value Theorem, it holds that

$$\Delta P(\theta) = |h(1) - h(0)| = |h'(\xi)| \quad \text{for some } \xi \in (0, 1).$$

Since $Y_t$ has a finite and bounded density everywhere, $h(t)$ is differentiable and its derivative is

$$h'(t) = \frac{d}{dt}\mathbb{P}\{Y_t > \theta\} = \mathbb{E}\left[\delta(Y_t - \theta) \cdot Y_t'\right] = \mathbb{E}\left[\delta(Y_t - \theta) \cdot x^\top \Delta\mathbf{M}\, \widetilde{x}\right] \,,$$

where $\delta$ is the Dirac delta function. Using the Cauchy–Schwarz inequality, we can write

$$\begin{aligned}
|h'(t)| &\leq \mathbb{E}\left[\delta(Y_t - \theta) \cdot |x^\top \Delta\mathbf{M}\, \widetilde{x}|\right] \\
&\leq \|\Delta\mathbf{M}\| \cdot \mathbb{E}\left[\delta(Y_t - \theta) \cdot \|x\| \cdot \|\widetilde{x}\|\right] \\
&= \|\Delta\mathbf{M}\| \cdot f_{Y_t}(\theta) \cdot \mathbb{E}\left[\|x\|\|\widetilde{x}\| \mid Y_t = \theta\right],
\end{aligned}$$

where $f_{Y_t}(\theta)$ is the density of $Y_t$ at $\theta$. Because $Y_t$ is non-degenerate for $t \in [0, 1]$, both $f_{Y_t}(\theta)$ and the conditional expectation are finite and bounded over $t$. Thus the linear dependence on $\|\Delta\mathbf{M}\|$ in Eq. (19) follows, since any $\xi \in (0, 1)$ satisfies the above conditions.

**(ii) Expectation Difference Bound (eq. (20)).** Define $H(t) := \mathbb{E}[x\widetilde{x}^\top \cdot \mathbb{I}\{Y_t > \theta\}]$. Then by the Mean Value Theorem, we have

$$\Delta\mathbf{E}(\theta) = \|H(1) - H(0)\| = \|H'(\xi)\| \quad \text{for some } \xi \in (0, 1).$$

Differentiating under the expectation gives

$$H'(t) = \mathbb{E}\left[x\widetilde{x}^\top \cdot \delta(Y_t - \theta) \cdot x^\top \Delta\mathbf{M}\, \widetilde{x}\right].$$

For any matrix norm, we have

$$\begin{aligned}
\|H'(t)\| &\leq \mathbb{E}[\|x\| \cdot \|\widetilde{x}\| \cdot |x^\top \Delta\mathbf{M}\widetilde{x}| \cdot \delta(Y_t - \theta)] \\
&\leq \|\Delta\mathbf{M}\| \cdot \mathbb{E}[\|x\|^2 \cdot \|\widetilde{x}\|^2 \cdot \delta(Y_t - \theta)] \\
&= \|\Delta\mathbf{M}\| \cdot f_{Y_t}(\theta) \cdot \mathbb{E}[\|x\|^2\|\widetilde{x}\|^2 \mid Y_t = \theta].
\end{aligned}$$

Again, all terms other than $\|\Delta\mathbf{M}\|$ are bounded for $t \in [0, 1]$, yielding the desired Eq. (20). $\qquad\square$

**Lemma 11.** *Let $x_1, \ldots, x_n \in \mathbb{R}^d$ be $n$ i.i.d. random vectors drawn from a Gaussian distribution $\mathcal{N}(0, \Sigma)$, where $\Sigma$ is a $d \times d$ positive definite covariance matrix, $d \geq 1, n \geq 1$. Let $S$ be a random subset of indices $\{1, \ldots, n\}$ generated by including each index $j \in \{1, \ldots, n\}$ independently with probability $p \in (0, 1]$. Let $n_c = |S|$ denote the number of selected samples, and define the sample covariance matrix for $n_c > 0$ as $\hat{\Sigma}_{n_c} = (1/n_c) \sum_{i \in S} x_i x_i^\top$. For a failure probability $\delta \in (0, 1)$, assume that $np > 8\log(2/\delta)$ holds. Then, with probability at least $1 - \delta$, both $n_c \geq np/2$ and the sample covariance matrix of the selected data satisfies:*

$$\left\|\hat{\Sigma}_{n_c} - \Sigma\right\|_2 \lesssim \|\Sigma\|_2 \sqrt{\frac{d + \log\frac{1}{\delta}}{np}} \,.$$

*Proof.* Define $k_{\min} := \left\lceil np - \sqrt{2np\log(2/\delta)} \right\rceil$. Note that $k_{\min} \geq np/2$ due to the assumption. Let

$$\mathcal{F}_1 := \{n_c < k_{\min}\}, \quad \mathcal{F}_2 := \left\{ n_c \geq k_{\min} \text{ and } \left\| \hat{\Sigma}_{n_c} - \Sigma \right\|_2 > \|\Sigma\|_2 \sqrt{\frac{d + \log\frac{1}{\delta}}{k_{\min}}} \right\}.$$

denote the failure events. A union bound over the two failure probabilities will give the desired result. Below we bound the individual failure probabilities.

Bounding $\mathbb{P}(\mathcal{F}_1)$: Define $\Delta_0 := \sqrt{2\log(2/\delta)/(np)}$, so that $k_{\min} = \lceil(1 - \Delta_0)np\rceil$. Since we assumed $np > 8\log(2/\delta)$, $\Delta_0 < 0.5$. By a standard Chernoff bound for binomial distributions, $\mathbb{P}(n_c < (1 - \Delta_0)np) \leq \exp(-np\Delta_0^2/2) = \exp(-\log(2/\delta)) = \delta/2$. Since $k_{\min} \geq (1 - \Delta_0)np$ (due to the ceil operation), it follows that $\mathbb{P}(\mathcal{F}_1) = \mathbb{P}(n_c < k_{\min}) \leq \mathbb{P}(n_c \leq (1 - \Delta_0)np) \leq \delta/2$.

Bounding $\mathbb{P}(\mathcal{F}_2)$: Using the law of total probability, we write

$$\mathbb{P}(\mathcal{F}_2) = \sum_{k=k_{\min}}^{n} \mathbb{P}\left( \left\| \frac{1}{k} \sum_{i \in S, |S|=k} x_i x_i^\top - \Sigma \right\|_2 > \|\Sigma\|_2 \sqrt{\frac{d + \log\frac{1}{\delta}}{k_{\min}}} \; \middle| \; n_c = k \right) \mathbb{P}(n_c = k)$$

For any $k \geq k_{\min}$, we have $1/\sqrt{k} \leq 1/\sqrt{k_{\min}}$. Thus, for $k \geq k_{\min}$:

$$\mathbb{P}\left( \left\| \frac{1}{k} \sum x_i x_i^\top - \Sigma \right\|_2 > \|\Sigma\|_2 \sqrt{\frac{d + \log\frac{1}{\delta}}{k_{\min}}} \; \middle| \; n_c = k \right) \leq$$

$$\mathbb{P}\left( \left\| \frac{1}{k} \sum x_i x_i^\top - \Sigma \right\|_2 > \|\Sigma\|_2 \sqrt{\frac{d + \log\frac{1}{\delta}}{k}} \; \middle| \; n_c = k \right).$$

And the right hand side is bounded by $\delta/2$ owing to standard matrix concentration results. So, $\mathbb{P}(\mathcal{F}_2) \leq \sum_{k=k_{\min}}^{n} (\delta/2)\mathbb{P}(n_c = k) \leq \delta/2$. $\qquad\square$

# D  A proof of Corollary 1

We present a proof of Corollary 1, which follows the proof presented in Nakada et al. [24] while fixing some typos. Before diving into the proof, we make some remarks.

**First**, the result stated in Corollary 1 is tighter than its counterpart Nakada et al. [24, Theorem 3.1] by a dimension factor. This is because we use tighter concentration, as detailed in the explanation between Eqs (28) and (29). **Second**, as remarked in Remark 4.1, Corollary 1 is not tight in the SNR parameters $\gamma, \widetilde{\gamma}$. **Third**, the result in Nakada et al. [24] is for a general covariance on the signal, $\Sigma_z$, and the noise, $\Sigma_\xi$, whereas our setting is more restricted from Assumptions 1 and 2. This restriction is required for the analysis of filtering in Theorem 1.

**Fourth**, the result in [24] is stated with probability $1 - O(1/n)$, whereas we state it with probability $1 - \exp(-d)$. Due to this, Corollary 1 as stated does not have a $\log n$ factor inside the square root, unlike Nakada et al. [24, Theorem 3.1]. **Fifth**, there is a small subtle difference in the setting of [24] and ours. We use $\eta$ to denote the fixed probability of clean samples in Assumption 1, whereas Nakada et al. [24] use $\eta$ to denote the fraction of clean samples in the *sampled* dataset, which is a random quantity. Using $n_c$ to denote the number of clean samples, we go through the additional step of controlling the error in $|n_c/n - \eta|$, which scales as $1/\sqrt{n}$, since this source of error is 1-dimensional. **Sixth**, the result in Nakada et al. [24, Theorem 3.1] is stated as $\min\{\sqrt{r}, .\}$. While it is true that the $\sin\Theta$ metric can be at most $\sqrt{r}$, the final step in the proof is the application Lemma 2, which requires a condition that translates to $n \gtrsim (1/\eta^2) \max\{d, \widetilde{d}\} (1 + \gamma^{-1})(1 + \widetilde{\gamma}^{-1})$. And so this is how we state the result in Corollary 1, which makes the stated upper bound always smaller than $\sqrt{r}$.

For clarity, we write the algorithm:
*Input.* $\mathbf{X} \in \mathbb{R}^{n \times d}, \widetilde{\mathbf{X}} \in \mathbb{R}^{n \times \widetilde{d}}, r \in \mathbb{Z}_+, \rho \in (0, \infty)$.
*Output.* $\mathbf{G}^\top \widetilde{\mathbf{G}} \in \mathbb{R}^{d \times \widetilde{d}}$ (with rank $= r$, since $\mathbf{G} \in \mathbb{R}^{r \times d}, \widetilde{\mathbf{G}} \in \mathbb{R}^{r \times \widetilde{d}}$) by minimizing Eq. (3).

**Step 1: Reduction of loss.** We show that

$$\mathcal{L}_0(\mathbf{G}, \widetilde{\mathbf{G}}) = -\mathrm{Tr}\left(\mathbf{G}\mathbf{S}_n\widetilde{\mathbf{G}}^\top\right) \,, \tag{21}$$

where $\mathbf{S}_n$ denotes the cross covariance matrix of the data, given by (Eq. (5) rewritten)

$$\mathbf{S}_n = \frac{1}{n-1}\sum_{i=1}^{n}(x_i - \overline{x})\left(\widetilde{x}_i - \overline{\widetilde{x}}\right)^\top \in \mathbb{R}^{d\times\widetilde{d}} \,.$$

*Proof.* Expand the LHS as

$$\mathcal{L}_0(\mathbf{G}, \widetilde{\mathbf{G}}) = \frac{1}{2n(n-1)}\left(\sum_{i=1}^{n}\left(\sum_{\substack{j=1\\j\neq i}}^{n}(s_{ij} - s_{ii}) + \sum_{\substack{j=1\\j\neq i}}^{n}(s_{ji} - s_{ii})\right)\right)$$

$$\overset{(a)}{=} \frac{1}{n(n-1)}\left(\sum_{i=1}^{n}\left(\sum_{\substack{j=1\\j\neq i}}^{n}(s_{ij} - s_{ii})\right)\right)$$

$$= \frac{1}{n(n-1)}\left(\sum_{i}\sum_{j\neq i}s_{ij} - (n-1)\sum_{i}s_{ii}\right)$$

$$= \frac{1}{n(n-1)}\left(\sum_{i}\sum_{j\neq i}s_{ij}\right) - \frac{1}{n}\left(\sum_{i}s_{ii}\right) \,, \tag{X}$$

where eq $(a)$ holds because the overall sum over the $n\times n$ similarity matrix is the same whether done over rows or columns.

For the RHS, we first rewrite $\mathbf{S}_n$ as

$$\mathbf{S}_n = \frac{1}{n-1}\left(\sum_{i=1}^{n}x_i\widetilde{x}_i^\top - n\overline{x}\overline{\widetilde{x}}^\top\right)$$

$$= \frac{1}{n-1}\left(\sum_{i=1}^{n}x_i\widetilde{x}_i^\top\right) - \frac{1}{n(n-1)}\left(\sum_{i=1}^{n}x_i\right)\left(\sum_{i=1}^{n}\widetilde{x}_i\right)^\top$$

$$= \frac{1}{n-1}\left(\sum_{i}x_i\widetilde{x}_i^\top\right) - \frac{1}{n(n-1)}\left(\sum_{i}x_i\widetilde{x}_i^\top + \sum_{i}\sum_{j\neq i}x_i\widetilde{x}_j^\top\right)$$

$$= \frac{1}{n-1}\left(1 - \frac{1}{n}\right)\left(\sum_{i}x_i\widetilde{x}_i^\top\right) - \frac{1}{n(n-1)}\left(\sum_{i}\sum_{j\neq i}x_i\widetilde{x}_j^\top\right)$$

$$= \frac{1}{n}\left(\sum_{i}x_i\widetilde{x}_i^\top\right) - \frac{1}{n(n-1)}\left(\sum_{i}\sum_{j\neq i}x_i\widetilde{x}_j^\top\right) \,.$$

Using the above, we rewrite the RHS as

$$-\mathrm{Tr}\left(\mathbf{G}\mathbf{S}_n\widetilde{\mathbf{G}}^\top\right) = -\mathrm{Tr}\left(\frac{1}{n}\left(\sum_{i}\mathbf{G}x_i\widetilde{x}_i^\top\widetilde{\mathbf{G}}^\top\right) - \frac{1}{n(n-1)}\left(\sum_{i}\sum_{j\neq i}\mathbf{G}x_i\widetilde{x}_j^\top\widetilde{\mathbf{G}}^\top\right)\right)$$

$$= \frac{1}{n(n-1)}\left(\sum_{i}\sum_{j\neq i}\mathrm{Tr}\left(\mathbf{G}x_i\widetilde{x}_j^\top\widetilde{\mathbf{G}}^\top\right)\right) - \frac{1}{n}\left(\sum_{i}\mathrm{Tr}\left(\mathbf{G}x_i\widetilde{x}_i^\top\widetilde{\mathbf{G}}^\top\right)\right)$$

(Linearity of Trace)

$$= \frac{1}{n(n-1)} \left( \sum_i \sum_{j \neq i} \langle \mathbf{G} x_i, \widetilde{\mathbf{G}} \widetilde{x}_j \rangle \right) - \frac{1}{n} \left( \sum_i \langle \mathbf{G} x_i, \widetilde{\mathbf{G}} \widetilde{x}_i \rangle \right)$$

(Cyclic nature of Trace)

$$= \frac{1}{n(n-1)} \left( \sum_i \sum_{j \neq i} s_{ij} \right) - \frac{1}{n} \left( \sum_i s_{ii} \right) .$$

(Definition of $s_{ij}$)

Comparing the above to eq (X) concludes the proof. $\qquad\square$

**Step 2: Closed-form solution.** We show that (Eq. (6) rewritten)

$$\arg \min_{\mathbf{G}, \widetilde{\mathbf{G}}} \mathcal{L}_\rho \left( \mathbf{G}, \widetilde{\mathbf{G}} \right) = \left\{ \left( \mathbf{G}, \widetilde{\mathbf{G}} \right) \ \middle| \ \mathbf{G}^\top \widetilde{\mathbf{G}} = \frac{1}{\rho} \ \text{SVD}_r \left( \mathbf{S}_n \right) \right\} .$$

Hence, even though the optimization problem is non-convex, there is a closed-form solution, and no optimization analysis is needed. In particular, the right singular vectors of $\mathbf{G}, \widetilde{\mathbf{G}}$ are determined independent of the choice of $\rho$. This result is from Nakada et al. [24, Lemma 2.1].

*Proof.* Using Step 1's result, we can write

$$\min_{\mathbf{G}, \widetilde{\mathbf{G}}} \mathcal{L}_\rho(\mathbf{G}, \widetilde{\mathbf{G}}) \equiv \max_{\mathbf{G}, \widetilde{\mathbf{G}}} \ \text{Tr} \left( \mathbf{G} \mathbf{S}_n \widetilde{\mathbf{G}}^\top \right) - \frac{\rho}{2} \|\mathbf{G}^\top \widetilde{\mathbf{G}}\|_F^2 . \tag{22}$$

The objective can be rewritten as

$$\text{Tr} \left( \mathbf{G} \mathbf{S}_n \widetilde{\mathbf{G}}^\top \right) - \frac{\rho}{2} \|\mathbf{G}^\top \widetilde{\mathbf{G}}\|_F^2 = \frac{\rho}{2} \left( \left\| \frac{\mathbf{S}_n}{\rho} \right\|_F^2 - \left\| \mathbf{G}^\top \widetilde{\mathbf{G}} - \frac{\mathbf{S}_n}{\rho} \right\|_F^2 \right) .$$

The optimization variables appear only in the second term. Since $\text{rank} \left( \mathbf{G}^\top \widetilde{\mathbf{G}} \right) = r$, by the Eckart-Young-Minsky Theorem, the solution is given by the best rank $r$ approximation of $\mathbf{S}_n / \rho$. $\qquad\square$

**Step 3: Relating error to op-norm concentration of $\mathbf{S}_n$.** We show the below, where $\mathbf{S}_n$ concentrates to $\mathbf{S} = \eta \, \mathbf{U} \widetilde{\mathbf{U}}^\top$.

$$\|\text{SVD}_r \left( \mathbf{S}_n \right) - \mathbf{S}\| \leq 2 \|\mathbf{S}_n - \mathbf{S}\| . \tag{23}$$

*Proof.* By triangle inequality, we have

$$\|\text{SVD}_r \left( \mathbf{S}_n \right) - \mathbf{S}\| \leq \|\text{SVD}_r \left( \mathbf{S}_n \right) - \mathbf{S}_n\| + \|\mathbf{S}_n - \mathbf{S}\| .$$

And for the first term on the right hand side, we use

$$\begin{aligned}
\|\text{SVD}_r \left( \mathbf{S}_n \right) - \mathbf{S}_n\| &= \sigma_{r+1} \left( \mathbf{S}_n \right) \\
&\leq^{(\dagger)} \sigma_{r+1} \left( \mathbf{S} \right) + \|\mathbf{S}_n - \mathbf{S}\| \\
&\leq^{(\dagger\dagger)} \|\mathbf{S}_n - \mathbf{S}\| .
\end{aligned}$$

In Eq. ($\dagger$), we used Lemma 1, and Eq. ($\dagger\dagger$) holds because $\sigma_{r+1} \left( \mathbf{S} \right) = 0$, since $\mathbf{S}$ is rank $r$. $\qquad\square$

**Step 4: Concentration of $\mathbf{S}_n$.** We show that with probability $1 - \exp \left( -\Omega(\max\{d, \widetilde{d}\}) \right)$,

$$\|\mathbf{S}_n - \mathbf{S}\| \lesssim \sqrt{\frac{\max\{d, \widetilde{d}\} (1 + \gamma^{-1})(1 + \widetilde{\gamma}^{-1})}{n}} + \widetilde{O} \left( \frac{1}{n} \right) . \tag{24}$$

Before we prove this, we remark that the condition of $n \gtrsim n_0$ (for the appropriate $n_0$ stated in the statement of Corollary 1) ensures that the $\widetilde{O}(1/n)$ term is at $\widetilde{O}(\eta^2)$ whereas the first term is $\widetilde{O}(\eta)$. This ensures that we are in the regime where the $1/\sqrt{n}$ term dominates.

*Proof.* We start with the expansion of $\mathbf{S}_n$,

$$\mathbf{S}_n = \frac{1}{n-1} \sum_{i=1}^{n} x_i \widetilde{x}_i^\top - \frac{n}{n-1} \overline{x} \overline{\widetilde{x}}^\top = \underbrace{\frac{1}{n} \sum_{i=1}^{n} x_i \widetilde{x}_i^\top}_{\mathbf{S}_n^{(1)}} - \underbrace{\frac{1}{n(n-1)} \sum_{i=1}^{n} \sum_{\substack{j=1 \\ j \neq i}}^{n} x_i \widetilde{x}_j^\top}_{\mathbf{S}_n^{(2)}} .$$

The main term that dictates the convergence is $\mathbf{S}_n^{(1)}$. The term $\mathbf{S}_n^{(2)}$ concentrates around *zero* (since samples $i \neq j$, $i, j \in [n]$ are independent), and the rate of convergence is $\widetilde{O}(1/n)$ due to averaging over $n^2$ terms, which is a higher order term. Let $n_c$ be a random variable that denotes the number of clean data points. We expand the sum in $\mathbf{S}_n^{(1)}$ below.

$$n\,\mathbf{S}_n^{(1)} = \sum_{i=1}^{n} x_i \widetilde{x}_i^\top = \underbrace{\sum_{i=1}^{n_c} \mathbf{U} z_i \widetilde{z}_i^\top \widetilde{\mathbf{U}}^\top}_{J_1} + \underbrace{\sum_{i=n_c+1}^{n} \mathbf{U} z_i \widetilde{z}_i^\top \widetilde{\mathbf{U}}^\top}_{J_2}$$

$$+ \underbrace{\sum_{i=1}^{n} \mathbf{U} z_i \widetilde{\xi}_i^\top}_{K_1} + \underbrace{\sum_{i=1}^{n} \xi_i \widetilde{z}_i^\top \widetilde{\mathbf{U}}^\top}_{K_2} + \underbrace{\sum_{i=1}^{n} \xi_i \widetilde{\xi}_i^\top}_{K_3} .$$

We control the error in each term separately. For terms $J_2, K_{1:3}$, we need a result like Nakada et al. [24, Proposition C.1] in the simple case of $X \perp \widetilde{X}$. For term $J_1$, we need it for $X = \widetilde{X}$.

The following two facts are going to be used multiple times. Here $X, Y$ denote random quantities, and all others are fixed quantities (matrices/vectors).

$$\text{w.h.p. } \|X - A\| \leq E_A, \ \|Y - B\| \leq E_B \implies \text{w.h.p. } \|X + Y - (A + B)\| \leq E_A + E_B , \quad (25)$$
$$\text{w.h.p. } \|X - A\| \leq E_A \implies \text{w.h.p. } \|MXN - MAN\| \leq \|M\|\|N\|E_A . \quad (26)$$

For the independent terms ($J_2, K_{1:3}$), we will use the below generic result. For $\mathbb{R}^{d_x} \ni x \sim \mathcal{N}(0, \Sigma_x)$ and $\mathbb{R}^{d_y} \ni y \sim \mathcal{N}(0, \Sigma_y)$ and $N$ i.i.d. draws from both, we have the below result from the application of a Matrix-Bernstein result.

$$\text{w.p. } 1 - e^{-t}, \quad \left\| \frac{1}{N} \sum_{i=1}^{N} x_i y_i^\top \right\| \lesssim \sqrt{\frac{\|\Sigma_x\| \cdot \|\Sigma_y\|}{N} (t + \log(d_x + d_y))} . \quad (27)$$

For the dependent term ($J_1$), we will use the below. Let $\mathbb{R}^{d_x} \ni x \sim \mathcal{N}(0, \Sigma_x)$ and $N$ i.i.d. draws from this. This is also known in the literature, for e.g., Bunea and Xiao [4, Theorem 2.2].

$$\text{w.p. } 1 - e^{-t}, \quad \left\| \frac{1}{N} \sum_{i=1}^{N} x_i x_i^\top - \Sigma_x \right\| \lesssim \|\Sigma_x\| \sqrt{\frac{t + \log(d_x)}{N}} . \quad (28)$$

Note that the above two concentration results are tighter than Nakada et al. [24, Proposition C.1] by a factor of dimension, since the proposition has trace terms too, whereas only operator norms appear in the above two equations. This manifests in Corollary 1 as stated being tighter than Nakada et al. [24, Theorem 3.1] by a dimension factor inside the square root (since we avoided $\log n$ but did not incur an additional dimension due to the failure probability of $\exp(-d)$). Finally, since $n_c = Bin(n, \eta)$, the ratio $n_c/n$ concentrates to $\eta$, with the error described by Hoeffding's inequality as

$$\mathbb{P}\left( \left| \frac{n_c}{n} - \eta \right| \geq \epsilon \right) \leq 2 \exp\left(-2n\epsilon^2\right) . \quad (29)$$

Using these results, we bound the individual terms of deviation. We first bound the independent terms using Eq. (27) with $t := \max\{d, \widetilde{d}\}$. The choice of $N$ is given with each setting. With probability $1 - \exp(-\Omega(\max\{d, \widetilde{d}\}))$, the following hold:

$$\left\| \frac{K_1}{n} \right\| \lesssim \sqrt{\frac{\|\Sigma_z\| \cdot \|\Sigma_{\widetilde{\xi}}\| \cdot \max\{d, \widetilde{d}\}}{n}} = \sqrt{\frac{\max\{d, \widetilde{d}\} \widetilde{\gamma}^{-1}}{n}} , \quad (N := n)$$

$$\left\|\frac{K_2}{n}\right\| \lesssim \sqrt{\frac{\|\Sigma_z\| \cdot \|\Sigma_\xi\| \cdot \max\{d,\widetilde{d}\}}{n}} = \sqrt{\frac{\max\{d,\widetilde{d}\}\,\gamma^{-1}}{n}}\,, \qquad (N := n)$$

$$\left\|\frac{K_3}{n}\right\| \lesssim \sqrt{\frac{\|\Sigma_\xi\| \cdot \|\Sigma_{\widetilde{\xi}}\| \cdot \max\{d,\widetilde{d}\}}{n}} = \sqrt{\frac{\max\{d,\widetilde{d}\}\,\gamma^{-1}\,\widetilde{\gamma}^{-1}}{n}}\,, \qquad (N := n)$$

$$\left\|\frac{J_2}{n}\right\| \lesssim \sqrt{1 - \frac{n_c}{n}} \cdot \sqrt{\frac{\|\Sigma_z\|^2 \cdot \max\{d,\widetilde{d}\}}{n}} = \sqrt{1 - \frac{n_c}{n}} \cdot \sqrt{\frac{\max\{d,\widetilde{d}\}}{n}}\,. \qquad (N := n - n_c)$$

We now bound the dependent term using Eq. (28). We need some additional machinery to deal with the random denominator, which we capture in Lemma 11. The requirement of $np \gtrsim \log(1/\delta)$ in the lemma translates to $n \gtrsim \max\{d,\widetilde{d}\}/\eta$, since we have $p := \eta$ and $\delta := \exp(-\max\{d,\widetilde{d}\})$. As we will see later, step 5 of the proof requires $n \gtrsim \max\{d,\widetilde{d}\}/\eta^2$, hence this requirement is already satisfied. With probability $1 - \exp(-\Omega(\max\{d,\widetilde{d}\}))$, it holds:

$$\left\|\frac{J_1}{n_c} - \mathbf{U}\widetilde{\mathbf{U}}^\top\right\| \lesssim \left\|\mathbf{U}\widetilde{\mathbf{U}}^\top\right\| \cdot \sqrt{\frac{\max\{d,\widetilde{d}\}}{n\eta}} \tag{30}$$

$$\implies \left\|\frac{J_1}{n} - \frac{n_c}{n}\mathbf{U}\widetilde{\mathbf{U}}^\top\right\| \lesssim \frac{n_c}{n}\sqrt{\frac{1}{\eta}} \cdot \sqrt{\frac{\max\{d,\widetilde{d}\}}{n}}$$

$$\implies \left\|\frac{J_1}{n} - \eta\,\mathbf{U}\widetilde{\mathbf{U}}^\top\right\| \lesssim \frac{n_c}{n}\sqrt{\frac{1}{\eta}} \cdot \sqrt{\frac{\max\{d,\widetilde{d}\}}{n}} + \left|\frac{n_c}{n} - \eta\right|\,.$$

For the concentration of $n_c/n$, we use Eq. (29) to get that with probability $1 - \exp(-\Omega(\max\{d,\widetilde{d}\}))$:

$$\left|\frac{n_c}{n} - \eta\right| \lesssim \sqrt{\frac{\max\{d,\widetilde{d}\}}{n}}\,. \tag{31}$$

We now add all the error bounds. For the combined error from terms $J_1$ and $J_2$, we note that $\sqrt{1 - n_c/n} \le 1$, and $(n_c/n\sqrt{\eta}) \le 2$ with high probability (since $n_c/n$ concentrates around $\eta$). The failure probability of this can be absorbed into the overall failure probability. Eq. (24) follows. $\qquad\square$

**Step 5: Relating singular vector recovery error to operator norm concentration.** We will apply Lemma 2 (a Davis-Kahan type result) to relate the $\sin\Theta$ metric to the operator norm. Combining Eqs. (24), (23) and (6), we get that with probability $1 - \exp(-\Omega(\max\{d,\widetilde{d}\}))$:

$$\left\|\mathbf{G}^\top\widetilde{\mathbf{G}} - \frac{\eta}{\rho}\mathbf{U}\widetilde{\mathbf{U}}^\top\right\| \lesssim \frac{1}{\rho}\left(\sqrt{\frac{\max\{d,\widetilde{d}\}\,(1+\gamma^{-1})(1+\widetilde{\gamma}^{-1})}{n}} + \widetilde{O}\left(\frac{1}{n}\right)\right)\,. \tag{32}$$

The instantiation for Lemma 2 is as follows: $A = \frac{\eta}{\rho}\mathbf{U}\widetilde{\mathbf{U}}^\top$, $\hat{A} = \mathbf{G}^\top\widetilde{\mathbf{G}}$. Note that both $A, \hat{A}$ are rank-$r$, and $\sigma_r(A) = \eta/\rho$. We get

$$\left\|\sin\Theta\left(\texttt{lsv}(\mathbf{G}^\top\widetilde{\mathbf{G}}),\mathbf{U}\right)\right\|_F \le \frac{\|\mathbf{G}^\top\widetilde{\mathbf{G}} - \frac{\eta}{\rho}\mathbf{U}\widetilde{\mathbf{U}}^\top\|_F}{\frac{\eta}{\rho} - \|\mathbf{G}^\top\widetilde{\mathbf{G}} - \frac{\eta}{\rho}\mathbf{U}\widetilde{\mathbf{U}}^\top\|_2}\,. \tag{33}$$

Now we will use three things. First, for the numerator, we use $\|M\|_F \le \sqrt{\text{rank}(M)} \cdot \|M\|_2$ for any matrix $M$. Second, for the denominator, we will need the additional condition of $n \gtrsim (1/\eta^2)\max\{d,\widetilde{d}\}(1+\gamma^{-1})(1+\widetilde{\gamma}^{-1})$ to ensure the second term is at most half of the first term. This also ensures that the $\widetilde{O}(1/n)$ does not dominate the $1/\sqrt{n}$ term. Third, triangle inequality with the fact that $\left\|\sin\Theta\left(\texttt{lsv}(\mathbf{G}^\top\widetilde{\mathbf{G}}),\texttt{rsv}(\mathbf{G})\right)\right\|_F = 0$ gives the final result. To see this fact, write

$$\mathbf{G}^\top\widetilde{\mathbf{G}} = V_{\mathbf{G}}\left(\Sigma_{\mathbf{G}}U_{\mathbf{G}}^\top U_{\widetilde{\mathbf{G}}}\Sigma_{\widetilde{\mathbf{G}}}\right)V_{\widetilde{\mathbf{G}}}^\top$$
$$= V_{\mathbf{G}}PSQ^\top V_{\widetilde{\mathbf{G}}}^\top\,. \qquad \text{(Using SVD of the middle component)}$$

Using the uniqueness of SVD, we get that $\mathtt{lsv}\left(\mathbf{G}^\top\widetilde{\mathbf{G}}\right) = V_\mathbf{G}P$ and $\mathtt{rsv}\left(\mathbf{G}^\top\widetilde{\mathbf{G}}\right) = V_{\widetilde{\mathbf{G}}}Q$. Since $P, Q$ are just orthogonal transforms, the subspace spanned by $V_\mathbf{G}$ and $V_\mathbf{G}P$ are the same, implying $\|\sin\Theta(V_\mathbf{G}, V_\mathbf{G}P)\|_F = 0$ (and analogously for $V_{\widetilde{\mathbf{G}}}$ and $V_{\widetilde{\mathbf{G}}}Q$).

Combining Eqs. (32) and (33) gives the desired result. Since the upper bound is valid for recovery of both $\mathbf{U}$ and $\widetilde{\mathbf{U}}$, Corollary 1 as stated follows.

## E   A proof of Proposition 1

Consider the following construction for the hard problem instance (lower bound): $(i)$ the latent dimension $r = 1$, and $(ii)$ the noise $\widetilde{\xi} = 0$ (i.e. $\widetilde{\gamma} = \infty$), but $\xi \neq 0$ (i.e. $\gamma$ is finite). This means the following proof recovers the $d\gamma^{-1}$ part from the $\max\{d\gamma^{-1}, \widetilde{d}\widetilde{\gamma}^{-1}\}$ term in Proposition 1. A similar argument can be made for the case when $\xi = 0, \widetilde{\xi} \neq 0$, leading to the $\max$ over both errors.

Owing to $r = 1$, this becomes a 1-dimensional vector recovery problem. Let $\mathbf{u}, \widetilde{\mathbf{u}} \in \mathbb{R}^d$ denote the vectors to recover. Upon seeing $\mathbf{S}_n$, there is no error in estimating $\widetilde{\mathbf{u}}$ since $\widetilde{\xi} = 0$, but there is error in estimating $\mathbf{u}$. To calculate this error, define $\mathbf{u}_n$ to be the top-left singular vector of $\mathbf{S}_n$. Note that $\mathbf{S}_n$ has only one non-zero singular value, since it fully lies on $\widetilde{\mathbf{u}}$ in the right singular vector space (i.e. $\mathbf{S}_n\mathbf{v} = 0$ for any $\mathbf{v} \perp \widetilde{\mathbf{u}}$). Hence

$$\mathbf{S}_n = \|\mathbf{S}_n\| \cdot \mathbf{u}_n \widetilde{\mathbf{u}}^\top . \tag{34}$$

**Step 0. Writing down $\mathbf{S}_n$.**

$$\mathbf{S}_n = \frac{1}{n-1}\sum_{i=1}^n x_i \widetilde{x}_i^\top - \frac{1}{n(n-1)}\left(\sum_{i=1}^n x_i\right)\left(\sum_{i=1}^n \widetilde{x}_i\right)^\top$$

$$= \underbrace{\frac{1}{n}\sum_{i=1}^n x_i \widetilde{x}_i^\top}_{\mathbf{S}_n^{(1)}} - \underbrace{\frac{1}{n(n-1)}\left(\sum_{i=1}^n \sum_{\substack{j=1 \\ j\neq i}}^n x_i \widetilde{x}_j^\top\right)}_{\mathbf{S}_n^{(2)}} .$$

We expand $\mathbf{S}_n^{(1)}$ below, using $n_c$ to denote the random variable denoting the clean samples. Note that $\mathbb{E}n_c = \eta n$. Similarly one can expand $\mathbf{S}_n^{(2)}$, however, the error of $\mathbf{S}_n^{(2)}$ will behave as $O(1/n)$ due to averaging over $n^2$ samples, which is a higher order term in the overall rate. That is, the behavior (in the large $n$ regime) will be largely dictated by $\mathbf{S}_n^{(1)}$.

$$\mathbf{S}_n^{(1)} = \frac{1}{n}\sum_{i=1}^n x_i \widetilde{x}_i^\top = \frac{1}{n}\sum_{i=1}^{n_c}(z_i\mathbf{u} + \xi_i)(z_i\widetilde{\mathbf{u}})^\top + \frac{1}{n}\sum_{i=n_c+1}^n (z_i\mathbf{u} + \xi_i)(\widetilde{z}_i\widetilde{\mathbf{u}})^\top .$$

As for the expectations, they are given by:

$$\mathbb{E}\left[\mathbf{S}_n^{(1)}\right] = \frac{1}{n}\mathbb{E}\left[\sum_{i=1}^{n_c} z_i^2\right]\mathbf{u}\widetilde{\mathbf{u}}^\top = \frac{1}{n}\mathbb{E}\left[n_c\right]\mathbf{u}\widetilde{\mathbf{u}}^\top = \eta\,\mathbf{u}\widetilde{\mathbf{u}}^\top ,$$

$$\mathbb{E}\left[\mathbf{S}_n^{(2)}\right] = 0 \ \ \text{(since all random quantities are zero-mean and independent)} .$$

**Step 1. Decompose $\sin\theta$ metric.** Our goal is a high probability lower bound on $|\sin\theta(\mathbf{u}_n, \mathbf{u})|$, where $\mathbf{u}_n$ is the random quantity. Note that

$$|\sin\theta(\mathbf{u}_n, \mathbf{u})| = \left\|\left(\mathbf{I}_d - \mathbf{u}\mathbf{u}^\top\right)\mathbf{u}_n\right\| . \tag{35}$$

To see this, note that $LHS = \sqrt{1 - (\mathbf{u}^\top\mathbf{u}_n)^2}$. Squaring both sides and expanding suffices.

**Step 2. Compute the metric for this case.** Using Eq. (34) in Eq. (35), we can write

$$|\sin\theta(\mathbf{u}_n, \mathbf{u})| = \frac{\left\|\left(\mathbf{I}_d - \mathbf{u}\mathbf{u}^\top\right)\mathbf{S}_n\widetilde{\mathbf{u}}\right\|}{\|\mathbf{S}_n\|} . \tag{36}$$

**Step 3. Computing the high probability bound.** We will give high probability lower bound on the numerator and denominator of Eq. (36) separately.

**Step 3.1. For the numerator:** We first expand $\mathbf{S}_n^{(1)}$ as

$$\mathbf{S}_n^{(1)}\widetilde{\mathbf{u}} = \frac{1}{n}\sum_{i=1}^{n_c}(z_i^2\mathbf{u} + z_i\xi_i) + \frac{1}{n}\sum_{i=n_c+1}^{n}(z_i\widetilde{z}_i\mathbf{u} + \widetilde{z}_i\xi_i)$$

$$\implies \left(\mathbf{I}_d - \mathbf{u}\mathbf{u}^\top\right)\mathbf{S}_n^{(1)}\widetilde{\mathbf{u}} = \left(\mathbf{I}_d - \mathbf{u}\mathbf{u}^\top\right)\left(\frac{1}{n}\sum_{i=1}^{n_c}z_i\xi_i + \frac{1}{n}\sum_{i=n_c+1}^{n}\widetilde{z}_i\xi_i\right)$$

$$\overset{d}{=} \left(\mathbf{I}_d - \mathbf{u}\mathbf{u}^\top\right)\left(\frac{1}{n}\sum_{i=1}^{n}z_i\xi_i\right).$$

Similarly, for $\mathbf{S}_n^{(2)}$ we have

$$\left(\mathbf{I}_d - \mathbf{u}\mathbf{u}^\top\right)\mathbf{S}_n^{(2)}\widetilde{\mathbf{u}} = \left(\mathbf{I}_d - \mathbf{u}\mathbf{u}^\top\right)\left(\frac{1}{n(n-1)}\sum_{i=1}^{n}\sum_{\substack{j=1\\j\neq i}}^{n}\widetilde{z}_j\xi_i\right)$$

$$\overset{d}{=} \left(\mathbf{I}_d - \mathbf{u}\mathbf{u}^\top\right)\left(\frac{1}{n(n-1)}\sum_{i=1}^{n}\sum_{\substack{j=1\\j\neq i}}^{n}z_j\xi_i\right).$$

Combining the two, we get

$$\left(\mathbf{I}_d - \mathbf{u}\mathbf{u}^\top\right)\mathbf{S}_n\widetilde{\mathbf{u}} = \left(\mathbf{I}_d - \mathbf{u}\mathbf{u}^\top\right)\underbrace{\left(\frac{1}{n-1}\sum_{i=1}^{n}\left(z_i - \bar{z}\right)\left(\xi_i - \bar{\xi}\right)\right)}_{\mathbf{w}_n}.$$

Now we want to compute a high confidence lower bound on the norm of the above. We first relate $\left\|\left(\mathbf{I}_d - \mathbf{u}\mathbf{u}^\top\right)\mathbf{w}_n\right\|$ to $\|\mathbf{w}_n\|$. This is because $\mathbf{w}_n$ is spherically symmetric, and $\left(\mathbf{I}_d - \mathbf{u}\mathbf{u}^\top\right)$ is a rank-$(d-1)$ matrix with all non-zero eigenvalues equal to one. We get

$$\left\|\left(\mathbf{I}_d - \mathbf{u}\mathbf{u}^\top\right)\mathbf{w}_n\right\| = \|\mathbf{w}_n\| \cdot \sqrt{1 - \left(\mathbf{u}^\top\hat{\mathbf{w}}_n\right)^2}.$$

Now due to $\mathbf{w}_n$ being spherically symmetric, $\|\mathbf{w}_n\|$ (the magnitude) and $\hat{\mathbf{w}}_n$ (the direction) are independent random quantities. Further, $\hat{\mathbf{w}}_n$ is uniformly distributed on $\mathbf{S}^{d-1}$.

For $\|\mathbf{w}_n\|$, we will use sharp Gaussian concentration. The intuition is that $\|\mathbf{w}_n\|$ cannot be too smaller than $\sqrt{d\gamma^{-1}/n}$, for large $d$. Concretely, it holds that

$$\text{w.p. } 1 - \delta, \ \left\|\frac{1}{n-1}\sum_{i=1}^{n}\left(z_i - \bar{z}\right)\left(\xi_i - \bar{\xi}\right)\right\| \geq \sqrt{\frac{\gamma^{-1}}{n}} \cdot \left(\sqrt{d} - \sqrt{2\ln\frac{1}{\delta}} - \sqrt{2}\right). \tag{37}$$

An appropriate choice of $\delta = \exp(-d/4)$, which results in

$$\text{w.p. } 1 - \exp\left(-d/4\right), \ \|\mathbf{w}_n\| \gtrsim \sqrt{\frac{d\gamma^{-1}}{n}}. \tag{38}$$

For the second term (with the direction $\hat{\mathbf{w}}_n$), this will be at least $\Omega(1)$ with high probability, since $\mathbf{u}^\top\hat{\mathbf{w}}_n$ will be large only with very small probability when then dimension $d$ is big enough. Concretely, it holds that

$$\text{w.p. } 1 - 2\exp\left(-d/4\right), \ \sqrt{1 - \left(\mathbf{u}^\top\hat{\mathbf{w}}_n\right)^2} \geq \sqrt{\frac{1}{2}}. \tag{39}$$

Overall, for the numerator, we conclude that

$$\text{w.p. } 1 - c\exp\left(-d/4\right), \ \text{Numerator} \gtrsim \sqrt{\frac{d\gamma^{-1}}{n}}. \tag{40}$$

**Step 3.2. For the denominator:** We need a high confidence upper bound on $\|\mathbf{S}_n\|$. We can use Matrix-Bernstein type analysis. Note that $\mathbb{E}[\mathbf{S}_n] = \eta\,\mathbf{u}\widetilde{\mathbf{u}}^\top$. And the deviation is dominated by

$$\mathbf{S}_n - \mathbb{E}\mathbf{S}_n \approx \frac{1}{n}\sum_{i\in[n]} z_i\xi_i\widetilde{\mathbf{u}}^\top + \frac{1}{n}\sum_{i\in[n(1-\eta)]} z_i\widetilde{z}_i\mathbf{u}\widetilde{\mathbf{u}}^\top\ .$$

Again, the dominating term is the first one. This means that we only have to show high confidence upper bound on $\|(1/n)\sum_i z_i\xi_i\|$, and hence the problem has reduced to vector concentration instead of matrix concentration. Analogous to Eq. (37), one can show

$$\text{w.p. } 1-\delta,\ \left\|\frac{1}{n}\sum_{i=1}^n z_i\xi_i\right\| \le \sqrt{\frac{\gamma^{-1}}{n}}\cdot\left(\sqrt{d}+\sqrt{2\ln\frac{1}{\delta}}\right)\ . \tag{41}$$

Overall, using the triangle inequality, we have

$$\text{w.p. } 1-\exp(-d/4),\ \|\mathbf{S}_n\| \le \underbrace{\|\mathbb{E}\mathbf{S}_n\|}_{=\eta}+2\sqrt{\frac{d\,\gamma^{-1}}{n}}\ . \tag{42}$$

**Step 4. Combined result:** From 3.1 and 3.2, for $n \ge 4d\gamma^{-1}/\eta^2$ (so the high-conf UB for $\|\mathbf{S}_n\|$ is $2\eta$),

$$\text{w.p. } 1-O\left(\exp(-d/4)\right),\ |\sin\theta(\mathbf{u}_n,\mathbf{u})| \gtrsim \frac{1}{\eta}\sqrt{\frac{d\,\gamma^{-1}}{n}}\ . \tag{43}$$

# F  Characterizing the score distribution of the oracle

The Bernoulli variable $c \in \{0,1\}$ captures the status of clean/corrupted nature of a sample. We first characterize the score distribution in both cases separately, and then create the relevant mixture distribution using the proportions $\eta, 1-\eta$ for clean, corrupted samples respectively.

Before the calculations, we state some Lemmas that will be used.

**Lemma 12.** *Let $X$ be distributed as $\mathcal{N}(0,\Omega)$. For a fixed matrix $\mathbf{A}$, it holds:*

$$\mathbb{E}[X^\top \mathbf{A}X] = \mathrm{Tr}\left(\mathbf{A}\Omega\right)\ ,$$
$$\mathbb{V}[X^\top \mathbf{A}X] = \frac{1}{2}\mathrm{Tr}\left(\left(\mathbf{A}+\mathbf{A}^\top\right)\Omega\left(\mathbf{A}+\mathbf{A}^\top\right)\Omega\right)\ .$$

**Lemma 13.** *Let $X$ be distributed as $\mathcal{N}(0,\Omega)$, and $\widetilde{X}$ be distributed as $\mathcal{N}(0,\widetilde{\Omega})$. Let $X,\widetilde{X}$ be independent of each other. For a fixed matrix $\mathbf{A}$, it holds:*

$$\mathbb{E}[X^\top \mathbf{A}\widetilde{X}] = 0\ ,$$
$$\mathbb{V}[X^\top \mathbf{A}\widetilde{X}] = \mathrm{Tr}\left(\Omega\mathbf{A}\widetilde{\Omega}\mathbf{A}^\top\right)\ .$$

Consider a block matrix $\mathbf{X}$ given as below

$$\mathbf{X} = \left[\begin{array}{cc} \mathbf{A} & \mathbf{B} \\ \mathbf{C} & \mathbf{D} \end{array}\right]\ .$$

**Lemma 14.** *For a block matrix $\mathbf{X}$ given as above, it holds that*

$$\mathrm{Tr}(\mathbf{X}) = \mathrm{Tr}(\mathbf{A}) + \mathrm{Tr}(\mathbf{D})\ .$$

**Lemma 15.** *For a block matrix $\mathbf{X}$ given as above, with $\mathbf{A},\mathbf{D}$ are square matrices, it holds that*

$$\mathbf{X}^2 = \left[\begin{array}{cc} \mathbf{A}^2+\mathbf{BC} & \mathbf{AB}+\mathbf{BD} \\ \mathbf{CA}+\mathbf{DC} & \mathbf{CB}+\mathbf{D}^2 \end{array}\right]\ .$$

**Case 0: Corrupted samples** ($c=0$ case). Let $Z_0 \stackrel{d}{=} \{S(x,\widetilde{x}; \mathbf{U}\widetilde{\mathbf{U}}^\top) \mid c=0\}$, with distribution $\mathcal{D}_0$. This (scalar) random variable is equivalent to $X^\top \mathbf{U}\widetilde{\mathbf{U}}^\top \widetilde{X}$, where $X,\widetilde{X}$ are independent and follow $X \sim \mathcal{N}\left(0, \mathbf{U}\mathbf{U}^\top + \gamma^{-1}\mathbf{I}_d\right)$, $\widetilde{X} \sim \mathcal{N}\left(0, \widetilde{\mathbf{U}}\widetilde{\mathbf{U}}^\top + \widetilde{\gamma}^{-1}\mathbf{I}_{\widetilde{d}}\right)$. This is in-line with Remark A.1. We invoke Lemma 13 to get the first two moments.

1. Mean: $0$.

2. Variance: $r\,(1+\gamma^{-1})(1+\widetilde{\gamma}^{-1})$.

$$\text{Variance} = \text{Tr}\left(\left(\mathbf{U}\mathbf{U}^{\top} + \gamma^{-1}\,\mathbf{I}_d\right)\,\mathbf{U}\widetilde{\mathbf{U}}^{\top}\,\left(\widetilde{\mathbf{U}}\widetilde{\mathbf{U}}^{\top} + \widetilde{\gamma}^{-1}\,\mathbf{I}_{\widetilde{d}}\right)\,\widetilde{\mathbf{U}}\mathbf{U}^{\top}\right)$$

$$= \text{Tr}\left(\mathbf{U}^{\top}\left(\mathbf{U}\mathbf{U}^{\top} + \gamma^{-1}\,\mathbf{I}_d\right)\mathbf{U}\;\;\widetilde{\mathbf{U}}^{\top}\left(\widetilde{\mathbf{U}}\widetilde{\mathbf{U}}^{\top} + \widetilde{\gamma}^{-1}\,\mathbf{I}_{\widetilde{d}}\right)\widetilde{\mathbf{U}}\right)$$

$$= \text{Tr}\left(\left(\mathbf{I}_r + \gamma^{-1}\mathbf{I}_r\right)\left(\mathbf{I}_r + \widetilde{\gamma}^{-1}\mathbf{I}_r\right)\right).$$

3. Tails: Since $X, \widetilde{X}$ are independent, the tails are described by the quadratic form on two independent Gaussians. This random variable is $(i)$ symmetric, and $(ii)$ uni-modal, and the tails decay exponentially.

**Case 1: Clean samples** ($c = 1$ case). Let $Z_1 \stackrel{d}{=} \{S(x, \widetilde{x}; \mathbf{U}\widetilde{\mathbf{U}}^{\top}) \mid c = 1\}$, with distribution $\mathcal{D}_1$. This random variable is equivalent to $X^{\top}\mathbf{B}X$, where $X = [x, \widetilde{x}]^{\top}$ follows $X \sim \mathcal{N}\left(0, \Sigma_1\right)$ (refer to Remark A.1); and $\mathbf{B}$ is a block matrix given as below. We invoke Lemma 12 to get the first two moments.

$$\mathbf{B} = \left[\begin{array}{cc} \mathbf{0}_{d\times d} & \mathbf{U}\widetilde{\mathbf{U}}^{\top} \\ \mathbf{0}_{\widetilde{d}\times d} & \mathbf{0}_{\widetilde{d}\times\widetilde{d}} \end{array}\right]_{(d+\widetilde{d})\times(d+\widetilde{d})}$$

1. Mean: $r$.

$$\text{Mean} = \text{Tr}(\mathbf{B}\Sigma_1)$$

$$= \text{Tr}\left(\left[\begin{array}{cc} \mathbf{U}\mathbf{U}^{\top} & \cdot \\ \cdot & \mathbf{0} \end{array}\right]\right)$$

$$= \text{Tr}(\mathbf{U}\mathbf{U}^{\top}) = \text{Tr}(\mathbf{I}_r) = r. \qquad\qquad \text{(Using Lemma 14)}$$

2. Variance: $r + r\,(1+\gamma^{-1})(1+\widetilde{\gamma}^{-1})$.

$$\text{Variance} = \frac{1}{2}\text{Tr}\left(\left(\mathbf{B} + \mathbf{B}^{\top}\right)\Sigma_1\left(\mathbf{B} + \mathbf{B}^{\top}\right)\Sigma_1\right)$$

$$= \frac{1}{2}\text{Tr}\left(\left[\begin{array}{cc} \mathbf{U}\mathbf{U}^{\top} & \overbrace{\mathbf{U}\widetilde{\mathbf{U}}^{\top} + \widetilde{\gamma}^{-1}\mathbf{U}\widetilde{\mathbf{U}}^{\top}}^{\mathbf{T}_1} \\ \underbrace{\widetilde{\mathbf{U}}\mathbf{U}^{\top} + \gamma^{-1}\widetilde{\mathbf{U}}\mathbf{U}^{\top}}_{\mathbf{T}_2} & \widetilde{\mathbf{U}}\widetilde{\mathbf{U}}^{\top} \end{array}\right]^2\right)$$

$$= \frac{1}{2}\text{Tr}\left(\left[\begin{array}{cc} \mathbf{U}\mathbf{U}^{\top} + \mathbf{T}_1\mathbf{T}_2 & \cdot \\ \cdot & \mathbf{T}_2\mathbf{T}_1 + \widetilde{\mathbf{U}}\widetilde{\mathbf{U}}^{\top} \end{array}\right]\right) \qquad \text{(Using Lemma 15)}$$

$$= \text{Tr}(\mathbf{I}_r) + \text{Tr}(\mathbf{T}_1\mathbf{T}_2). \qquad\qquad \text{(Using Lemma 14)}$$

3. Tails: Since $X, \widetilde{X}$ are dependent, the tails are described by the quadratic form on two dependent Gaussians. The tails decay exponentially, and are described by the Hanson-Wright inequality. A similar calculation as the variance provides the exact parameters, and the inequality becomes:

$$\mathbb{P}\left(|Z_1 - \mathbb{E}Z_1| > t\right) \lesssim \exp\left(-c\min\left\{\frac{2\,t^2}{r\,(1+(1+\gamma^{-1})(1+\widetilde{\gamma}^{-1}))},\right.\right.$$

$$\left.\left.\frac{\sqrt{2}\,t}{\sqrt{r\,(1+(1+\gamma^{-1})(1+\widetilde{\gamma}^{-1}))}}\right\}\right). \qquad (44)$$

# G   A proof of Theorem 1

In this section, we present a proof of Theorem 1. We first define one additional piece of notation. For $\mathbf{U}$, let $\mathbf{U}_\perp \in \mathbb{R}^{d \times (d-r)}$ denote the completion of the orthonormal basis. That is, the matrix $\mathbf{U}_{\text{full}} = [\mathbf{U} \; \mathbf{U}_\perp] \in \mathbb{R}^{d \times d}$ is such that $\mathbf{U}_{\text{full}}^\top \mathbf{U}_{\text{full}} = \mathbf{I}_d = \mathbf{U}_{\text{full}} \mathbf{U}_{\text{full}}^\top$. Similarly define $\widetilde{\mathbf{U}}_\perp \in \mathbb{R}^{\widetilde{d} \times (\widetilde{d}-r)}$.

Recall that we have $n$ samples of the form $\{(x_i, \widetilde{x}_i)\}_{i=1}^n$, i.i.d from the mixture distribution (with $\eta, 1-\eta$ ratios for clean, corrupted respectively). Let $n_{\text{T}}$ samples be used to train the teacher, and let $N = n_{\text{T}} - n$ samples be used to train the student. Let $\rho_{\text{T}}, \rho$ be the respective regularization parameters, and let $(\mathbf{G}_{\text{T}}, \widetilde{\mathbf{G}}_{\text{T}}), (\mathbf{G}, \widetilde{\mathbf{G}})$ denote the respective embedding matrices at the solution of Eq. (3). Consider a general threshold $\theta \in \mathbb{R}$ that is used to filter the dataset based on the teacher scores. Note that we have ensured that $\theta$ is independent of the $N$ samples to be filtered, since it depends only on the $n_{\text{T}}$ samples used for teacher training. For the teacher, from Corollary 1, we know that with probability $1 - \exp(-\Omega(\max\{d, \widetilde{d}\}))$:

$$\left\| \mathbf{G}_{\text{T}}^\top \widetilde{\mathbf{G}}_{\text{T}} - \frac{\eta}{\rho_{\text{T}}} \mathbf{U} \widetilde{\mathbf{U}}^\top \right\| \leq \frac{1}{\rho_{\text{T}}} \left( \sqrt{\frac{\max\{d, \widetilde{d}\} \, (1 + \gamma^{-1}) \, (1 + \widetilde{\gamma}^{-1})}{n_{\text{T}}}} + \widetilde{O}\left(\frac{1}{n_{\text{T}}}\right) \right) . \tag{45}$$

Here $(\mathbf{G}_{\text{T}}, \widetilde{\mathbf{G}}_{\text{T}})$ are random quantities that depend on the $n_{\text{T}}$ samples used. For the rest of the analysis, we will assume them to be fixed (since they don't depend on the randomness of the remaining $N$ samples). Finally, we will give a high probability guarantee that will use the confidence bound in Eq. (45) as one of the terms in the combined error bound, with an appropriate choice of $n_{\text{T}}$ and $\rho_{\text{T}}$. We now study the student with data filtering. It is useful to define

$$\mathbf{M}_{\text{T}} := \mathbf{G}_{\text{T}}^\top \widetilde{\mathbf{G}}_{\text{T}} , \quad \mathbf{M}_{\text{O}} := (\eta/\rho_{\text{T}}) \, \mathbf{U} \widetilde{\mathbf{U}}^\top . \tag{46}$$

These are the matrices used for scoring the samples by the teacher and its oracle version, respectively. Note that $\text{rank}(\mathbf{M}_{\text{O}}) = r$ since both $\mathbf{U}, \widetilde{\mathbf{U}}$ are rank-$r$ matrices. From the teacher guarantee in Eq. (45), it holds that $\mathbf{M}_{\text{T}} \to \mathbf{M}_{\text{O}}$ as $n_{\text{T}} \to \infty$. Recall that the scoring function is $S(x, \widetilde{x}; \mathbf{M}) = x^\top \mathbf{M} \widetilde{x}$, and a sample $(x, \widetilde{x})$ is selected/retained iff $S(x, \widetilde{x}; \mathbf{M}_{\text{T}}) > \theta$.

We define certain quantities that will be central to the analysis. Akin to Eq. (5), we define the empirical cross-covariance matrix of the data *after selection* in Eq. (47). Let $n_{\text{sel,T}}(\theta)$ be the number of samples selected, which is a random variable with $\mathbb{E}[n_{\text{sel,T}}(\theta)] = N \, P_{\text{T}}(\theta)$. Let $I_{\text{sel,T}}(\theta) \subseteq [N]$ denote the indices of the points selected. That is, $i \in I_{\text{sel,T}}(\theta) \iff S(x_i, \widetilde{x}_i; \mathbf{M}_{\text{T}}) > \theta$. Similarly, define $n_{\text{sel,O}}(\theta)$ and $I_{\text{sel,O}}(\theta)$. Construct the empirical cross-covariance matrix for the filtered dataset:

$$\mathbf{S}_{N,\text{T}}(\theta) := \frac{1}{n_{\text{sel,T}}(\theta) - 1} \underbrace{\sum_{i \in I_{\text{sel,T}}(\theta)} (x_i - \overline{x}(\theta)) \left(\widetilde{x}_i - \overline{\widetilde{x}}(\theta)\right)^\top}_{\mathbf{Q}_{N,\text{T}}(\theta)} . \tag{47}$$

To analyze its asymptotic limit, we define $\mathbf{S}(\theta)$ as the limit of the cross-covariance, for both the teacher and the oracle. Similarly, let $P(\theta)$ denote the probability mass of data that is retained (also in the limit of $n \to \infty$), for both the teacher and the oracle. These are described in Eqs (48), (49).

$$\mathbf{S}_{\text{T}}(\theta) = \mathbb{E}\left[x\widetilde{x}^\top \mid S(x, \widetilde{x}; \mathbf{M}_{\text{T}}) > \theta\right] \in \mathbb{R}^{d \times \widetilde{d}} , \quad P_{\text{T}}(\theta) = \mathbb{P}\left\{S(x, \widetilde{x}; \mathbf{M}_{\text{T}}) > \theta\right\} ; \tag{48}$$

$$\mathbf{S}_{\text{O}}(\theta) = \mathbb{E}\left[x\widetilde{x}^\top \mid S(x, \widetilde{x}; \mathbf{M}_{\text{O}}) > \theta\right] \in \mathbb{R}^{d \times \widetilde{d}} , \quad P_{\text{O}}(\theta) = \mathbb{P}\left\{S(x, \widetilde{x}; \mathbf{M}_{\text{O}}) > \theta\right\} . \tag{49}$$

Note that $\mathbf{S}_{\text{T}}(\theta), \mathbf{S}_{\text{O}}(\theta)$ are the limits of $\mathbf{S}_{N,\text{T}}(\theta), \mathbf{S}_{N,\text{O}}(\theta)$ as $N \to \infty$. The threshold $\theta \to -\infty$ recovers the no filtering case, i.e. both $\mathbf{S}_{N,\text{T}}(\theta), \mathbf{S}_{N,\text{O}}(\theta)$ approach $\mathbf{S}_N$. We will now follow proof steps similar to Section D. Steps 1 and 2 hold for a general cross covariance matrix, and can be used directly. Steps 3 and 4 are concerned with the limit of $\mathbf{S}_n(\theta)$ as $n \to \infty$, and how it concentrates around the limit. These steps will change significantly. Finally, we will be able to reuse Lemma 2 for step 5. We detail each of these proof steps below.

**Step 1.** Following the exact same proof steps as in Section D, the unregularized contrastive loss objective on the $n_{\text{sel,T}}(\theta)$ samples is equivalent to

$$\mathcal{L}_0(\mathbf{G}, \widetilde{\mathbf{G}}) = -\text{Tr}\left(\mathbf{G} \, \mathbf{S}_{N,\text{T}}(\theta) \, \widetilde{\mathbf{G}}^\top\right) . \tag{50}$$

**Step 2.** Again, following the exact same proof steps as in Section D, the solution to the $\rho$-regularized minimization problem is given by

$$\arg\min_{\mathbf{G},\widetilde{\mathbf{G}}} \mathcal{L}_\rho\left(\mathbf{G},\widetilde{\mathbf{G}}\right) = \left\{ \left(\mathbf{G},\widetilde{\mathbf{G}}\right) \;\middle|\; \mathbf{G}^\top\widetilde{\mathbf{G}} = \frac{1}{\rho}\,\text{SVD}_r\left(\mathbf{S}_{N,\text{T}}(\theta)\right) \right\} . \tag{51}$$

**Step 3.** This step changes from Section D. We use the following:

$$\left\|\text{SVD}_r\left(\mathbf{S}_{N,\text{T}}(\theta)\right) - \mathbf{S}_\text{O}(\theta)\right\| \leq \sigma_{r+1}\left(\mathbf{S}_\text{O}(\theta)\right) + 2\left\|\mathbf{S}_{N,\text{T}}(\theta) - \mathbf{S}_\text{O}(\theta)\right\| . \tag{52}$$

By triangle inequality, we have

$$\left\|\text{SVD}_r\left(\mathbf{S}_{N,\text{T}}(\theta)\right) - \mathbf{S}_\text{O}(\theta)\right\| \leq \left\|\text{SVD}_r\left(\mathbf{S}_{N,\text{T}}(\theta)\right) - \mathbf{S}_{N,\text{T}}(\theta)\right\| + \left\|\mathbf{S}_{N,\text{T}}(\theta) - \mathbf{S}_\text{O}(\theta)\right\| .$$

And for the first term on the right hand side, we use

$$\left\|\text{SVD}_r\left(\mathbf{S}_{N,\text{T}}(\theta)\right) - \mathbf{S}_{N,\text{T}}(\theta)\right\| = \sigma_{r+1}\left(\mathbf{S}_{N,\text{T}}(\theta)\right)$$
$$\leq^{(\dagger)} \sigma_{r+1}\left(\mathbf{S}_\text{O}(\theta)\right) + \left\|\mathbf{S}_{N,\text{T}}(\theta) - \mathbf{S}_\text{O}(\theta)\right\| ,$$

where we used Lemma 1 in Eq ($\dagger$).

**Step 3'.** Analysis of $\mathbf{S}_\text{O}(\theta)$: The main difference in Eq. (23) and Eq. (52) is the term $\sigma_{r+1}(\mathbf{S}_\text{O}(\theta))$. This additional step of the proof analyzes the properties of $\mathbf{S}_\text{O}(\theta)$. In particular, we will show that $\mathbf{S}_\text{O}(\theta)$ is rank-$r$, and hence $\sigma_{r+1}(\mathbf{S}_\text{O}(\theta)) = 0$. Additionally, we establish upper and lower bounds on the singular values of $\mathbf{S}_\text{O}(\theta)$ that will be used later in the proof. From Eq. (49), we simplify to write

$$\mathbf{S}_\text{O}(\theta) = \mathbb{E}\left[ x\widetilde{x}^\top \;\middle|\; x^\top \mathbf{U}\widetilde{\mathbf{U}}^\top\widetilde{x} > \frac{\theta\rho_\text{T}}{\eta} \right] ,$$

where $(x,\widetilde{x})$ is drawn from the mixture model: $\eta \cdot \mathcal{N}(0,\Sigma_1) + (1-\eta) \cdot \mathcal{N}(0,\Sigma_0)$. To simplify notation, define $\ddot{\theta} := {(\theta\rho_\text{T})}/{\eta}$. From the conditioning event, it seems that $\mathbf{U}^\top x$ and $\widetilde{\mathbf{U}}^\top\widetilde{x}$ is a good 'basis' for a decomposition. Pre-multiply and post-multiply to recover this basis for the $x\widetilde{x}^\top$ term inside the expectation as

$$\mathbf{S}_\text{O}(\theta) = \underbrace{\mathbf{U}_\text{full}\mathbf{U}_\text{full}^\top}_{=\mathbf{I}_d} \mathbb{E}\left[ x\widetilde{x}^\top \;\middle|\; x^\top\mathbf{U}\widetilde{\mathbf{U}}^\top\widetilde{x} > \ddot{\theta} \right] \underbrace{\widetilde{\mathbf{U}}_\text{full}\widetilde{\mathbf{U}}_\text{full}^\top}_{=\mathbf{I}_{\widetilde{d}}}$$

$$= \mathbf{U}_\text{full}\, \mathbb{E}\left[ \left( \begin{array}{cc} \overbrace{(\mathbf{U}^\top x)(\widetilde{\mathbf{U}}^\top\widetilde{x})^\top}^{r\times r} & \overbrace{(\mathbf{U}^\top x)(\widetilde{\mathbf{U}}_\perp^\top\widetilde{x})^\top}^{r\times(\widetilde{d}-r)} \\ \underbrace{(\mathbf{U}_\perp^\top x)(\widetilde{\mathbf{U}}^\top\widetilde{x})^\top}_{(d-r)\times r} & \underbrace{(\mathbf{U}_\perp^\top x)(\widetilde{\mathbf{U}}_\perp^\top\widetilde{x})^\top}_{(d-r)\times(\widetilde{d}-r)} \end{array} \right) \;\middle|\; (\mathbf{U}^\top x)^\top(\widetilde{\mathbf{U}}^\top\widetilde{x}) > \ddot{\theta} \right] \widetilde{\mathbf{U}}_\text{full}^\top .$$

Call the top left entry in this decomposition to be the 'dominant', and the other three as 'non-dominant'. We will show the non-dominant entries will be zero. The following reparametrization makes things cleaner.

$$\mathbf{U}^\top x = z + \underbrace{\mathbf{U}^\top\xi}_{\varepsilon}, \;\; \mathbf{U}_\perp^\top x = \underbrace{\mathbf{U}_\perp^\top\xi}_{\varepsilon_\perp} ; \qquad \widetilde{\mathbf{U}}^\top\widetilde{x} = \widetilde{z} + \underbrace{\widetilde{\mathbf{U}}^\top\widetilde{\xi}}_{\widetilde{\varepsilon}}, \;\; \widetilde{\mathbf{U}}_\perp^\top\widetilde{x} = \underbrace{\widetilde{\mathbf{U}}_\perp^\top\widetilde{\xi}}_{\widetilde{\varepsilon}_\perp} .$$

Let's further simplify the expressions with another transformation. The subscripts $S, N$ denote the signal (containing some noise) and noise part.

$$\underbrace{x_S}_{\in\mathbb{R}^r} \leftarrow z + \varepsilon, \;\; \underbrace{x_N}_{\in\mathbb{R}^{d-r}} \leftarrow \varepsilon_\perp ; \quad \underbrace{\widetilde{x}_S}_{\in\mathbb{R}^r} \leftarrow \widetilde{z} + \widetilde{\varepsilon}, \;\; \underbrace{\widetilde{x}_N}_{\in\mathbb{R}^{\widetilde{d}-r}} \leftarrow \widetilde{\varepsilon}_\perp .$$

Due to the diagonal structure of $\Sigma_\xi, \Sigma_{\widetilde{\xi}}$, we infer the distributions as

$$\varepsilon \sim \mathcal{N}\left(0,\frac{1}{\gamma}\mathbf{I}_r\right), \;\; \varepsilon_\perp \sim \mathcal{N}\left(0,\frac{1}{\gamma}\mathbf{I}_{(d-r)}\right); \quad \widetilde{\varepsilon} \sim \mathcal{N}\left(0,\frac{1}{\widetilde{\gamma}}\mathbf{I}_r\right), \;\; \widetilde{\varepsilon}_\perp \sim \mathcal{N}\left(0,\frac{1}{\widetilde{\gamma}}\mathbf{I}_{(\widetilde{d}-r)}\right).$$

And crucially, due to the diagonal structure of $\Sigma_\xi, \Sigma_{\widetilde{\xi}}$, we infer that $\{\varepsilon, \varepsilon_\perp, \widetilde{\varepsilon}, \widetilde{\varepsilon}_\perp\}$ are all *mutually independent*, and independent of $z, \widetilde{z}$. This entails that the transformed vector is Gaussian with mean zero and covariance given as below.

$$
\begin{pmatrix} x_S \\ x_N \\ \widetilde{x}_S \\ \widetilde{x}_N \end{pmatrix} \sim \mathcal{N}\left( 0, \begin{pmatrix} (1 + 1/\gamma)\,\mathbf{I}_r & \mathbf{0} & \mathbf{0}\,(\mathbf{I}_r) & \mathbf{0} \\ . & (1/\gamma)\,\mathbf{I}_{(d-r)} & \mathbf{0} & \mathbf{0} \\ .\,(.) & . & (1 + 1/\widetilde{\gamma})\,\mathbf{I}_r & \mathbf{0} \\ . & . & . & (1/\widetilde{\gamma})\,\mathbf{I}_{(\widetilde{d}-r)} \end{pmatrix} \right) . \tag{53}
$$

The above is for the corrupted case (w.p. $1 - \eta$). In the clean case (w.p. $\eta$), the blue entries change to $\mathbf{I}_r$ due to the relation of $z = \widetilde{z}$. Our $\mathbb{E}[.]$ notation includes the expectation over this randomness along with the randomness of $x, \widetilde{x}$. Denote by $\Omega_0$ and $\Omega_1$ the covariances of the signal part, i.e. $(x_S, \widetilde{x}_S)$ in these two cases:

$$
\Omega_0 := \begin{pmatrix} (1 + 1/\gamma)\,\mathbf{I}_r & \mathbf{0} \\ \mathbf{0} & (1 + 1/\widetilde{\gamma})\,\mathbf{I}_r \end{pmatrix} , \quad \Omega_1 := \begin{pmatrix} (1 + 1/\gamma)\,\mathbf{I}_r & \mathbf{I}_r \\ \mathbf{I}_r & (1 + 1/\widetilde{\gamma})\,\mathbf{I}_r \end{pmatrix} . \tag{54}
$$

Overall, under the transformation, the expectation simplifies to

$$
\mathbf{S}_O(\theta) = \mathbf{U}_{\text{full}}\,\mathbb{E}\left[ \begin{pmatrix} x_S\widetilde{x}_S^\top & x_S\widetilde{x}_N^\top \\ x_N\widetilde{x}_S^\top & x_N\widetilde{x}_N^\top \end{pmatrix} \,\middle|\, x_S^\top\widetilde{x}_S > \ddot{\theta} \right] \widetilde{\mathbf{U}}_{\text{full}}^\top . \tag{55}
$$

Due to $x_N, \widetilde{x}_N$ being independent of all other entries via Eq. (53), and since the conditioning event in Eq. (55) only involves $x_S, \widetilde{x}_S$, we conclude that the non-dominant entries in the expectation will be *zero*. Hence we are left with the simplified rank-$r$ form for the $d \times \widetilde{d}$ matrix:

$$
\mathbf{S}_O(\theta) = \mathbf{U}\,\mathbb{E}\left[ x_S\widetilde{x}_S^\top \,|\, x_S^\top\widetilde{x}_S > \ddot{\theta} \right] \widetilde{\mathbf{U}}^\top = \mathbf{U}\left( \eta \cdot \mathbb{E}_{(x_S,\widetilde{x}_S)\sim\mathcal{N}(0,\Omega_1)}\left[ x_S\widetilde{x}_S^\top \,|\, x_S^\top\widetilde{x}_S > \ddot{\theta} \right] \right.
$$
$$
\left. + \; (1 - \eta)\cdot\mathbb{E}_{(x_S,\widetilde{x}_S)\sim\mathcal{N}(0,\Omega_0)}\left[ x_S\widetilde{x}_S^\top \,|\, x_S^\top\widetilde{x}_S > \ddot{\theta} \right] \right) \widetilde{\mathbf{U}}^\top .
$$

We will now use Lemma 9 to simplify both the terms above. Note that $\Omega_1, \Omega_0$ satisfy the lemma's requirement of the block diagonal covariance.

$$
\mathbf{S}_O(\theta) = \mathbf{U}\left( \eta\, f_1(\theta)\,\mathbf{I}_r + (1 - \eta)\, f_0(\theta)\,\mathbf{I}_r \right) \widetilde{\mathbf{U}}^\top = \left( \eta\, f_1(\theta) + (1 - \eta)\, f_0(\theta) \right) \mathbf{U}\widetilde{\mathbf{U}}^\top , \tag{56}
$$

where the following conditions hold on $f_1, f_0$ (converting back from $\ddot{\theta}$ to $\theta$):

$$
\max\{1, (\theta\rho_T)/\eta\, r\} + e\,\sqrt{((1+\gamma^{-1})(1+\widetilde{\gamma}^{-1})+1)/r} \geq f_1(\theta) \geq \max\{1, (\theta\rho_T)/\eta\, r\} ,
$$
$$
\max\{0, (\theta\rho_T)/\eta\, r\} + e\,\sqrt{((1+\gamma^{-1})(1+\widetilde{\gamma}^{-1}))/r} \geq f_0(\theta) \geq \max\{0, (\theta\rho_T)/\eta\, r\} .
$$

Using the above equations, and the special case of $\theta = 0$ in Lemma 9, we conclude:

$$
f_1(0) \geq 1, \quad f_0(0) \geq \frac{2}{\pi r} \cdot \sqrt{(1 + \gamma^{-1})(1 + \widetilde{\gamma}^{-1})} , \tag{57}
$$

$$
f_1\left( \frac{r\eta}{2\rho_T} \right) \geq 1, \quad f_0\left( \frac{r\eta}{2\rho_T} \right) \geq \frac{1}{2} . \tag{58}
$$

We will use these inequalities in step 5. In particular, since $\|\mathbf{S}_O(\theta)\| = \eta\, f_1(\theta) + (1 - \eta)\, f_0(\theta)$,

$$
\text{for } \theta \in [0, r\eta/2\rho_T] , \quad \|\mathbf{S}_O(\theta)\| \geq \frac{2}{\pi r} \cdot \sqrt{(1 + \gamma^{-1})(1 + \widetilde{\gamma}^{-1})} . \tag{59}
$$

**Step 4.** Concentration of $\mathbf{S}_{N,T}(\theta)$ to $\mathbf{S}_O(\theta)$: We break this into subparts as below.

**Step 4.1.** Concentration of $\mathbf{S}_{N,T}(\theta)$ to $\mathbf{S}_T(\theta)$: Using the below substeps, we show that with probability $1 - \exp(-\Omega(\max\{d, \widetilde{d}\}))$:

$$
\|\mathbf{S}_{N,T}(\theta) - \mathbf{S}_T(\theta)\| \leq \sqrt{\frac{\max\{d, \widetilde{d}\}\,\text{poly}(\gamma^{-1}, \widetilde{\gamma}^{-1})}{N\,P_T(\theta)}} + \widetilde{O}\left( \frac{1}{N\,P_T(\theta)} \right) . \tag{60}
$$

**Step 4.1.1.** Replacing the random denominator: Recall $n_{\text{sel,T}}(\theta) = \sum_{i=1}^{N} \mathbb{I}\{S(x_i, \widetilde{x}_i; \mathbf{M}_{\text{T}}) > \theta\}$ is the (random) number of selected samples. Since the teacher's score matrix $\mathbf{M}_{\text{T}}$ and threshold $\theta$ are fixed independently of these $N$ samples, the indicators are i.i.d. Bernoulli random variables with mean $P_{\text{T}}(\theta)$. By a standard Chernoff bound for sums of independent Bernoulli variables, $n_{\text{sel,T}}(\theta)$ concentrates sharply around its expectation: for any $0 < \delta < 1$,

$$\mathbb{P}\Big\{ |n_{\text{sel,T}}(\theta) - NP_{\text{T}}(\theta)| \geq \delta \, NP_{\text{T}}(\theta) \Big\} \;\leq\; 2\exp\Big(-\Omega(\delta^2 \, NP_{\text{T}}(\theta))\Big).$$

In particular, choosing $\delta = \left( \sqrt{\max\{d, \widetilde{d}\}/N\,P_{\text{T}}(\theta)} \right)$, we conclude that

$$\text{w.p. } 1 - \exp(-\Omega(\max\{d, \widetilde{d}\})), \quad n_{\text{sel,T}}(\theta) = \left( 1 \pm \sqrt{\frac{\max\{d, \widetilde{d}\}}{N\,P_{\text{T}}(\theta)}} \right) NP_{\text{T}}(\theta). \tag{61}$$

On this high-probability event, the following holds (recall the definition of $\mathbf{Q}_{N,\text{T}}(\theta)$ from Eq. (47)).

$$\begin{aligned}
\left\| \frac{1}{n_{\text{sel,T}}(\theta) - 1} \mathbf{Q}_{N,\text{T}}(\theta) - \frac{1}{NP_{\text{T}}(\theta) - 1} \mathbf{Q}_{N,\text{T}}(\theta) \right\| &= \frac{|n_{\text{sel,T}}(\theta) - N\,P_{\text{T}}(\theta)|}{(n_{\text{sel,T}}(\theta) - 1)\,(N\,P_{\text{T}}(\theta) - 1)} \|\mathbf{Q}_{N,\text{T}}(\theta)\| \\
&\lesssim^{(\dagger)} \frac{|n_{\text{sel,T}}(\theta) - N\,P_{\text{T}}(\theta)|}{(N\,P_{\text{T}}(\theta) - 1)^2} \|\mathbf{Q}_{N,\text{T}}(\theta)\| \\
&\lesssim^{(\dagger\dagger)} \frac{\sqrt{\max\{d, \widetilde{d}\}/N\,P_{\text{T}}(\theta)} \cdot N\,P_{\text{T}}(\theta)}{(N\,P_{\text{T}}(\theta) - 1)^2} \|\mathbf{Q}_{N,\text{T}}(\theta)\| \\
&\lesssim \sqrt{\frac{\max\{d, \widetilde{d}\}}{NP_{\text{T}}(\theta)}} \cdot \left\| \frac{\mathbf{Q}_{N,\text{T}}(\theta)}{N\,P_{\text{T}}(\theta)} \right\| \\
&\lesssim^{(\dagger\dagger\dagger)} \sqrt{\frac{\max\{d, \widetilde{d}\}}{NP_{\text{T}}(\theta)}}.
\end{aligned}$$

In $(\dagger)$, we used Eq. (61), which implies that $0.5\,N\,P_{\text{T}}(\theta) \leq n_{\text{sel,T}}(\theta) \leq 1.5\,N\,P_{\text{T}}(\theta)$ when $NP_{\text{T}}(\theta) \gtrsim \max\{d, \widetilde{d}\}$ (which is indeed true, since in Step 5 we set $N = n/2$ & $n \gtrsim \max\{d, \widetilde{d}\}$ is assumed in Theorem 1, and in Step 4.3 we ensure that $P_{\text{T}}(\theta) \gtrsim 1$). In $(\dagger\dagger)$, we again used Eq. (61) directly. In $(\dagger\dagger\dagger)$, we used that $\|\mathbf{Q}_{N,\text{T}}(\theta)\|$ grows on the order of $NP_{\text{T}}(\theta)$ (since it is the sum of $n_{\text{sel,T}}(\theta)$ i.i.d. outer products each with bounded expectation). Thus, overall, replacing the random $n_{\text{sel,T}}(\theta)$ by $NP_{\text{T}}(\theta)$ in the normalization incurs an error of order $\sqrt{\max\{d, \widetilde{d}\}/NP_{\text{T}}(\theta)}$ with high probability. In the subsequent analysis, we may therefore work with the fixed denominator $NP_{\text{T}}(\theta)$ for convenience.

**Step 4.1.2.** The centered vs un-centered version: We have that

$$\frac{1}{N\,P_{\text{T}}(\theta) - 1} \sum_{i \in I_{\text{sel,T}}(\theta)} (x_i - \overline{x}(\theta)) \left( \widetilde{x}_i - \overline{\widetilde{x}}(\theta) \right)^{\top} =$$

$$\frac{1}{N\,P_{\text{T}}(\theta)} \sum_{i \in I_{\text{sel,T}}(\theta)} x_i \widetilde{x}_i^{\top} - \frac{1}{N\,P_{\text{T}}(\theta)\,(N\,P_{\text{T}}(\theta) - 1)} \sum_{i \in I_{\text{sel,T}}(\theta)} \sum_{\substack{j \in I_{\text{sel,T}}(\theta) \\ j \neq i}} x_i \widetilde{x}_j^{\top}.$$

The second term on the right hand side concentrates to $\mathbb{E}\left[ x\widetilde{y}^{\top} \mid x^{\top}\mathbf{M}_{\text{T}}\,\widetilde{x} > \theta, y^{\top}\mathbf{M}_{\text{T}}\,\widetilde{y} > \theta \right]$, where $(x, \widetilde{x})$ and $(y, \widetilde{y})$ are i.i.d. from the joint mixture distribution. This expectation is *zero*, which we formally characterize in Lemmas 6 and 7. The rate of concentration is $\widetilde{O}\left( \frac{1}{N\,P_{\text{T}}(\theta)} \right)$, due to averaging over $(N\,P_{\text{T}}(\theta))^2$ terms, and is hence a higher order term.

**Step 4.1.3.** Analysis of the fixed-denominator un-centered version: The selected samples satisfy the property of being *i.i.d from the conditional law* of the selection rule. In particular, for each $i \in I_{\text{sel,T}}(\theta)$ the matrix $\mathbf{X}_i := x_i \widetilde{x}_i^{\top}$ has expectation $\mathbb{E}[\mathbf{X}_i] = \mathbf{S}_{\text{T}}(\theta)$ and these matrices $\{\mathbf{X}_i : i \in I_{\text{sel,T}}(\theta)\}$ are independent. Using a Matrix-Bernstein concentration result (Eqs. (27) and (28)), it follows that with

probability $1 - \exp(-\Omega(\max\{d, \widetilde{d}\}))$:

$$\left\| \frac{1}{N\, P_{\mathrm{T}}(\theta)} \sum_{i \in I_{\mathrm{sel,T}}(\theta)} x_i \widetilde{x}_i^\top - \mathbf{S}_{\mathrm{T}}(\theta) \right\| \lesssim \sqrt{\frac{\max\{d, \widetilde{d}\}\, \mathrm{poly}(\gamma^{-1}, \widetilde{\gamma}^{-1})}{N\, P_{\mathrm{T}}(\theta)}} \;.$$

**Step 4.2.** Error between teacher and oracle: We show that $\|\mathbf{S}_{\mathrm{T}}(\theta) - \mathbf{S}_{\mathrm{O}}(\theta)\|$ scales proportionally to $\|\mathbf{M}_{\mathrm{T}} - \mathbf{M}_{\mathrm{O}}\|$, and the latter is precisely bounded by Eq. (45). To show this, we first simplify the conditional expectation in $\mathbf{S}_{\mathrm{O}}(\theta), \mathbf{S}_{\mathrm{T}}(\theta)$, define $\mathbf{E}_{\mathrm{O}}(\theta), \mathbf{E}_{\mathrm{T}}(\theta)$ as:

$$\mathbf{E}_{\mathrm{O}}(\theta) := \mathbb{E}\left[ x\widetilde{x}^\top\, \mathbb{I}(x^\top \mathbf{M}_{\mathrm{O}} \widetilde{x} > \theta) \right] \iff \mathbf{S}_{\mathrm{O}}(\theta) = \mathbf{E}_{\mathrm{O}}(\theta)/P_{\mathrm{O}}(\theta) \;; \tag{62}$$

$$\mathbf{E}_{\mathrm{T}}(\theta) := \mathbb{E}\left[ x\widetilde{x}^\top\, \mathbb{I}(x^\top \mathbf{M}_{\mathrm{T}} \widetilde{x} > \theta) \right] \iff \mathbf{S}_{\mathrm{T}}(\theta) = \mathbf{E}_{\mathrm{T}}(\theta)/P_{\mathrm{T}}(\theta) \;. \tag{63}$$

where $\mathbb{I}(.)$ denotes the indicator. Let $\Delta\mathbf{E}(\theta) := \mathbf{E}_{\mathrm{T}}(\theta) - \mathbf{E}_{\mathrm{O}}(\theta)$ and $\Delta P(\theta) := P_{\mathrm{T}}(\theta) - P_{\mathrm{O}}(\theta)$. Also define $\Delta\mathbb{I}(\theta; x, \widetilde{x}) := \mathbb{I}(x^\top \mathbf{M}_{\mathrm{T}} \widetilde{x} > \theta) - \mathbb{I}(x^\top \mathbf{M}_{\mathrm{O}} \widetilde{x} > \theta)$. Then, we write

$$\mathbf{S}_{\mathrm{T}}(\theta) - \mathbf{S}_{\mathrm{O}}(\theta) = \frac{\mathbf{E}_{\mathrm{T}}(\theta)}{P_{\mathrm{T}}(\theta)} - \frac{\mathbf{E}_{\mathrm{O}}(\theta)}{P_{\mathrm{O}}(\theta)}$$

$$= \frac{(\mathbf{E}_{\mathrm{O}}(\theta) + \Delta\mathbf{E}(\theta))\, P_{\mathrm{O}}(\theta) - \mathbf{E}_{\mathrm{O}}(\theta)\, (P_{\mathrm{O}}(\theta) + \Delta P(\theta))}{P_{\mathrm{T}}(\theta)\, P_{\mathrm{O}}(\theta)} = \frac{\Delta\mathbf{E}(\theta)}{P_{\mathrm{T}}(\theta)} - \frac{\Delta P(\theta)}{P_{\mathrm{T}}(\theta)} \cdot \underbrace{\frac{\mathbf{E}_{\mathrm{O}}(\theta)}{P_{\mathrm{O}}(\theta)}}_{\mathbf{S}_{\mathrm{O}}(\theta)} \;.$$

$$\implies \|\mathbf{S}_{\mathrm{T}}(\theta) - \mathbf{S}_{\mathrm{O}}(\theta)\|_2 \leq \frac{1}{P_{\mathrm{T}}(\theta)} \left( \|\Delta\mathbf{E}(\theta)\|_2 + |\Delta P(\theta)| \cdot \|\mathbf{S}_{\mathrm{O}}(\theta)\|_2 \right) \;.$$

We will now bound $\|\Delta\mathbf{E}(\theta)\|_2$ and $|\Delta P(\theta)|$ in terms of $\|\mathbf{M}_{\mathrm{T}} - \mathbf{M}_{\mathrm{O}}\|_2$. Recall that $(x, \widetilde{x})$ follow the mixture distribution (Remark A.1). Decomposing the expectations and probabilities into respective mixtures, we get

$$\Delta\mathbf{E}(\theta) = \eta\, \mathbb{E}_{(x, \widetilde{x}) \sim \mathcal{N}(0, \Sigma_1)} \left[ x\widetilde{x}^\top \Delta\mathbb{I}(\theta; x, \widetilde{x}) \right] + (1 - \eta)\, \mathbb{E}_{(x, \widetilde{x}) \sim \mathcal{N}(0, \Sigma_0)} \left[ x\widetilde{x}^\top \Delta\mathbb{I}(\theta; x, \widetilde{x}) \right] \;,$$

$$\Delta P(\theta) = \eta\, \mathbb{E}_{(x, \widetilde{x}) \sim \mathcal{N}(0, \Sigma_1)} \left[ \Delta\mathbb{I}(\theta; x, \widetilde{x}) \right] + (1 - \eta)\, \mathbb{E}_{(x, \widetilde{x}) \sim \mathcal{N}(0, \Sigma_0)} \left[ \Delta\mathbb{I}(\theta; x, \widetilde{x}) \right] \;.$$

From the above, since both $\eta, 1 - \eta$ are smaller than 1, we get that

$$\|\Delta\mathbf{E}(\theta)\|_2 \leq \|\Delta\mathbf{E}_1(\theta)\|_2 + \|\Delta\mathbf{E}_0(\theta)\|_2 \;, |\Delta P(\theta)| \leq |\Delta P_1(\theta)| + |\Delta P_0(\theta)| \;,$$

where the subscripts $1, 0$ denote the fully clean, corrupted cases respectively (i.e. $\eta = 1, \eta = 0$ respectively). Lemma 10 captures the general form of this, and we invoke this lemma on both the clean data (with covariance $\Sigma_1$) and the noisy data (with covariance $\Sigma_0$). Note that $\mathrm{rank}(\mathbf{M}_{\mathrm{O}}) \geq 2$ is satisfied since $\mathrm{rank}(\mathbf{M}_{\mathrm{O}}) = r$ and we assumed $r \geq 2$ in the statement of Theorem 1. Further, the condition of $\|\mathbf{M}_{\mathrm{T}} - \mathbf{M}_{\mathrm{O}}\| < \sigma_r(\mathbf{M}_{\mathrm{O}})$ is satisfied due to $n \gtrsim (1/\eta^2) \max\{d, \widetilde{d}\}\, (1 + \gamma^{-1})\, (1 + \widetilde{\gamma}^{-1})$, since $\mathbf{M}_{\mathrm{O}}$ has $r$ non-zero singular values all equal to $\eta/\rho_{\mathrm{T}}$ and Eq. (45) with the condition on $n$ implies that $\|\mathbf{M}_{\mathrm{T}} - \mathbf{M}_{\mathrm{O}}\| \lesssim \eta/\rho_{\mathrm{T}}$ (note that implicitly the condition also ensures that the contribution of the $\widetilde{O}(1/n)$ term is bounded). The appropriate constants inside the $\gtrsim$ notation will ensure the required condition. Overall, we get

$$\|\mathbf{S}_{\mathrm{T}}(\theta) - \mathbf{S}_{\mathrm{O}}(\theta)\|_2 \lesssim \frac{1 + \|\mathbf{S}_{\mathrm{O}}(\theta)\|_2}{P_{\mathrm{T}}(\theta)} \|\mathbf{M}_{\mathrm{T}} - \mathbf{M}_{\mathrm{O}}\|_2 \;. \tag{64}$$

**Step 4.3.** Analysis of $P_{\mathrm{T}}(\theta)$ and $P_{\mathrm{O}}(\theta)$: In this part, we show that both $P_{\mathrm{T}}(\theta)$ and $P_{\mathrm{O}}(\theta)$ can be lower bounded by an absolute constant (say, $1/10$) for the relevant regime of filtering threshold $\theta$.

*Argument for $P_{\mathrm{T}}(\theta)$:* Using Step 4.2, we have $P_{\mathrm{T}}(\theta) \geq P_{\mathrm{O}}(\theta) - |\Delta P(\theta)|$, and the deviation is small since $|\Delta P(\theta)| \lesssim \|\mathbf{M}_{\mathrm{T}} - \mathbf{M}_{\mathrm{O}}\|$. Using Eq. (45), we note that a large $\rho_{\mathrm{T}}$ can make $\|\mathbf{M}_{\mathrm{T}} - \mathbf{M}_{\mathrm{O}}\|$ arbitrarily small. Indeed in Step 5, we will set $\rho_{\mathrm{T}}$ to a large value. Since the deviation is small, we can use, for instance, $P_{\mathrm{T}}(\theta) \geq (1/2)P_{\mathrm{O}}(\theta)$. Hence, arguing $P_{\mathrm{O}}(\theta)$ is large suffices, which we do below.

*Argument for $P_{\mathrm{O}}(\theta)$:* Next, we show that $P_{\mathrm{O}}(\theta)$ is 'large enough' for the choices of $\theta \in \{0, r\eta/2\rho_{\mathrm{T}}\}$, and we will use these fixed points in Step 5. Recall from Section 6.2, due to the mixture distribution, the below holds. Here we have accounted for the scaling factor in the definition of $\mathbf{M}_{\mathrm{O}}$.

$$P_{\mathrm{O}}(\theta) = \eta\, P_1\left( \frac{\theta\rho_{\mathrm{T}}}{\eta} \right) + (1 - \eta)\, P_0\left( \frac{\theta\rho_{\mathrm{T}}}{\eta} \right) \;. \tag{65}$$

In Step 5, we will consider the fixed points $\theta \in \{0, {}^{r\eta}/2\rho_{\mathrm{T}}\}$, and so we need lower bounds on $P_0(0), P_0(r/2)$ and $P_1(0), P_1(r/2)$. We state them below:

$$P_0(0) \geq 0.5 , \quad P_1(0) \geq c , \tag{66}$$
$$P_0(r/2) \geq 0 , \quad P_1(r/2) \geq c , \tag{67}$$

where $c > 0$ is an absolute constant. For $P_0(.)$, we have lower bounds $0.5$ (due to symmetry) and $0$ (trivially). For $P_1(.)$, we simply invoke the observation that both $\{0, r/2\}$ are below the mean of the distribution (refer to Figure 2a), and so an appropriate constant $c$ exists satisfying the above. Overall, we conclude that $P_{\mathrm{O}}(0) = \Omega(1)$ and $P_{\mathrm{O}}({}^{r\eta}/2\rho_{\mathrm{T}}) = \Omega(\eta)$.

**Step 5.** Final guarantee via application of Lemma 2: Using Eqs. (60) and (64) in Eq. (52) with Eq. (51), and combining the guarantee from Eq. (45), with probability $1 - \exp(-\Omega(\max\{d, \widetilde{d}\}))$:

$$\left\| \mathbf{G}^\top \widetilde{\mathbf{G}} - \frac{1}{\rho} \mathbf{S}_{\mathrm{O}}(\theta) \right\| \lesssim \frac{1}{\rho} \left( \sqrt{\frac{\max\{d, \widetilde{d}\} \, \mathrm{poly}(\gamma^{-1}, \widetilde{\gamma}^{-1})}{N \, P_{\mathrm{T}}(\theta)}} + \widetilde{O}\left( \frac{1}{N \, P_{\mathrm{T}}(\theta)} \right) \right)$$
$$+ \frac{1}{\rho \, \rho_{\mathrm{T}}} \left( \frac{1 + \|\mathbf{S}_{\mathrm{O}}(\theta)\|_2}{P_{\mathrm{T}}(\theta)} \right) \left( \sqrt{\frac{\max\{d, \widetilde{d}\} \, (1 + \gamma^{-1}) \, (1 + \widetilde{\gamma}^{-1})}{n_{\mathrm{T}}}} + \widetilde{O}\left( \frac{1}{n_{\mathrm{T}}} \right) \right) .$$

We set $n_{\mathrm{T}} = n/2$, and so $N = n - n_{\mathrm{T}} = n/2$ (as in Algorithm 1). For $\rho_{\mathrm{T}}$, we note that it can be chosen arbitrarily large to reduce the second term in the error above. This is because any $\rho_{\mathrm{T}} > 0$ will allow the teacher parameters $\mathbf{G}_{\mathrm{T}}, \widetilde{\mathbf{G}}_{\mathrm{T}}$ to recover the subspace spanned by $\mathbf{U}, \widetilde{\mathbf{U}}$ respectively, but a large choice of $\rho_{\mathrm{T}}$ will make the operator norm small. This does not cause the filtering to change, since the threshold $\theta$ changes multiplicatively with $\rho_{\mathrm{T}}$ (effectively scaling the picture in Figure 2).

The condition of $n \gtrsim \frac{1}{\eta^2} \max\{d, \widetilde{d}\}(1 + \gamma^{-1})(1 + \widetilde{\gamma}^{-1})$ is inherited from Corollary 1 (to be able to use eq (45)). The additional condition on $n$, from the application of Lemma 2 to the above equation (similar to Eq. (33)), results in a larger factor than $1/\eta^2$, hence is already satisfied.

Now we apply Lemma 2 on the above equation, and follow the argument similar to step 5 in Section D. An additional factor of $\sqrt{r}$ appears due to the norm being the chordal distance (frobenius norm). Using Eq. (56) and Eq. (65), we get that with probability $1 - \exp(-\Omega(\max\{d, \widetilde{d}\}))$, the error $\mathrm{ERR}\left(\mathbf{G}, \widetilde{\mathbf{G}}\right)$ is upper bounded (up to constants) by:

$$\underbrace{\frac{1}{[\eta f_1(\theta) + (1 - \eta) f_0(\theta)]}}_{\text{from } \|\mathbf{S}_{\mathrm{O}}(\theta)\|} \underbrace{\frac{1}{\sqrt{\eta \, P_1\left(\theta \rho_{\mathrm{T}}/\eta\right) + (1 - \eta) \, P_0\left(\theta \rho_{\mathrm{T}}/\eta\right)}}}_{\text{from } \sqrt{P_{\mathrm{T}}(\theta)}} \sqrt{\frac{r \, \max\{d, \widetilde{d}\} \, \mathrm{poly}(\gamma^{-1}, \widetilde{\gamma}^{-1})}{n}} .$$

Finally, we plug in the values $\theta \in \{0, {}^{\eta r}/2\rho_{\mathrm{T}}\}$ to recover the terms $T_0, T_{0.5}$ as stated in Theorem 1. Using Eq. (57) and (66), the scaling term of the error above becomes

$$\frac{1}{[\eta + (1 - \eta) \, ({}^2/\pi r)] \cdot \sqrt{\eta \, c + (1 - \eta) \, ({}^1/2)}} \lesssim r \quad \text{for any } \eta \in (0, 1] .$$

Using Eq. (58) and (67), the scaling term of the error above becomes

$$\frac{1}{[\eta + (1 - \eta) \, ({}^1/2)] \cdot \sqrt{\eta \, c}} \lesssim \frac{1}{\sqrt{\eta}} .$$

The above describes both regimes of behavior, and why an extra factor of $r$ appears in the term $T_0$, compared to the term $T_{0.5}$, in Theorem 1. This concludes the argument.

## H   Discussion on robustness of the choice of filtering threshold

We note that the error achieved by teacher-based filtering can be fairly robust to the choice of $\theta$, the filtering threshold. Our synthetic experiment in Figure 3a was conducted with a fixed, untuned threshold of $\theta = 0$. Further, we conduct an experiment measuring the sensitivity of the final error with respect to the choice of $\theta$. In the setting of Figure 3a with $n = 10000$ samples, we fix $\eta = 0.3$

(in-line with the empirically observed clean fraction in CLIP data [11]) and (implicitly) vary the filtering threshold $\theta$ of the teacher-based filtering (by explicitly varying the fraction of data retained in the filtering step). The below table shows that the error of teacher-based filtering is relatively flat for values of $\theta$ in the vicinity of the optimal threshold $\theta^*$. An analogous experiment on real data [11, Figure 2] makes a similar observation.

| Fraction of data retained | Mean error ($\pm 1\,\sigma$) ($\times 10^{-4}$) |
|:---:|:---:|
| 1% | $28.76 \pm 4.00$ |
| 10% | $11.79 \pm 1.20$ |
| 20% | $9.85 \pm 1.39$ |
| **30%** | $9.08 \pm 1.15$ |
| 40% | $8.97 \pm 1.09$ |
| 50% | $8.71 \pm 1.05$ |
| 100% | $16.51 \pm 2.03$ |

Table 1: Mean error vs. fraction of data retained.

# I   Discussion on the potential of robust statistics for the analysis of filtering

An initial instinct based on Figure 2 is to use ideas from robust statistics. As discussed in Remark 6.2, we can expect $\mathcal{D}_0$ and $\mathcal{D}_1$ to be well-separated, which means there will exist some $\theta \in \mathbb{R}$ (a reasonable guess is $\theta \approx r/2$) such that the selected data is mostly clean. After filtering, the picture resembles the robust statistics setting: an $\alpha$ corruption on the clean distribution for some small $\alpha$. This is a reasonable approach overall, but has two shortcomings. *First*, this approach will *not* achieve zero error as $n \to \infty$. We are shooting for $f(\eta) \cdot 1/\sqrt{n}$ which is better than $1/\sqrt{n} + g(\eta)$, since the latter is non-zero even when $n \to \infty$. This approach will end up getting the latter. This is because the canonical rate in robust statistics is $\sqrt{d/n_{\text{sel}}} + \alpha$. Under filtering, $n_{\text{sel}}$ and $\alpha$ are functions of $\theta$. One can determine the optimal $\theta$ to balance the tradeoff, but to get a final rate of the form $f(\eta) \cdot 1/\sqrt{n}$, this will require some *conditions* on $n, \eta$ (possibly $\eta$ bigger than a threshold, and $n$ smaller than a threshold). Since our case has stochastic corruption which is weaker than adversarial corruption, we can expect to prove something for all $n$ and all $\eta$. *Second*, this approach performs a "reductive" operation of treating data as only clean v/s corrupted, and assuming the corrupted part provides no signal. This is a closely linked argument to the first one above. The crucial observation is that the right tail of the corrupted data (i.e. $\mathcal{D}_0$ in Figure 2) actually provides 'close to clean' samples. This is because these just happened to be samples such that the $z, \widetilde{z}$ – albeit independently sampled in a high-dimensional space – happened to have a high inner product (small angle). Our adopted approach, based on the conditional properties of the Gaussian distribution, formalizes this intuition that the right tail of $\mathcal{D}_0$ also provides signal.

