# OpenReview forum: "Understanding the Gain from Data Filtering in Multimodal Contrastive Learning"
_NeurIPS.cc/2025/Conference — NeurIPS 2025 poster_

### Official Review · Reviewer_M3vJ · 2025-06-26

**Clarity:** 3
**Significance:** 3
**Originality:** 2
**Rating:** 4
**Confidence:** 3

**Summary:**

This paper provides a theoretical analysis of teacher-based data filtering in the context of multimodal contrastive learning. It builds on prior linear models of vision-language representation learning and studies how data filtering (particularly using a trained model to score and retain high-quality samples) affects learning performance under a stochastic corruption model.

**Questions:**

- how limiting is the linear assumption? or in other words, how far from real-world experiments are we when using the linear assumption? I understand the necessity/simplicity of using a linear assumption. Nevertheless, I would be curious to hear the authors' opinion on this.
- is filtering mainly related to the multimodal pairing or also on noisy samples in general? does it make a difference at all to distinguish between the two cases?

**Ethical Concerns:**

["NO or VERY MINOR ethics concerns only"]

**Final Justification:**

Accept

**Limitations:**

yes

**Paper Formatting Concerns:**

----

**Quality:**

3

**Strengths And Weaknesses:**

### Strengths

- Theoretical novelty: The result that filtering can outperform an “oracle” that uses only clean samples is surprising and nontrivial. The analysis highlights the interaction between stochastic corruption and contrastive inner product objectives.
- Relevance to practice: Teacher-based filtering is increasingly used in real-world large-scale multimodal pretraining (e.g., CLIP, DataComp), yet lacked formal justification.
- Clarity of analysis: The paper is well-structured and first builds intuition before going into formal details. The progression from baseline to teacher-based methods is systematic.


### Weaknesses

- Restrictive assumptions: The analysis relies on linear encoders, Gaussian noise, and exact knowledge of latent dimension $r$. These assumptions simplify analysis but limit applicability to realistic settings involving deep networks and complex noise.
- Limited empirical validation: While a synthetic experiment is provided, real-world empirical evaluation is absent. The paper refers to past work (e.g., Fang et al.) but doesn’t test its theory in new real-world settings.
- Teacher training is idealized: The teacher is assumed to generalize well in the corrupted regime, which may not hold in practice (especially when $\eta$ is small). While this is acknowledged, more discussion on when this assumption breaks would help.

---

> ### Author Rebuttal · Authors · 2025-07-31
>
> We thank the reviewer for their detailed and constructive feedback. We appreciate the positive comments about the theoretical novelty of the analysis, and the practical relevance of the problem of data filtering. In the following, we address the concerns raised.
>
> # Weaknesses
>
> ### (1) Restrictive assumptions.
>
> **Linearity + Gaussianity:** The goal of our work is to provide theoretical justification for the common empirical observation that data filtering using a teacher CLIP model improves performance. The setup of Linearity and Gaussian noise provides a **tractable setting** to study this phenomenon, and is motivated by **past works** that make the same assumptions [6, 7]. Beyond CLIP filtering, **other phenomena in deep learning** have also been successfully characterized in the linear setup, e.g., benign overfitting [8] (also known as double descent), scaling laws [11], and implicit bias [14, 15]. Overall, the linear assumption, though not directly applicable in some real cases, provides a useful abstraction that often **preserves the primary problem characteristics** and helps to build a deeper understanding through a theoretical lens. We believe our work (and the assumptions we have), although simple, is within the typical assumptions made in theoretical anlayses of deep learning, which has proven to be productive and insightful (e.g., [8, 11, 14, 15]).
>
> **Latent dimension $r$:** In practice, the latent dimension $r$ (i.e., the embedding space dimension) is typically a design choice and is therefore known at training time. Theoretically, assuming the underlying $r$ is known allows us to **isolate the effects of data filtering from the separate, well-studied problem of subspace rank estimation** (for e.g., in [12]). Thanks for pointing this out, we think it's useful to add this comment in Line 150.
>
> ### (2) Limited empirical validation.
>
> The teacher-based filtering (Figure 1c, Algorithm 1) has been relatively **well-studied empirically**, for instance in [1, 2, 3]. The focus of this paper is to theoretically analyze this approach, and demonstrate a provable benefit. Since this involved **no new algorithmic modifications**, we relied on synthetic experiments (Figure 3a) and reused a real-data experiment (Figure 3b), instead of a suite of new experiments.
>
> That said, we plan to run small-scale CLIP training experiments (equivalent to the `small` scale in DataComp [1]) with a focus on the data quality, particularly focusing on the misalignment in image-text pairs. The main idea of this experiment is to have two data sources (one *clean* and one *noisy*), similar to Figure 3(b) replicated from [2]. We can sample data with $\eta$, $1-\eta$ proportions to create the training dataset. From this, we can observe the variation in the learnt model's performance with varying $\eta$.
>
> ### (3) Teacher training is idealized. What happens when $\eta$ is small?
>
> **Meta comment:** We are somewhat unclear about what this question asks. Below we provide a response, but please clarify the question further if this does not answer it. In particular, please also point out which line number you are referring to when you say "while this is acknowledged".
>
> **Response:** The modeling assumptions result in the teacher achieving error $1/\eta \sqrt{n}$ for $\eta > \eta_0$, as shown in Corollary 1 for an appropriate $\eta_0$. We would like to stress the condition of $\eta > \eta_0$, as this shows that even the current simplistic setup **captures that $\eta$ needs to be large enough for the teacher to learn something meaningful** (the rightmost points in Figure 3(a) validate this empirically, via the growing error bars and deviation from the theoretical trend). That said, you are right to believe the threshold $\eta_0$ in practice will be much higher than the one predicted by the Corollary 1. We believe the below two factors will play a crucial role in this:
>
> - Introduction of non-linearity, since parameter estimation becomes statistically harder with non-linear function classes.
> - The model in Section 3.1 only assumes a binary distinction of clean v/s noisy to model data quality. Real datasets display a broader range of data quality across samples.
>
> We can include a discussion about this in the revised manuscript. However, a detailed study would involve a significant modification to the setup in Section 3.1, and falls outside the scope of the current work.
>
> # Questions
>
> ### (1) Comment on the limitations of the linear assumption.
>
> Linear models have been surprisingly successful in explaining important empirical phenomena in highly nonlinear deep neural networks. This includes benign overfitting (also known as double descent) [8], scaling laws [11], and implicit bias [14, 15]. Closest to our problem might be [9], where the authors analyze linear regression to demonstrate the gain of data filtering. For a detailed justification of using linear models to study deep neural networks, we refer to a nice lecture note [16] by Andrea Montanari at Stanford.
>
> Despite these successes, linear models have notable limitations. They assume that learning operates in a regime where the deep neural network can be well approximated by its first-order Taylor expansion. It's crucial to understand that training within this regime is significantly less powerful than fully training neural networks. While it's possible to train neural networks in a way that makes them amenable to this first-order approximation, this often occurs as a side effect of very specific parametrization choices.
>
> This implies that while we can restrict hyperparameter choices to make linear models more accurate at approximating actual deep neural network training dynamics, doing so would likely compromise the practical performance of the resulting models. Although the high-level insights derived from linear modeling have proven surprisingly successful, the specific theorems themselves may not accurately predict the behavior of nonlinear model training. Therefore, the lessons learned from linear models should generally be interpreted qualitatively, serving more as a justification or general guidance rather than providing immediate predictive insights.
>
> In this paper, we observed the same thing with respect to the real data experiment in Figure 3(b). As indicated in lines 73-74, the qualitative message of the improved $\eta$ dependence, derived from the linear modeling theory, holds true with nonlinear models as well.
>
>
> ### (2) Discussion of data filtering beyond unaligned pairings.
>
> This is an insightful question, thank you for raising this! Indeed, the community is increasingly observing the utility of data filtering in various settings. We will answer this in two parts:
>
> 1. **Data filtering for multimodal data beyond unaligned pairings.**
>
> Indeed, beyond unaligned pairings, data filtering is also used to remove other kinds of noisy data (i.e., simply bad image or text quality). For example, empirically, the creation of large scale datasets (e.g. LAION-5B, DataComp-1B) involved various heuristic-based filters to remove data with 'bad' samples in individual modalities.
>
> Theoretically, we believe it does make sense to distinguish between these two cases. A better model to study the latter would try to capture that a significant portion of the marginal distributions $D_x, D_{\widetilde{x}}$ are individually noisy beyond the faulty pairings. The generative model described in Section 3.1 does not capture this due to a common prior $D_z$ for both modalities.
>
> Overall, this would be akin to studying the $\eta = 1$ case in the new setup described above, and asking, does filtering still help? This is a very valid direction of future work, but formulating the right mathematical assumptions and the subsequent analysis falls outside the current scope.
>
> Note that our main result -- improving the error dependence from $1/\eta$ to $1/\sqrt{\eta}$ (or better) -- is a statement about solving the pairing problem. This **gain cannot be achieved by simply addressing per-modality noise**. Hence, it is needed to distinguish this case with others studied in the past. Of course, the practical reality is a mixture of these two cases (unaligned pairs + noisy samples in individual modalities). In this work, we focus on the alignment problem only.
>
> 2. **Data filtering for general data.**
>
> Beyond multimodal data, data filtering has also been studied in the usual statistical framework, assuming access to $n$ samples (not necessarily multimodal) [9, 10]. Again, it is worth noting that none of these frameworks capture the pairing problem. Further, these works also largely study the linear case for their theoretical contributions.
>
> # References
>
> [1] DataComp: In search of the next generation of multimodal datasets, arxiv:2304.14108.
>
> [2] Data Filtering Networks, arxiv:2309.17425.
>
> [3] CLIPLoss and Norm-Based Data Selection Methods for Multimodal Contrastive Learning, arxiv:2405.19547.
>
> [4] (unused reference)
>
> [5] (unused reference)
>
> [6] Understanding Multimodal Contrastive Learning and Incorporating Unpaired Data, arxiv:2302.06232.
>
> [7] The Power of Contrast for Feature Learning: A Theoretical Analysis, arxiv:2110.02473.
>
> [8] Benign Overfitting in Linear Regression, arxiv:1906.11300.
>
> [9] Towards a statistical theory of data selection under weak supervision, arxiv:2309.14563.
>
> [10] Iterative Least Trimmed Squares for Mixed Linear Regression, arxiv:1902.03653.
>
> [11] Scaling Laws in Linear Regression: Compute, Parameters, and Data, arxiv:2406.08466.
>
> [12] Optimal Estimation and Rank Detection for Sparse Spiked Covariance Matrices, arxiv:1305.3235.
>
> [13] (unused reference)
>
> [14] The Implicit Bias of Gradient Descent on Separable Data, arXiv:1710.10345.
>
> [15] In Search of the Real Inductive Bias: On the Role of Implicit Regularization in Deep Learning, arXiv:1412.6614.
>
> [16] Six Lectures on Linearized Neural Networks, arXiv:2308.13431.

---

> > ### Comment · Reviewer_M3vJ · 2025-08-04
> >
> > I would like to thank the authors for the detailed reply and explanations.
> > I raised my score, accordingly.

---

> ### Author Response · Authors · 2025-08-07
>
> Thank you and please let us know if there are any more questions!

---

### Official Review · Reviewer_qb7P · 2025-07-01

**Clarity:** 3
**Significance:** 3
**Originality:** 3
**Rating:** 4
**Confidence:** 2

**Summary:**

This paper derives **bounds for the error in the contrastive learning** setup when the **data is filtered using a teacher** that is trained from the same data.

The contrastive learning setup is **simplified** to make it amenable to mathematical analysis: the data pairs are assumed to be low rank signals (i.e. underlying vector $z$ with dimensionality $r$ is projected to dimensionality $d$ where signal $x$ is observed) to which zero mean isotropic Gaussian noise is added. With probability $\eta$ the observed pairs are projected from the same $z$, and with probability $1-\eta$ the pairs are sampled independently (in both cases sampled from standard multivariate Gaussian). The contrastive training process is modeled as learning  the projection that maps $x$ back to $z$. The data filtering is modeled as first learning a linear projection on half of the data, and the doing threshold-based filtering of the pairs after applying the learnt mapping. The error finally is defined as the chordal distance between the subspaces of the learned projection and the generating projection.

In this simplified setup, the authors show that the error is bounded by $\eta^{-1}n^{-0.5}$ in the unfiltered setup, and by $\eta^{-0.5}n^{-0.5}$ and $n^{-0.5}$ in the filtered setup (when $\eta$ is large and when $\eta$ is small). Note that the rate $n^{-0.5}$ is even better than the rate achieved when training on data that is filtered by an oracle (which is also $\eta^{-0.5}n^{-0.5}$).

**Questions:**

1. What would the "Slope = 1" and "Slope = 0.5" lines look like on FIgure 3b? Is it possible to translate the error bounds as used in this paper (chordal distances of subspaces) to error rates that can be measured in a more realistic setup? Maybe it would be insightful to only remove some of the simplifications but keep others (e.g. generated data with more realistic non-linear training setup) to see which of the findings hold true?

1. The paper argues that in the small $\eta$ regime, teacher-based filtering has a better bound than oracle-based filtering. The authors explain in appendix H why this is the case. Under which circumstances could something similar be observed in a more realistic and practically relevant setup? Are there any insights from the presented analysis that would explain when filtering can be harmful?

1. The introduction (lines 22-23) states "smaller but higher quality subsets of the data have been observed to result in better models". But there are other studies that show that similar models can be trained directly on noisy large scale datasets (e.g. [ALIGN](https://arxiv.org/abs/2102.05918)). The submission argues that filtering with a teacher that is trained on the same data distribution is better than oracle-based filtering. Could it be argued that oracle-based filtering is more similar to the manual filtering that was applied to the CLIP data, while teacher-based filtering could in fact be a mechanism that is more similar to a partly trained network that can learn to ignore noisy data from the same distribution?

Small comments:

1. Legend Figure 2: consider writing $50000$ instead of $50k$ because the latter (with $k$ in italic font) reads like a variable.

1. Lines 62-69, lines 223-234: Maybe also refer to Section H in this context since that section has some additional information about why something can be learned from corrupted samples.

1. Line 541: Even if the code is simple, I would still find it useful to have it published. That would be beneficial to check some implementation details, and useful to build work on top of what is presented in the paper.

**Ethical Concerns:**

["NO or VERY MINOR ethics concerns only"]

**Final Justification:**

I remain with my original rating 4=borderline accept.

The authors have been very responsive in the rebuttal period and given clear answers to all of my questions. Even though I'm not too familiar with the subject matter (theoretical analysis), after the review process I am fairly confident that the presented work is technically sound and well anchored in the existing literature.

The reason for not increasing my original rating is mainly that the **authors have still not entirely convinced me about the practical usability of the presented bounds** – the one weakness that I already mentioned in my original review.
My concerns could have been resolved, if the authors showed how the quantitative results from the paper could be applied to improve practically important decisions like "dataset election" or "valuating data augmentation" (as the authors have mentioned in some of the rebuttals, e.g. the [rebuttal to my review](https://openreview.net/forum?id=2xDdVkWrqq&noteId=TkuFczv3As)).
To clarify this point I have created a new thread about the [Practical Applicability of the Theoretical Findings](https://openreview.net/forum?id=2xDdVkWrqq&noteId=PffKfAm0vr), in which the authors have replied that this kind of practical decisions cannot be improved directly with the theoretical findings from the paper, but they promised to add more results to the paper that tries to bridge the gap partially, which I think would strengthen the paper.

**Limitations:**

yes

**Quality:**

3

**Strengths And Weaknesses:**

Disclaimer: I am a ML practitioner and have experience with data filtering setup with contrastive models, but I am a novice when it comes to their mathematical analysis. I have read all the parts from the main paper and appendix sections A, B, H – but have skipped appendix sections C-G because I could not give good feedback on the mathematical derivations. I hope that my review of the other parts is still useful and look forward to the discussions from the more mathematically oriented reviewers.

**Strengths**

1. The paper studies a **practically relevant problem**: contrastive learning is still very relevant, for example most state-of-the-art vision encoders are trained with a contrastive loss on image/text pairs. Data filtering is often an integral part in this context and it's important to understand its effects on the learning process.

1. The paper is **well structured and clearly written**. The assumptions are clearly outlined, the main mathematical findings are presented in a concise manner that makes it possible to understand how the results were derived, while keeping all of the details for the long derivations in the appendix.

1. The derivations build on top of an existing literature, like [5] and [22], and extend these previous studies by providing new insights into what happens when the data is filtered using a teacher trained on the same data distribution.

**Weaknesses**

1. From a practical perspective, all the **simplifying assumptions** that make the setup amenable to mathematical analysis create a **large gap with a realistic setup** used to train relevant models. It is hard to translate the final results to something other than "data filtering can help in the contrastive learning setup", which is also what is shown qualitatively in Figure 3b. There is a substantial difference between the theoretical results with all the strong simplifications, and a realistic setup like the ones cited in the introduction. Furthermore, the practical setup (Figure 3b) – which does not make reference to the derived bounds anymore but simply shows that performance without filtering decreases more quickly when $\eta$ becomes smaller – operates at extremely low performance (best achieved accuracy with $\eta=1$ is below 30%).

---

> ### Author Rebuttal · Authors · 2025-07-31
>
> We thank the reviewer for their detailed and constructive feedback. We appreciate the positive comments about the relevance of the theoretical analysis, and the practical impact of data filtering. In the following, we address the concerns raised.
>
> # Weaknesses
>
> ### (1) Connection between theory and practice.
>
> In the following, we will make two points to justify our choices and their usefulness.
>
> **1. Why the linear setup analysis is useful?**
>
> The goal of our work is to provide theoretical justification for the common empirical observation that data filtering using a teacher CLIP model improves performance. The setup with Linearity and Gaussian noise provides a **tractable setting** to study this phenomenon, and is motivated by **past works** that make the same assumptions [6, 7]. Beyond CLIP filtering, **other phenomena in deep learning** have also been successfully characterized in the linear setup, e.g., benign overfitting [8] (also known as double descent), scaling laws [11], and implicit bias [14, 15]. These topics are foundations of what we call the Theory of Deep Learning, and are textbook materials in courses taught at top institutions. Overall, the linear assumption, though not directly applicable in some real cases, provides a useful abstraction that often **preserves the primary problem characteristics** and helps to build a deeper understanding through a theoretical lens. We believe our work (and the assumptions we have), although simple, is within the typical assumptions made in theoretical anlayses of deep learning, which has proven to be productive and insightful (e.g., [8, 11, 14, 15]). For a detailed justification of using linear models to study deep neural networks, we refer to a nice lecture note [16] by Andrea Montanari at Stanford.
>
> **2. What are the practical implications?**
>
> Beyond its theoretical contributions, Theorem 1 also offers some useful guidance to practitioners by formalizing the trade-off between data quantity ($n$) and quality (via clean fraction $\eta$). Our analysis allows practitioners to move from qualitative intuition to principled, **quantitative decision-making in common data curation scenarios**. The following two examples illustrate this in detail:
> - **Dataset election:** Consider a practitioner faced with the choice to pick a dataset among many alternatives. Often the choice involves a large, noisy dataset (large $n$, small $\eta$) versus a smaller, curated one (small $n$, large $\eta$). Further, the high-level aggregates (like $n$ and $\eta$ or approximate $\eta$) are usually available, but the actual datasets might be behind a paywall. By plugging these values into the error bounds from Theorem 1 (i.e., comparing which option yields a smaller error), one can make a more informed decision about which dataset is likely to produce a better model.
> - **Valuating Data Augmentation:** When considering whether to invest resources in collecting a small amount of clean data to augment a large, noisy corpus, our theorem provides quantitative guidance. It helps estimate the expected performance gain from the improved $\eta$ and increased $n$, allowing a practitioner to assess if the return on investment is worthwhile.
>
> Crucially, the two distinct error regimes identified in Theorem 1 provide non-obvious insights. The $1/\sqrt{\eta}$ error rate in the low-$\eta$ regime suggests that for very noisy datasets, simply increasing data volume ($n$) can be surprisingly effective, a practical takeaway that is not immediately intuitive.
>
> Lastly, in relation to a point raised by the reviewer, we would like to argue that a 30% zero-shot accuracy on ImageNet is non-trivial, since no explicit classifier was trained on top of the CLIP embeddings (ImageNet has a 1000 classes, so a random guesser would get 0.1% accuracy).
>
> # Questions
>
> ### (1a) About the theoretical metric (chordal distance) v/s real-world metrics.
>
> Since the metric between Fig 3(a) and 3(b) is different (note the different y-axis), the "slope 1" and "slope 0.5" lines would not make sense on the metric used in Fig 3(b). Relatedly, note that the metric used in Figure 3(b) is not a theoretical distance like the chordal distance. Instead, it is `1 - ImageNet Accuracy`, a downstream metric that practitioners care about.
>
> ### (1b) Weakening the assumptions (e.g., removing linearity).
>
> Indeed, weakening the assumptions can be insightful by showing the differences in the observed trends. For example, what would Theorem 1 look like without the linearity assumption? A full analysis of this requires non-trivial analysis that falls outside the scope of this work. That said, linear models have been successfully used to explain many phenomena in modern deep learning, e.g., benign overfitting [8] (also known as double descent), scaling laws [11], and implicit bias [14, 15]. The lessons learned from linear models generally serve as a useful qualitative guide to more complex scenarios. In this paper, we observed the same thing with respect to the real data experiment in Figure 3(b). As indicated in lines 73-74, the qualitative message of the improved dependence, derived from the linear modeling theory, holds true with nonlinear models as well.
>
> ### (2) Can teacher-based filtering be harmful?
>
> This is a great question, thank you for raising this. Indeed, it is a fruitful direction to study how filtering behaves in setups other than the one presented in Section 3.1. One such idea is relaxing Assumption 2 to a general covariance matrix for the noise $\xi$. This exercise indicates that teacher-based filtering can introduce an algorithmic bias. That is, the error with data filtering stays above zero even when $n \rightarrow \infty$ (i.e. a statistically inconsistent estimate). Note that the no-filtering algorithm does not suffer from this.
>
> Future work could study the precise behaviour of this bias, and how to mitigate this potential harmful effect of data filtering. For instance, this might reveal that the standardization of data in the right basis could mitigate this (e.g., normalizing the images in certain ways). This is a very interesting direction, but requires a detailed investigation from both theoretical and empirical angles.
>
> ### (3a) Could it be argued that oracle-based filtering is more similar to the manual filtering that was applied to the CLIP data?
>
> Note that oracle based filtering assumes knowledge of ground-truth parameters (Figure 1b). In this sense, it is hard to compare it with manual filtering, which involves applying heuristics like length of text, quality of image, etc. to filter the dataset. Note that the datasets are huge (billions of samples), meaning that manual filtering can not involve human annotation. In light of this, we don't expect any manual filtering (heuristic-based) to be able to perform like oracle filtering.
>
> ### (3b) Teacher-based filtering could in fact be a mechanism that is more similar to a partly trained network that can learn to ignore noisy data from the same distribution?
>
> This is indeed true. Lines 201-203 posit that the teacher learns a useful signal despite the presence of noisy samples, which help in data filtering. The reason teacher-based can outperform even oracle filtering is the incidental alignment in some noisy samples (Lines 223-225).
>
> # Small comments
>
> Thank you for rasining points 1 and 2, we will include them in the revised manuscript. For 3, we indeed plan to release the code publicly on github!
>
> # References
>
> [1] DataComp: In search of the next generation of multimodal datasets, arxiv:2304.14108.
>
> [2] Data Filtering Networks, arxiv:2309.17425.
>
> [3] CLIPLoss and Norm-Based Data Selection Methods for Multimodal Contrastive Learning, arxiv:2405.19547.
>
> [4] Quality Not Quantity: On the Interaction between Dataset Design and Robustness of CLIP, arxiv:2208.05516.
>
> [5] Improving Multimodal Datasets with Image Captioning, arxiv:2307.10350.
>
> [6] Understanding Multimodal Contrastive Learning and Incorporating Unpaired Data, arxiv:2302.06232.
>
> [7] The Power of Contrast for Feature Learning: A Theoretical Analysis, arxiv:2110.02473.
>
> [8] Benign Overfitting in Linear Regression, arxiv:1906.11300.
>
> [9] Towards a statistical theory of data selection under weak supervision, arxiv:2309.14563.
>
> [10] Iterative Least Trimmed Squares for Mixed Linear Regression, arxiv:1902.03653.
>
> [11] Scaling Laws in Linear Regression: Compute, Parameters, and Data, arxiv:2406.08466.
>
> [12] Optimal Estimation and Rank Detection for Sparse Spiked Covariance Matrices, arxiv:1305.3235.
>
> [13] Going Beyond Nouns With Vision & Language Models Using Synthetic Data, arXiv:2303.17590.
>
> [14] The Implicit Bias of Gradient Descent on Separable Data, arXiv:1710.10345.
>
> [15] In Search of the Real Inductive Bias: On the Role of Implicit Regularization in Deep Learning, arXiv:1412.6614.
>
> [16] Six Lectures on Linearized Neural Networks, arXiv:2308.13431.

---

> > ### Comment · Reviewer_qb7P · 2025-08-06
> >
> > I would like to thank the authors for their interesting and thoughtful rebuttal to my review, which answered most of my questions satisfactorily.
> >
> > I noticed that all reviewers raised questions about the degree to which the theoretical findings can be applied in a practical setting, so I created a new thread [Practical Applicability of the Theoretical Findings](https://openreview.net/forum?id=2xDdVkWrqq&noteId=PffKfAm0vr) for discussion.
> >
> > Other than the points raised in that separate thread, I only have one remaining question with respect to the response **(2) Can teacher-based filtering be harmful?**: The authors mention in their rebuttal that the exercise of relaxing Assumption 2 to a general covariance matrix for the noise $\xi$ indicates that teacher-based filtering can introduce an algorithmic bias. Where exactly can I find this derivation in the manuscript? I think the discussion of this point (which was also mentioned in the rebuttal to reviewer AyoJ), and its practical relevance, might be an interesting addition to the paper.

---

> ### Author Response · Authors · 2025-08-07
>
> Thank you for your response, and we are glad to have answered most of your questions. Thanks also for creating the common thread. We will respond to those questions raised in the thread itself.
>
> Regarding the algorithmic bias, as mentioned in our rebuttal response, this is an interesting direction but requires a detailed investigation from both theoretical and empirical angles. We do not have this in the submitted manuscript. After submission, we worked on the analysis when the noise is not isotropic, and identified that this creates an extra term in the error, that corresponds to a bias. Since this analysis is very similar to the one we have in the appendix in the submission (but with a more general assumption on the noise covariance $\Sigma$), we are planning to add this result in the revision. At the high-level, the error after filtering in Theorem 1 gets the additional bias term (for a general $\theta$), like below:
>
> $$ERR(\mathbf{G}(\theta), \widetilde{\mathbf{G}}(\theta)) \lesssim E(n, \eta, \theta, \Sigma) + B(\eta, \theta, \Sigma).$$
>
> The terms $E$ and $B$ also depend on the dimensions $r, d, \tilde{d}$, but we suppress that for clarity. The bias $B$ importantly does not depend on $n$, but it goes to zero when $(i)$ $\eta=1$, since we only have clean data in that case, $(ii)$ $\theta = -\infty$, since that corresponds to no filtering, and $(iii)$ the noise covariance $\Sigma$ is isotropic.
>
> We agree that this will be an interesting addition to the literature of data filtering, and we will add discussion of this result and its practical relevance in the revision also. However, empirical investigation of this in the realistic setting is outside the scope of this paper. We believe it is an important topic of interest for future research with respect to how such theoretical insights can help improve data filtering in practice.

---

### Official Review · Reviewer_AyoJ · 2025-07-02

**Clarity:** 3
**Significance:** 3
**Originality:** 3
**Rating:** 4
**Confidence:** 3

**Summary:**

This paper explores the advantage of data filtering in improving the performance of contrastive learning in multimodal representation learning. Specifically, the paper provides a theoretical explanation for the empirical success of teacher-based data filtering, which uses a pre-trained model to filter out low-quality data from large-scale multimodal datasets. The authors establish bounds on the performance improvement obtained through data filtering and demonstrate the advantages of this method over traditional contrastive learning without filtering.

**Questions:**

Please check the former section.

**Ethical Concerns:**

["NO or VERY MINOR ethics concerns only"]

**Final Justification:**

After the rebuttal, my major concerns are addressed by the authors, and I keep my positive score.

**Limitations:**

Please check the former section.

**Paper Formatting Concerns:**

no formatting concerns

**Quality:**

3

**Strengths And Weaknesses:**

Strength:

1. The paper provides a detailed theoretical analysis of the benefits of data filtering in contrastive learning.
2. The idea that removing low-quality data pairs improves multimodal contrastive learning is intuitive, and the analysis about it has a great potential.

Weakness:

1. The idea of filtering out low-quality data pairs to improve multimodal contrastive learning is highly intuitive. It's a common approach in many domains, like active learning, where selecting high-quality data is known to improve model performance. While the theoretical analysis is valuable, it doesn’t lead to the development of any new or better algorithms. The potential for algorithmic innovation stemming from the theory is not explored in the paper. How to leverage this theory to design more efficient or scalable algorithms for data filtering in real-world scenarios?
2. The theoretical results provided by the authors are not sufficiently linked to real-world applications. It remains unclear how the theory could directly inform practices in real-world cases. For instance, how would one practically implement a better teacher-based filtering approach?
3. The experiments are conducted only on synthetic data and one real-world dataset. A more diverse set of experiments is needed to validate the theoretical results.

---

> ### Author Rebuttal · Authors · 2025-07-31
>
> We thank the reviewer for their detailed and constructive feedback. We appreciate the positive comment about the comprehensiveness of the theoretical contributions, and also about the suprising finding of the independent-of-$\eta$ regime in Theorem 1. In the following, we address the concerns raised.
>
> ### (1) How to use the theory to get better algorithms?
>
> This is a great question, thank you for raising this. To the best of our knowledge, this is the first work that provably shows a benefit of teacher-based data filtering. We believe this is a significant contribution, which lays the necessary theoretical groundwork that future algorithmic advancements can build upon.
>
> One such idea is relaxing Assumption 2 to a general covariance matrix for the noise $\xi$. This exercise indicated that teacher-based filtering can introduce an algorithmic bias. That is, the error with data filtering stays above zero even when $n \rightarrow \infty$ (i.e. a statistically inconsistent estimate). Future work could study the precise behaviour of this bias, which could in turn lead to algorithmic improvements. For instance, this might reveal that the standardization of data in the right basis could mitigate this (e.g., normalizing the images in certain ways). This is a very interesting direction, but requires a detailed investigation from both theoretical and empirical angles.
>
>
> ### (2) What are the practical implications?
>
> Beyond its theoretical contributions, Theorem 1 also offers some useful guidance to practitioners by formalizing the trade-off between data quantity (i.e. $n$) and quality (via clean fraction $\eta$). This allows practitioners to move from qualitative intuition to **quantitative decision-making in common data curation scenarios**. The following two examples illustrate this in detail:
> - **Dataset election:** Consider a practitioner faced with the choice to pick a dataset among many alternatives. Often the choice involves a large, noisy dataset (large $n$, small $\eta$) versus a smaller, curated one (small $n$, large $\eta$). Further, the high-level aggregates (like $n$ and $\eta$ or approximate $\eta$) are usually available, but the actual datasets might be behind a paywall. By plugging these values into the error bounds from Theorem 1 (i.e., comparing which option yields a smaller error), one can make a more informed decision about which dataset is likely to produce a better model.
> - **Valuating Data Augmentation:** When considering whether to invest resources in collecting a small amount of clean data to augment a large, noisy corpus, our theorem provides quantitative guidance. It helps estimate the expected performance gain from the improved $\eta$ and increased $n$, allowing a practitioner to assess if the return on investment is worthwhile.
>
> Crucially, the two distinct error regimes identified in Theorem 1 provide non-obvious insights. The $1/\sqrt{\eta}$ error rate in the low-$\eta$ regime suggests that for very noisy datasets, simply increasing data volume ($n$) can be surprisingly effective, a practical takeaway that is not immediately intuitive.
>
> ### (3) Limited empirical validation.
>
> The teacher-based filtering (Figure 1c, Algorithm 1) has been relatively well-studied empirically, for instance in [1, 2, 3]. The focus of this paper is to theoretically analyze this approach, and demonstrate a provable benefit. Since this involved no new algorithmic modifications, we relied on synthetic experiments (Figure 3a) and reused a real-data experiment (Figure 3b), instead of a suite of new experiments.
>
> That said, we plan to run small-scale CLIP training experiments (equivalent to the `small` scale in DataComp [1]) with a focus on the data quality, particularly focusing on the misalignment in image-text pairs. The main idea of this experiment is to have two data sources (one *clean* and one *noisy*), similar to Figure 3(b) replicated from [2]. We can sample data with $\eta$, $1-\eta$ proportions to create the training dataset. From this, we can observe the variation in the learnt model's performance with varying $\eta$.
>
> ### References
>
> [1] DataComp: In search of the next generation of multimodal datasets, arxiv:2304.14108.
>
> [2] Data Filtering Networks, arxiv:2309.17425.
>
> [3] CLIPLoss and Norm-Based Data Selection Methods for Multimodal Contrastive Learning, arxiv:2405.19547.
>
> [4] Quality Not Quantity: On the Interaction between Dataset Design and Robustness of CLIP, arxiv:2208.05516.
>
> [5] Improving Multimodal Datasets with Image Captioning, arxiv:2307.10350.
>
> [6] Understanding Multimodal Contrastive Learning and Incorporating Unpaired Data, arxiv:2302.06232.
>
> [7] The Power of Contrast for Feature Learning: A Theoretical Analysis, arxiv:2110.02473.
>
> [8] Benign Overfitting in Linear Regression, arxiv:1906.11300.
>
> [9] Towards a statistical theory of data selection under weak supervision, arxiv:2309.14563.
>
> [10] Iterative Least Trimmed Squares for Mixed Linear Regression, arxiv:1902.03653.
>
> [11] Scaling Laws in Linear Regression: Compute, Parameters, and Data, arxiv:2406.08466.
>
> [12] Optimal Estimation and Rank Detection for Sparse Spiked Covariance Matrices, arxiv:1305.3235.
>
> [13] Going Beyond Nouns With Vision & Language Models Using Synthetic Data, arXiv:2303.17590.
>
> [14] The Implicit Bias of Gradient Descent on Separable Data, arXiv:1710.10345.
>
> [15] In Search of the Real Inductive Bias: On the Role of Implicit Regularization in Deep Learning, arXiv:1412.6614.

---

> > ### Comment · Reviewer_AyoJ · 2025-08-05
> >
> > Thanks for the authors' response. Some of my concerns are addressed.  But I haven't been following the recent literature on this topic very closely, so I would recommend that the AC place more emphasis on the feedback from the other reviewers.

---

> ### Author Response · Authors · 2025-08-07
>
> Thank you for your response and please let us know if there are any specific concerns still remaining, that you would like us to elaborate on!

---

### Official Review · Reviewer_vAKC · 2025-07-03

**Clarity:** 3
**Significance:** 3
**Originality:** 3
**Rating:** 4
**Confidence:** 4

**Summary:**

This paper studies the theory of teacher-based data filtering for multimodal contrastive training. The study analyzes the performance of filtered contrastive learning within the standard bimodal data generation framework. Compared with unfiltered contrastive learning whose error grows in proportion to the inverse of the clean-pair fraction, the teacher-based filtering reduces the dependence. In highly corrupted settings (i.e. very small fraction of clean data), the teacher-based filtering setup achieves an error rate that no longer depends on clean data fraction. Synthetic experiments validates the corollary and theorem, and supporting real experiments from Fang et al. [10] also show that the qualitative conclusions drawn from the theory hold with real image-text data too.

**Questions:**

See questions listed in the above weaknesses section.

**Ethical Concerns:**

["NO or VERY MINOR ethics concerns only"]

**Final Justification:**

I appreciate the value of theoretical analysis. The additional experiments from authors showed the robustness of choosing a reasonable $\theta$. I also read other reviewers' comments and noticed the shared point around practical applicability as summarized by reviewer qb7P. Taking that into consideration, this paper still falls slightly above the acceptance threshold and I would keep my original rating.

**Limitations:**

yes

**Paper Formatting Concerns:**

No Paper Formatting Concerns

**Quality:**

3

**Strengths And Weaknesses:**

Strengths:
- Novel theoretical analysis into data filtering for multimodal contrastive learning. Data filtering has been widely adopted in contrastive learning. This paper analysed theoretically why teacher-based filtering improves multimodal contrastive learning. The surprising finding as mentioned in #2 is novel, that filtering can surpass the “oracle” that uses only clean data when the clean data fraction is low.
- This paper includes validation experiments with synthetic data, showing that results from synthetic experiments align well with the theoretical rates from the corollary and the theorem. They also connect the results with the results from real experiments from Fang et al. [10].


Weaknesses:
- The experimental evidence is limited that the synthetic experiments may not capture all factors that matter in real-world datasets. And the real experiments from Figure 3(b) are created with an artificial setup that mixes CC-12M (high quality) and CommonPool (low quality). As a result, it is unclear how the theoretical analysis translates to actual improvements in downstream tasks or metrics.
- The selected two-stage training procedure is very practical and relies on a lot of empirical iterations. The theory guarantees rely on an optimal threshold $\theta^*$ that depends on unknown data parameters (e.g. clean data fraction). In practice with noisy web-scale data, choosing this threshold is non-trivial to make sure neither too many bad samples are kept nor too many good samples are dropped. It would be nice to demonstrate how the analysis could be useful for practitioners.

---

> ### Author Rebuttal · Authors · 2025-07-30
>
> We thank the reviewer for their detailed and constructive feedback. We appreciate the positive comments about the novelty of the theoretical contributions, and about the suprising finding of the independent-of-$\eta$ regime in Theorem 1. In the following, we address the concerns raised.
>
> ### (1) Experiments are in a controlled setting, theory's applicability to real data and real metrics is unclear.
>
> It is true that both Figures 3(a) and 3(b) explicitly control for the clean fraction $\eta$, and real data in the wild displays other factors of variation (e.g., data quality in the individual modalities -- image and text). Firstly, we'd like to point out that teacher-based filtering is known to work well in the wild too, with real data (e.g., [1, Figure 2], [2,3]).
>
> The controlled experimental setting in this paper was a **deliberate choice to rigorously isolate the dependence** on $\eta$. *Because* we varied only $\eta$ in Figure 3(b), it demonstrates (with real data) that teacher-based filtering can help improve the $\eta$ dependence on the downstream performance. Note the y-axis for Figure 3(b) is `1 - ImageNet Accuracy`, a **key downstream metric** that practitioners care about. This empirically validates what our theory predicts: (1) the fraction of aligned pairs is a critical factor that determines downstream model performance, and (2) teacher-based filtering improves the dependence on $\eta$.
>
> On the theoretical side, we believe it is indeed possible to extend the framework to include other factors of variation for a more complete picture. For instance, one could try to capture that a significant portion of the marginal distributions $D_x, D_{\widetilde{x}}$ are individually noisy beyond the faulty pairings (i.e. many images and captions are simply garbage). This is an interesting and useful direction, but capturing this with the right mathematical assumptions and the subsequent analysis falls outside the scope of the current work.
>
>
> ### (2a) Theory uses the optimal threshold $\theta^{\star}$, how to select $\theta$ practically?
>
> Indeed, the choice of the filtering threshold $\theta$ is an important hyperparameter and requires care in its selection. Let us explain two points in response to this query:
> - We note that the error achieved by teacher-based filtering is fairly **robust to the choice of $\theta$**. Our synthetic experiment in Figure 3(a) was conducted with a fixed, untuned threshold of $\theta=0$. Further, we conducted an experiment measuring the sensitivity of final error with respect to the choice of $\theta$. The experiment details and results are below (###).
> - In practice, this scalar hyperparameter can be **tuned like other commonly used hyperparameters in machine learning**. For example, convex optimization theory states its theorems for the optimal learning rate $\alpha^{\star}$ and in practice the learning rate $\alpha$ needs to be tuned. Similarly, we state our theorem with the optimal threshold $\theta^{\star}$ and in practice the threshold $\theta$ needs to be tuned. In practice, one should tune the size of retained dataset, instead of the parameter $\theta$. A practical rule of thumb is to filter out between 70%~80% of training data, in the case of internet scale datasets [1].
>
> ***### Experiment for error robustness to choice of filtering threshold $\theta$:***
> In the setting of Figure 3(a) with $n = 10000$ samples, we fix $\eta = 0.3$ (in-line with the empirically observed clean fraction in CLIP data [1]) and vary the filtering threshold $\theta$ of the teacher-based filtering. Note that we implicitly vary $\theta$ by explicitly varying the fraction of data retained in the filtering step. The below table shows that the error of teacher-based filtering is relatively flat for values of $\theta$ in the vicinity of the optimal threshold $\theta^{\star}$. An analogous experiment on real data [1, Figure 2] makes a similar observation.
>
> | Fraction of data retained   |  Mean error ($\pm$ 1 $\sigma$) |
> |------------|-------------------------|
> | 1%         |  $28.76 \pm 4.00$ $\text{ } \text{ } \text{ }$ ($\times 1e-4$) |
> | 10%        |  $11.79 \pm 1.20$ $\text{ } \text{ } \text{ }$ ($\times 1e-4$) |
> | 20%        |  $9.85 \pm 1.39$ $\text{ } \text{ } \text{ } \text{ } \text{ }$ ($\times 1e-4$) |
> | **30%**    |  $9.08 \pm 1.15$ $\text{ } \text{ } \text{ } \text{ } \text{ }$ ($\times 1e-4$) |
> | 40%        |  $8.97 \pm 1.09$ $\text{ } \text{ } \text{ } \text{ } \text{ }$ ($\times 1e-4$) |
> | 50%        |  $8.71 \pm 1.05$ $\text{ } \text{ } \text{ } \text{ } \text{ }$ ($\times 1e-4$) |
> | 100%       |  $16.51 \pm 2.03$ $\text{ } \text{ } \text{ }$ ($\times 1e-4$) |
>
> ### (2b) What are the practical implications of the analysis?
>
> Beyond its theoretical contributions, Theorem 1 can offer some useful guidance to practitioners by formalizing the trade-off between data quantity ($n$) and quality (via clean fraction $\eta$). This allows practitioners to move from qualitative intuition to principled, **quantitative decision-making in common data curation scenarios**. The following examples illustrate this in detail:
>
> - **Dataset election:** Consider a practitioner faced with the choice to pick a dataset among many alternatives. Often the choice involves a large, noisy dataset (large $n$, small $\eta$) versus a smaller, curated one (small $n$, large $\eta$). Further, let's assume that the dataset provider also provides an approximation of the data quality in terms of estimated $\eta$, but the actual datasets might be behind a paywall. By plugging these values into the error bounds from Theorem 1 (i.e., comparing which option yields a smaller error), one can make a more informed decision about which dataset is likely to produce a better model.
>
> - **Valuating Data Augmentation:** When considering whether to invest resources in collecting a small amount of clean data to augment a large, noisy corpus, our theorem provides quantitative guidance. It helps estimate the expected performance gain from the increased $\eta$ and increased $n$, allowing a practitioner to assess if the return on investment is worthwhile.
>
> Crucially, the two distinct error regimes identified in Theorem 1 provide non-obvious insights. The $1/\sqrt{\eta}$ error rate in the low-$\eta$ regime suggests that for very noisy datasets, simply increasing data volume ($n$) can be surprisingly effective, a practical takeaway that is not immediately intuitive.
>
> ### References
>
> [1] DataComp: In search of the next generation of multimodal datasets, arxiv:2304.14108.
>
> [2] Data Filtering Networks, arxiv:2309.17425.
>
> [3] CLIPLoss and Norm-Based Data Selection Methods for Multimodal Contrastive Learning, arxiv:2405.19547.
>
> [4] Quality Not Quantity: On the Interaction between Dataset Design and Robustness of CLIP, arxiv:2208.05516.
>
> [5] Improving Multimodal Datasets with Image Captioning, arxiv:2307.10350.
>
> [6] Understanding Multimodal Contrastive Learning and Incorporating Unpaired Data, arxiv:2302.06232.
>
> [7] The Power of Contrast for Feature Learning: A Theoretical Analysis, arxiv:2110.02473.
>
> [8] Benign Overfitting in Linear Regression, arxiv:1906.11300.
>
> [9] Towards a statistical theory of data selection under weak supervision, arxiv:2309.14563.
>
> [10] Iterative Least Trimmed Squares for Mixed Linear Regression, arxiv:1902.03653.
>
> [11] Scaling Laws in Linear Regression: Compute, Parameters, and Data, arxiv:2406.08466.
>
> [12] Optimal Estimation and Rank Detection for Sparse Spiked Covariance Matrices, arxiv:1305.3235.
>
> [13] Going Beyond Nouns With Vision & Language Models Using Synthetic Data, arXiv:2303.17590.
>
> [14] The Implicit Bias of Gradient Descent on Separable Data, arXiv:1710.10345.
>
> [15] In Search of the Real Inductive Bias: On the Role of Implicit Regularization in Deep Learning, arXiv:1412.6614.

---

> > ### Comment · Reviewer_vAKC · 2025-08-07
> >
> > I thank the authors for the response. I appreciate the value of theoretical analysis. The additional experiments showed the robustness of choosing a reasonable $\theta$. I also read other reviewers' comments and noticed the shared point around practical applicability as summarized by reviewer qb7P.  I will update the final justification accordingly.

---

> ### Author Response · Authors · 2025-08-07
>
> As <48 hours are left in the discussion period, we would like to follow-up on this. Please let us know if you have any unresolved questions / further concerns and we would be happy to answer!

---

### Official Review · Reviewer_cV4q · 2025-07-04

**Clarity:** 3
**Significance:** 3
**Originality:** 3
**Rating:** 4
**Confidence:** 4

**Summary:**

The paper investigates the benefits of using teacher-based data filtering in multimodal contrastive learning.  It provides a theoretical analysis of how data filtering can improve the performance of contrastive learning models when dealing with noisy, web-scale datasets.  The authors characterize the performance of filtered contrastive learning under a standard bimodal data generation model and show that teacher-based filtering can significantly reduce the error dependence on the fraction of clean data.  Specifically, they demonstrate that the error rate can improve from 1/η to 1/√η in the large η regime and become independent of η in the small η regime, where η represents the fraction of correctly matched modality pairs.  The paper also includes synthetic experiments that validate the theoretical findings and highlight the effectiveness of teacher-based filtering in enhancing model robustness and accuracy.  The contributions provide a deeper understanding of the role of data quality in multimodal representation learning and offer insights into practical data curation strategies.

**Questions:**

see weaknesses.

**Ethical Concerns:**

["NO or VERY MINOR ethics concerns only"]

**Final Justification:**

The author has addressed some of my concerns and I will maintain the score.

**Limitations:**

see weaknesses.

**Quality:**

3

**Strengths And Weaknesses:**

Strength：
1. This paper provides a theoretical explanation of the benefits of data filtering in multimodal contrastive learning. By analyzing the error dependence of contrastive learning before and after filtering, it proves the effectiveness of teacher-model-guided data filtering in improving data quality.

2. This paper shows that when the data quality is poor, the teacher-guided data filtering can achieve an error rate independent of data quality, which breaks the dependence on data quality in traditional contras - tive learning and provides a new perspective for dealing with large-scale noisy data.

The theoretical results are verified by synthetic experiments. The experimental results are consistent with the theoretical analysis, which further confirms the performance improvement of data filtering guided by teacher model in different data quality scenarios, and provides guidance for data filtering strategies in practical applications.

Weakness：
1.For some emerging techniques related to data filtering, such as diffusion model-based data augmentation methods, the discussion may not be in-depth enough. These methods may have potential advantages when dealing with noisy data, but the paper does not compare the differences and connections between these methods and teacher model-guided data filtering in multimodal contrastive learning in detail.

2.The paper assumes a linear contrastive learning framework and a Gaussian noise model, which may be limited in practical applications. For example, the noise in real data may have a more complex distribution, and the relationship between multimodal data may not be completely linear.

3.The teacher-model-guided data filtering methods mentioned in the paper rely on a pre-trained teacher model, which may require additional computational resources and data to train and may not be practical enough for resource-limited scenarios. And how much impact does different pre-trained teacher models have on the results? Moreover, the paper does not discuss in detail how to select the optimal filtering threshold, which may be a critical issue in practical applications.

4.To better demonstrate the effectiveness of the approach, one may consider conducting experiments on real-world datasets to evaluate the performance in more complex and diverse data environments.

5. In addition, only the influence of data quality on model performance is considered in the experiment, and the influence of other factors (such as data volume, model architecture, etc.) on the results is not explored. The experiment can be further extended to analyze the interaction between these factors and data quality.

---

> ### Author Rebuttal · Authors · 2025-07-30
>
> We thank the reviewer for their detailed and constructive feedback, and we appreciate the positive comments about the theoretical contributions of the paper. In the following, we address the concerns raised.
>
> ### (1) Did not compare with synthetic data augmentation methods.
>
> The **empirical** study of data curation for multimodal learning has progressed rapidly, from (i) filtering data using a teacher model (e.g., [2, 3]), to (ii) augmenting with synthetic data (e.g., [13]); and finally to (iii) the combination of the above [5]. Meanwhile, **theoretical** understanding of the above progression has lagged significantly behind. Our contribution is to build a foundational understanding of the first phase, i.e., teacher-based filtering.
>
> However, we believe some of the models we study can be extended to study the second and third phases. For instance, synthetically generated 'clean' pairs could be modeled as new samples $(y,\widetilde{y})$ being augmented to the original dataset comprising of $(x, \widetilde{x})$, with $y$ and $x$ following different distributions through distinct priors on the latent $z$ and/or different properties on the noise $\xi$. By controlling the distribution of $(y,\widetilde{y})$, one could attempt to explain some empirically observed phenomena, such as adding synthetic data helps only up to some point, and either plateaus or starts to hurt performance. This is likely a fruitful direction for future research, but a rigorous investigation of these more complex scenarios is beyond the scope of this paper.
>
> ### (2) Real data may not satisfy the assumptions about linearity and Gaussian noise.
>
> The goal of our work is to provide theoretical justification for the common empirical observation that data filtering using a teacher CLIP model improves performance. The setup with Linearity and Gaussian noise provides a **tractable setting** to study this phenomenon, and is motivated by **past works** that make the same assumptions [6, 7]. Beyond CLIP filtering, **other phenomena in deep learning** have also been successfully characterized in the linear setup, e.g., benign overfitting [8] (also known as double descent), scaling laws [11], and implicit bias [14, 15]. These topics are foundations of what we call the Theory of Deep Learning, and are textbook materials in courses taught at top institutions. Overall, the linear assumption, though not directly applicable in some real cases, provides a useful abstraction that often **preserves the primary problem characteristics** and helps to build a deeper understanding through a theoretical lens. We believe our work (and the assumptions we have), although simple, is within the typical assumptions made in theoretical anlayses of deep learning, which has proven to be productive and insightful (e.g., [8, 11, 14, 15]).
>
> ### (3a) Reliance on a pre-trained teacher model which requires additional data and compute.
>
> Our analysis **does not assume the need for an external pre-trained teacher model**. As mentioned in Lines 60-61 and detailed in Algorithm 1, our framework involves training the teacher also using the given (noisy) dataset. That is, the "Train-Filter-Train" approach uses the same initial dataset, making it a **self-contained approach** that reflects real-world usage.
>
> One subtle difference is regarding sample splitting. For theoretical convenience (as mentioned in Lines 204-206), we split the initial dataset into two halves of $n/2$ to leverage independence of random variables. In practice, such data spliting is not necessary and it works just fine to use the entire dataset to train the teacher model, and filter the same dataset to train the student model.
>
> On the other hand, as you correctly pointed out, the approach does require additional data and compute (in our case, it is 2x since teacher and student require equal resources since they are trained on $n/2$ samples each). This need for additional data/compute aligns with practical usage (e.g., [1, 2, 3]).
>
> ### (3b) How to select optimal theta practically.
>
> This is a great question, thank you for raising this. We would like to expand on the choice of the filtering threshold $\theta$ by explaining two points:
> - We note that the error achieved by teacher-based filtering is fairly **robust to the choice of $\theta$**. Our synthetic experiment in Figure 3(a) was conducted with a fixed, untuned threshold of $\theta=0$. Further, we conducted an experiment measuring the sensitivity of final error with respect to the choice of $\theta$. The experiment details and results are below (###).
> - In practice, this scalar hyperparameter can be **tuned like other commonly used hyperparameters in machine learning**. For example, convex optimization theory states its theorems for the optimal learning rate $\alpha^{\star}$ and in practice the learning rate $\alpha$ needs to be tuned. Similarly, we state our theorem with the optimal threshold $\theta^{\star}$ and in practice the threshold $\theta$ needs to be tuned. In practice, one should tune the size of retained dataset, instead of the parameter $\theta$. A practical rule of thumb is to filter out between 70%~80% of training data, in the case of internet scale datasets [1].
>
> ***### Experiment for error robustness to choice of filtering threshold $\theta$:***
> In the setting of Figure 3(a) with $n = 10000$ samples, we fix $\eta = 0.3$ (in-line with the empirically observed clean fraction in CLIP data [1]) and vary the filtering threshold $\theta$ of the teacher-based filtering. Note that we implicitly vary $\theta$ by explicitly varying the fraction of data retained in the filtering step. The below table shows that the error of teacher-based filtering is relatively flat for values of $\theta$ in the vicinity of the optimal threshold $\theta^{\star}$. An analogous experiment on real data [1, Figure 2] makes a similar observation.
>
> | Fraction of data retained   |  Mean error ($\pm$ 1 $\sigma$) |
> |------------|-------------------------|
> | 1%         |  $28.76 \pm 4.00$ $\text{ } \text{ } \text{ }$ ($\times 1e-4$) |
> | 10%        |  $11.79 \pm 1.20$ $\text{ } \text{ } \text{ }$ ($\times 1e-4$) |
> | 20%        |  $9.85 \pm 1.39$ $\text{ } \text{ } \text{ } \text{ } \text{ }$ ($\times 1e-4$) |
> | **30%**    |  $9.08 \pm 1.15$ $\text{ } \text{ } \text{ } \text{ } \text{ }$ ($\times 1e-4$) |
> | 40%        |  $8.97 \pm 1.09$ $\text{ } \text{ } \text{ } \text{ } \text{ }$ ($\times 1e-4$) |
> | 50%        |  $8.71 \pm 1.05$ $\text{ } \text{ } \text{ } \text{ } \text{ }$ ($\times 1e-4$) |
> | 100%       |  $16.51 \pm 2.03$ $\text{ } \text{ } \text{ }$ ($\times 1e-4$) |
>
> ### (4) Limited experimental validation.
>
> The teacher-based filtering (Figure 1c, Algorithm 1) has been relatively well-studied empirically, for instance in [1, 2, 3].  The focus of this paper is to theoretically analyze this approach, and demonstrate a provable benefit. Since this involved no new algorithmic modifications, we relied on synthetic experiments (Figure 3a) and reused a real-data experiment (Figure 3b), instead of a suite of new experiments.
>
> That said, we plan to run small-scale CLIP training experiments (equivalent to the `small` scale in DataComp [1]) with a focus on the data quality, particularly focusing on the misalignment in image-text pairs. The main idea of this experiment is as follows:
> - Similar to Figure 3(b) replicated from [2], we would like to have two data sources: one *clean* and one *noisy*. We can sample data with $\eta$, $1-\eta$ proportions to create the training dataset. From this, we can observe the variation in the learnt model's performance with varying $\eta$.
> - Further, we plan to study the effect of changing the clean data source (similar to the idea in [4]), model architecture and size, and the total dataset size.
>
> ### (5) Other factors like data volume, model architecture, etc. not analyzed.
>
> The central research question of our paper is to isolate and understand the specific role of the clean data fraction $\eta$, a critical but less understood variable in the context of web-scale datasets. Our goal is to show that the train-filter-train paradigm achieves better dependence in $\eta$ compared to the baseline of [6]. To study its effect rigorously, we deliberately kept other factors constant in our experiments. However, we agree that factors like data volume and model architecture are important variables in model performance (note that the theory does capture the dependence on data volume, i.e. number of samples $n$).
>
> ### References
>
> [1] DataComp: In search of the next generation of multimodal datasets, arxiv:2304.14108.
>
> [2] Data Filtering Networks, arxiv:2309.17425.
>
> [3] CLIPLoss and Norm-Based Data Selection Methods for Multimodal Contrastive Learning, arxiv:2405.19547.
>
> [4] Quality Not Quantity: On the Interaction between Dataset Design and Robustness of CLIP, arxiv:2208.05516.
>
> [5] Improving Multimodal Datasets with Image Captioning, arxiv:2307.10350.
>
> [6] Understanding Multimodal Contrastive Learning and Incorporating Unpaired Data, arxiv:2302.06232.
>
> [7] The Power of Contrast for Feature Learning: A Theoretical Analysis, arxiv:2110.02473.
>
> [8] Benign Overfitting in Linear Regression, arxiv:1906.11300.
>
> [9] Towards a statistical theory of data selection under weak supervision, arxiv:2309.14563.
>
> [10] Iterative Least Trimmed Squares for Mixed Linear Regression, arxiv:1902.03653.
>
> [11] Scaling Laws in Linear Regression: Compute, Parameters, and Data, arxiv:2406.08466.
>
> [12] Optimal Estimation and Rank Detection for Sparse Spiked Covariance Matrices, arxiv:1305.3235.
>
> [13] Going Beyond Nouns With Vision & Language Models Using Synthetic Data, arXiv:2303.17590.
>
> [14] The Implicit Bias of Gradient Descent on Separable Data, arXiv:1710.10345.
>
> [15] In Search of the Real Inductive Bias: On the Role of Implicit Regularization in Deep Learning, arXiv:1412.6614.

---

> > ### Comment · Reviewer_cV4q · 2025-08-05
> >
> > Thanks for the responses. I will maintain my score.

---

### Comment · Reviewer_qb7P · 2025-08-06
**Practical Applicability of the Theoretical Findings**

[Reviewer cV4q](https://openreview.net/forum?id=2xDdVkWrqq&noteId=lUYlSU6iH4) mentioned that the linear contrastive learning framework with a Gaussian noise model may be limited in practical applications and suggested to conduct experiments on real-world datasets. [Reviewer vAKC](https://openreview.net/forum?id=2xDdVkWrqq&noteId=9dHaLlqo1M) mentioned that it is unclear how the theoretical analysis translates to actual improvements in downstream tasks or metrics. [Reviewer AyoJ](https://openreview.net/forum?id=2xDdVkWrqq&noteId=tEUGw71Nd6) criticized that the theoretical results are not sufficiently linked to real-world applications. [Reviewer qb7P](https://openreview.net/forum?id=2xDdVkWrqq&noteId=lMf2Ga6TeR) was wondering if the simplifying assumptions create a large gap with a realistic setup and that the findings might be hard to translate. [Reviewer M3vJ](https://openreview.net/forum?id=2xDdVkWrqq&noteId=2gn74FaKJq) finally remarked that the restrictive assumptions simplify analysis but limit applicability to realistic settings.

From all these reviews, I think the point about **practical applicability of the findings is key to the review of this paper**, and for this reason I think it makes sense to discuss these points separately from the individual reviews, hoping that the authors and other reviewers join for a fruitful discussion.

---

> ### Comment · Reviewer_qb7P · 2025-08-06
>
> In response to these points, the authors have replied with the following arguments (copied from the [rebuttal to qb7P](https://openreview.net/forum?id=2xDdVkWrqq&noteId=TkuFczv3As)):
>
> > **1. Why the linear setup analysis is useful?**
>
> > The goal of our work is to provide theoretical justification for the common empirical observation that data filtering using a teacher CLIP model improves performance. The setup with Linearity and Gaussian noise provides a **tractable setting** to study this phenomenon, and is motivated by **past works** that make the same assumptions [6, 7]. Beyond CLIP filtering, **other phenomena in deep learning** have also been successfully characterized in the linear setup, e.g., benign overfitting [8] (also known as double descent), scaling laws [11], and implicit bias [14, 15]. These topics are foundations of what we call the Theory of Deep Learning, and are textbook materials in courses taught at top institutions. Overall, the linear assumption, though not directly applicable in some real cases, provides a useful abstraction that often **preserves the primary problem characteristics** and helps to build a deeper understanding through a theoretical lens. We believe our work (and the assumptions we have), although simple, is within the typical assumptions made in theoretical anlayses of deep learning, which has proven to be productive and insightful (e.g., [8, 11, 14, 15]). For a detailed justification of using linear models to study deep neural networks, we refer to a nice lecture note [16] by Andrea Montanari at Stanford.
>
> > **2. What are the practical implications?**
>
> > Beyond its theoretical contributions, Theorem 1 also offers some useful guidance to practitioners by formalizing the trade-off between data quantity ($n$) and quality (via clean fraction $\eta$). Our analysis allows practitioners to move from qualitative intuition to principled, **quantitative decision-making in common data curation scenarios**. The following two examples illustrate this in detail:
> > - **Dataset election**: Consider a practitioner faced with the choice to pick a dataset among many alternatives. Often the choice involves a large, noisy dataset (large $n$, small $\eta$) versus a smaller, curated one (small $n$, large $\eta$). Further, the high-level aggregates (like $n$ and $\eta$ or approximate $\eta$) are usually available, but the actual datasets might be behind a paywall. By plugging these values into the error bounds from Theorem 1 (i.e., comparing which option yields a smaller error), one can make a more informed decision about which dataset is likely to produce a better model.
> > - **Valuating Data Augmentation**: When considering whether to invest resources in collecting a small amount of clean data to augment a large, noisy corpus, our theorem provides quantitative guidance. It helps estimate the expected performance gain from the improved $\eta$ and increased $n$, allowing a practitioner to assess if the return on investment is worthwhile.
>
> > Crucially, the two distinct error regimes identified in Theorem 1 provide non-obvious insights. The $1/\sqrt{\eta}$ error rate in the low-$\eta$ regime suggests that for very noisy datasets, simply increasing data volume ($n$) can be surprisingly effective, a practical takeaway that is not immediately intuitive.

---

> > ### Comment · Reviewer_qb7P · 2025-08-06
> >
> > The experimental Section 7 of the paper shows two things:
> >
> > 1. Fig 3(a) verifies the bounds with synthetic data. This lends support to the mathematical derivations, but says nothing about the applicability in a realistic setup. The trend for the "no filtering" indeed  has a slope of 1, while the teacher-based filtering quickly deviates from the expected slope of 0.5 – it would be interesting to know if the trend is a better fit if more synthetic data was generated, as suggested by the authors in the caption of Fig 3.
> >
> > 2. Fig 3(b) shows that also in a realistic setup, the error slope is steeper without filtering, but the quantitative findings cannot be verified in this setting because the error metric is different (practical 1-accuracy vs. the theoretical chordal distance) – or, as the authors state, "the qualitative message of the improved dependence, derived from the linear modeling theory, holds true with nonlinear models as well".
> >
> > I think the interesting question is though not if filtering improves the dependence on η, which is already known from previous work, but **whether the theoretical insights from the submission** (i.e. the different bounds with and without filtering), **can be used for quantitative decision-making**, or whether they are artifacts induced by the simplifying assumptions.
> >
> > If the authors could provide examples where their predictions are used for dataset election or data augmentation (as mentioned in their reply above), this would indeed give credence to the point that the predictions can be put to practical use.
> >
> > Similarly, running experiments on DataComp (as mentioned on the rebuttal to [M3vJ](https://openreview.net/forum?id=2xDdVkWrqq&noteId=2gn74FaKJq)) could also underline the practical usefulness of the derived bounds.

---

> ### Author Response · Authors · 2025-08-08
> **On point 1**
>
> We highly appreciate Reviewer qb7P for initiating this discussion.
>
> We would first like to briefly clarify the point: "trend is a better fit if more synthetic data was generated, as suggested by the authors in the caption of Fig 3".
>
> We alluded to the required condition of $n \gtrsim 1/\eta^2$ in the caption of Fig 3 **not** to mean that more synthetic data would "improve the fit". We meant that more data would make the trend hold for even smaller values of $\eta$ (which corresponds to the right side of the two curves in Fig 3a). This was in connection with the error bars growing at the rightmost point of Fig 3a (corresponding to $\eta = 1e-3$). Adding more synthetic data would get the curve to be smooth even at $\eta = 1e-3$ and the confidence interval smaller.

---

> ### Author Response · Authors · 2025-08-08
> **On point 2 and the remainder of the comments**
>
> We agree with the reviewers that practical implications are important, but we would like to better explain the main contributions of our theoretical results. We respectfully argue that our theoretical analysis is still important and significant, even if it might not directly lead to a quantifiable practical implication. In other words, we would like to take this opportunity to answer **"Why are we theoretically studying stylized models?"**
>
> 1. First of all, the **quantitative decision making** examples of *Data Election* and *Valuating Data Augmentation* are meant to be plausible scenarios where theoretical insights could help quantitatively, but we believe they are not the main contributions of this paper.
>
> 2. We also agree with the reviewers that just the empirical fact that filtering helps in CLIP training is not the main point of this paper (or our analysis) either.
>
> 3. The importance and significance of our results come from the fact that **we provide an understanding of WHY filtering helps**. For the first time, we are able to give some answers to the question: "why does filtering in multi-modal representation learning give improved performance?". We agree that our explanation is not perfect; it is theoretical, it is on stylized models, and it still leaves many questions open. However, we humbly argue that this is how discoveries and advances happen in theoretical research. They happen slowly, over multiple papers, building upon one another.
>
> 4. If we agree that (principled) explanations and (deeper) understandings can be important contributions, then our main contribution is the discovery that "filtering helps when there are **mis-matched modalities** in the samples, because filtering removes those misaligned examples". This is opposed to how people understood filtering previously (e.g., [9, 10] and many others), which was that filtering removes more noisy examples (i.e., points with "bad" image or text). Without our analysis, the only **principled** explanation of why filtering works in CLIP training would have been that "noisy data are removed in filtering", which is not the complete picture. Of course one could empirically test such hypotheses and arrive at similar conclusions. But we are hoping that the reviewers also see the value in explaining (even already known phenomenon) theoretically. Even if a fact is known, providing a theoretical analysis gives principled explanations and perhaps deeper understanding. In short, **we are the first to explain in a principled way why filtering works, through the lens of mis-alignment**.
>
> 5. One way to interpret the significance of our results might be by comparing it to related theoretical results. Closest to our work (and the main motivations for our work) are [7] and [6]. Ji et al [7] studied the spiked covariance model like we do, but focused on the question: "Is there an advantage in **unimodal** contrastive learning over other feature learning methods (autoencoders and GANs)?". In particular, [7] did not consider noisy data or the mis-alignment problem at all, which is highly relevant for practical large-scale datasets. Following up on this, Nakada et al [6] considered the mis-alignment problem in the **multimodal** setup, but only studied the no filtering algorithm in [6, Theorem 3.1] and established the $1/\eta$ rate for our baseline (Corollary 1). In this progression, the analysis of data filtering, and how it can provably achieve a better rate by filtering out samples with mis-matched modalities, is our main contribution. In this series of works, i.e., [7]-->[6]-->[our submission], we believe our work provides explanation of a more practically interesting phenomena of "filtering helps significantly in CLIP training" with emphasis on providing the reason for such gain: mis-aligned modalities. **Note that this explanation is particularly interesting since mis-aligned modalities can only be observed in multi-modal settings, and we are the first to study the gains of filtering in this setting.**

---

> ### Author Response · Authors · 2025-08-08
> **On running more experiments**
>
> We would like to add one more point:
>
> 6. Training datacomp CLIP is outside the scope of this theoretical paper. However, we could do more extensive experiments in a setting that is more complex than our synthetic setup, but still simpler than DataComp-CLIP training. This is in the spirit of [9], which received an honorable mention in the ICLR 2024 outstanding paper award. Let us explain what [9] is about first. Kolossov et al [9] studies a stylized problem of generalized linear regression (with linear models on a single modality) and provides theoretical analyses of running a single linear regression on full data vs. running linear regression on filtered data. It is shown that, perhaps surprisingly, filtering out the data can provide improvement, even in linear regression. The "real" experiments in [9] are still linear model training, but on real data. Extensive experimental results confirm their theoretical findings. In a similar spirit, we believe we could also achieve similar experimental confirmation by studying the following setting: linear representation learning on real data. The main challenge is finding a dataset where meaningful representations can be learned with a linear model. We could take existing encoders for images and texts, and just train a linear layer on those embeddings, for example. Note that in the case of [9], finding datasets where linear regression and logistic regression gives meaningful predictions is not too hard. We plan to add such real but still manageable experiments to confirm our theoretical findings.
>
> We thank Reviewer qb7P again for creating this discussion. We hope this resolves some concerns about the importance of the result. Please let us know if there are any more questions!

---

> > ### Comment · Reviewer_qb7P · 2025-08-08
> >
> > I would like to thank the authors again for their detailed and informative reply.
> >
> > I appreciate the further clarifications on the first point; they have resolved my initial questions.
> >
> > I concur with the authors' assessment that their work is well-situated within the existing literature, a point I also noted in my initial review.
> >
> > Regarding the practical applicability of the findings, I understand the challenges in bridging the theoretical analysis based on linear assumptions with real-world complexities. I acknowledge the authors' sentiment that conducting experiments on a dataset like DataComp falls outside the scope of this paper. A more detailed discussion of these limitations in the manuscript would be beneficial for clarity (e.g. what kind of practical answers can or cannot be answered with the derived bounds). The suggestion to **include experiments that try to bridge the gap partially**, such as training a linear layer on top of existing representations from real data, is an excellent one, and I believe that incorporating such results **would substantially strengthen the paper's contributions**.

---

> > > ### Author Response · Authors · 2025-08-08
> > >
> > > We thank the reviewer for leading the discussions and the constructive feedback throughout the interaction. We plan to discuss the challenges we face in running large-scale experiments and bridge the gap with intermediate-scale experiments.

---

### Decision · Program_Chairs · 2025-09-17

**Decision:**

Accept (poster)

**Comment:**

The paper was reviewed by 5 experts in the field. After extensive discussions, all reviewers reached a consensus to accept the paper. The AC reads the paper, the reviews, and the authors' responses carefully. The AC shares a similar concern with reviewer qb7P on the practical implications of the findings in this work. However, the AC understands the difficulty of dealing with the complexity of modern deep neural networks in a theoretical work. Overall, the AC believes this paper is a solid theoretical work that provides a nice way of explaining the effectiveness of using model-based data filtering CLIP-style contrastive models.